# Mechanical gating of tendon fibrogenic transcription in systemic sclerosis

**Amro A. Hussien** [1,2], **Robert Knell**[1], **Stefania L. Wunderli**[1,2], **Matthew R. Aronoff**[3], **Barbara Niederoest**[1,2], **Florian Renoux**[4], **Amela Hukara**[4], **Jasper Foolen**[1,2,5], **Helma Wennemers** [3], **Oliver Distler** [4] **& Jess G. Snedeker** [1,2] ✉

Fibrosis arises from aberrant tissue repair in systemic sclerosis and is characterized by progressive stiffening across multiple tissues, including tendons. How altered tissue mechanics sustain fibrotic remodeling remains poorly understood, in part because of limited experimental models that capture key biophysical features of fibrosis. Here we develop a modular cantilever-based mechano-culture platform that enables tendon-like constructs to be maintained under controlled static tension. We show that elevated matrix tension induces fibroblast-to-myofibroblast activation and scar-like phenotypes in vitro. Analysis of preclinical and clinical models of systemic sclerosis reveals that increased three-dimensional matrix stiffness inversely correlates with transcription of major profibrotic collagens, while positively regulating genes associated with stromal–immune interactions. Co-culture with bone marrow–derived macrophages overrides tension-dependent suppression of matrix gene expression, suggesting that immune cues can supersede mechanical checkpoints. These findings demonstrate how tissue mechanics orchestrates reciprocal interactions between stromal and immune compartments to drive progressive fibrosis, and establish a reductionist platform for dissecting mechano-regulatory pathways in fibrotic diseases.

Systemic sclerosis (SSc) is a chronic immune-mediated rheumatic disease, characterized by microvascular damage, non-resolving inflammation, and progressive fibrosis of connective tissues[1–4]. The disease represents significant clinical and socioeconomic burdens to patients, their caretakers and healthcare systems[5–7]. Globally, the prevalence of SSc is estimated to be around 10–30 cases per 100,000 population[8]. Despite significant clinical advances in managing systemic sclerosis, musculoskeletal comorbidities contribute to the overall reduced quality of life of SSc patients[9,10]. Tendon involvement occurs early in the development of SSc, and it is one of the strongest predictive factors of overall disease progression[11,12]. While the aetiopathogenesis of SSc remains poorly understood, a complex picture of genetic susceptibility, small vessel damage, sustained autoimmune

inflammatory and profibrotic activation and self-amplifying dysfunctional repair responses is slowly coming into focus[9,13]. Nevertheless, relevant insights into molecular mechanisms underpinning tendon involvement are scarce, and further progress is potentially hindered by ethical concerns limiting access to tendon biopsies or scarcity in experimental models that mimic key aspects of the tendon fibrotic niche[11,14,15].

The hallmark of SSc is progressive fibrosis that is dominated by persistent activation of resident stromal cells, and excessive deposition and crosslinking of the extracellular matrix (ECM). This process is thought to ultimately mediate self-amplifying activation loops accompanied by irreversible replacement of tissue stroma with a rigid, mechanically-strained connective tissue[16–19]. How extracellular matrix

[1]Institute for Biomechanics, ETH Zurich, Zurich, Switzerland. [2]Balgrist University Hospital, University of Zurich, Zurich, Switzerland. [3]Laboratory of Organic Chemistry, ETH Zurich, Switzerland. [4]Center of Experimental Rheumatology, Department of Rheumatology, University Hospital Zurich, University of Zurich, Zurich, Switzerland. [5]Department of Biomedical Engineering, Eindhoven University of Technology, Eindhoven, Netherlands. ✉e-mail: snedeker@ethz.ch

mechanical cues contribute to the initiation or progression of fibrosis is an area of intense investigation[19–21]. In tendon, repetitive mechanical overloading disrupts the underlying homeostatic tension that is critical for maintaining extracellular matrix structure and function[22]. Injury responses are intimately linked to cell- and tissue-level mechanical homeostasis, with the extracellular matrix providing biophysical signals for appropriate remodeling and resolution responses[22–24].

For instance, it is now evident that matrix stiffening precedes the onset of clinical fibrosis[21,25,26]. Whereas matrix mechanical cues are instrumental in the successful transition from activated wound healing programs to successful resolution, persistent aberrant matrix stiffening is implicated in instilling fibroblast activation and pathological tissue remodeling[20]. How these processes play out in the context of tendon involvement in SSc is unknown.

Engineered biomimetic in vitro tissue models provide an attractive avenue to explore the interplay of matrix mechanical cues and progression of fibrosis in a precise and controlled manner[27]. The most widely used technique for tuning matrix stiffness while simultaneously characterizing the magnitude of cell-generated forces involve measuring deformations of planar two-dimensional (2D) membranes or 3D hydrogel substrates as a proxy for physical forces[28,29]. One major limitation of such approaches is the inherent assumption that more deformation translates to more traction forces, in addition to the confounding coupling of several multiple biophysical properties (e.g., stiffness and porosity). An alternative approach is combining synthetic elastic substrates of tunable stiffnesses and traction force microscopy (TFM). In this method, cells are seeded on deformable, non-degradable 2D planar surfaces that are patterned with small trackable markers, e.g., fluorescent beads or nanoparticles[30,31]. Cells and the underlying substrate are sequentially imaged in a stressed (contractile) state, and again after cytoskeletal relaxation. The two images are analyzed to track the beads displacements and to quantify the traction forces that are needed to displace the fluorescent markers. Although the TFM methods are powerful in resolving forces with a subcellular spatial resolution, it requires significant experimental and computational resources that are complex, laborious and often not accessible to laboratories lacking engineering expertise[32]. Furthermore, cells in 2D planar in vitro models are highly polarized and often interact with the underlying substrate only on one side while in contact with cell culture medium on the other end[33].

Hence, such reductionist approaches do not fully capture the complexity between form and function of connective tissues like tendons. In vivo, tendon stromal cells reside in highly aligned 3D niches that are rich in collagens and other proteinaceous components, such as proteoglycans, while being exposed to substantial levels of mechanical tension. Cellular tension in 3D environments can be modified by changing the intrinsic mechanical properties of the environment within which cells are encapsulated (e.g., hydrogel intrinsic stiffness), or by manipulating the extrinsic boundary stiffness by constraining the edges of isotropic fibrillar hydrogels (e.g., collagen biopolymers)[34,35]. Physically-constrained hydrogels culture models offer an attractive approach to reconstitute this complex mechano-reciprocity of tendon cell-ECM interactions in vitro[36–40]. In this approach, stromal cells are embedded in ECM-derived hydrogels that are anchored to rigid pillars to geometrically limit isotropic contractions of the matrix, i.e., uniform compactions in all directions. As a result, cell-generated contractile forces compact the microtissue around the anchoring pillars, deflecting the free ends of each pillar inwards toward the center and driving cell-matrix alignment longitudinally along the axis of principal stresses. The shape of microtissues can be controlled by adjusting the geometry and spacing of the rigid pillars[41], while the tissue contractile forces can be quantified using beam theory or Hooke's law[42]. Changing the compliance of the constraining pins by making it highly compliant (i.e., deflectable) or minimally compliant (i.e., rigid with minimal deflection), it is possible to tune the overall resistance of the matrix and the *emergent effective stiffness*[36,43]. In turn, this resistance regulates the cellular contractility and the mechanical forces that the cells can exert on the hydrogel matrix.

Here, we hypothesize that elevated matrix tension mediates the persistence of fibrotic remodeling in SSc tendons. First, we engineered a modular mechano-culture platform that enables one to easily vary static mechanical tension of tendon-like mimetics, independently of changing bulk tissue properties, thus emulating the complex mechano-reciprocity of tendon structure-function relationship.

Next, we characterized tendon involvement in a transgenic mouse model approximating the systemic phenotypes of human SSc and used these models to explore the fibrogenic stromal-immune interactions in vitro. We conclude that mechanically-driven, feed-forward loops are likely to be a driving feature of tendon pathology in SSc.

## Results

### Development of mechano-culture tissue model for tendon mechanobiology

Tendons are mechanically-anchored fibrous tissues, wherein resident stromal cells are longitudinally aligned along the axis of mechanical tension[44,45]. To recapitulate the tensional state of tendons in vitro, we engineered a modular, mechano-culture platform that supports manipulation and monitoring of cellular traction forces generated by 3D tendon-like hydrogel constructs under easily variable mechanical tension (Fig. 1A, B). We constructed the device by assembling off-the-shelf components that can conveniently sourced by non-engineering laboratories, thus circumventing the need for the often inaccessible methods such as photolithography or other microfabrication techniques[32].

The device consists of two interlocked 12-well plates that are separated by a spacer. It is accessible from all sides to facilitate cell culture media replenishment and gas exchange. The upper plate contains long vertical medical-grade steel posts, which firmly lock into an optically clear bottom plate. To provide hydrogel anchoring points, a ceramic bead is glued on top of each post. When the plates are locked together, the two vertical posts in each well are contained within an elliptical hydrogel reservoir. This design feature supports anchoring the 3D tendon-like tissues around the posts, while simultaneously maintaining tissue constructs in proximity to the floor of the bottom well for microscopic observations (Fig. 1A).

In general, hydrogel stiffness can be modulated by increasing the monomer amounts or by increasing its crosslinking density. However, both approaches eventually manipulate bulk material properties; i.e., ligand density and presentation, and porosity. To decouple the effects of hydrogel rigidity from other bulk properties, we modulated the boundary rigidity of tethered collagen hydrogels by varying the base:curing agent ratios of the anchoring PDMS mats (Fig. 1B). This strategy facilitates tuning the hydrogel effective stiffness (i.e., increased resistance felt by cells), while keeping initial collagen amounts and crosslinking density constant.

First, we performed numerical simulations to explore which design parameters may have the largest impact on the steel post spring constant. Using finite element analysis (FEA) methods, we simulated the post transverse deflections in response to a 1000 μN force acting on the surface of the bead, while varying post geometries, PDMS mat properties and boundary conditions (Fig. 1C). The cantilever post was modeled as a stainless steel material, whereas the bead and PDMS layer were set as linear elastic materials. The FEA model predicted that the PDMS stiffness and the active length of the post (i.e., the length from the surface of PDMS mat to the post's free end, or in other words the PDMS thickness) have the strongest influence on the deflection of the posts (Fig. 1D). Since the active length of the post is restricted by the design requirements and well dimensions, we reasoned that varying the PDMS stiffness is the best strategy to modulate boundary mechanics.

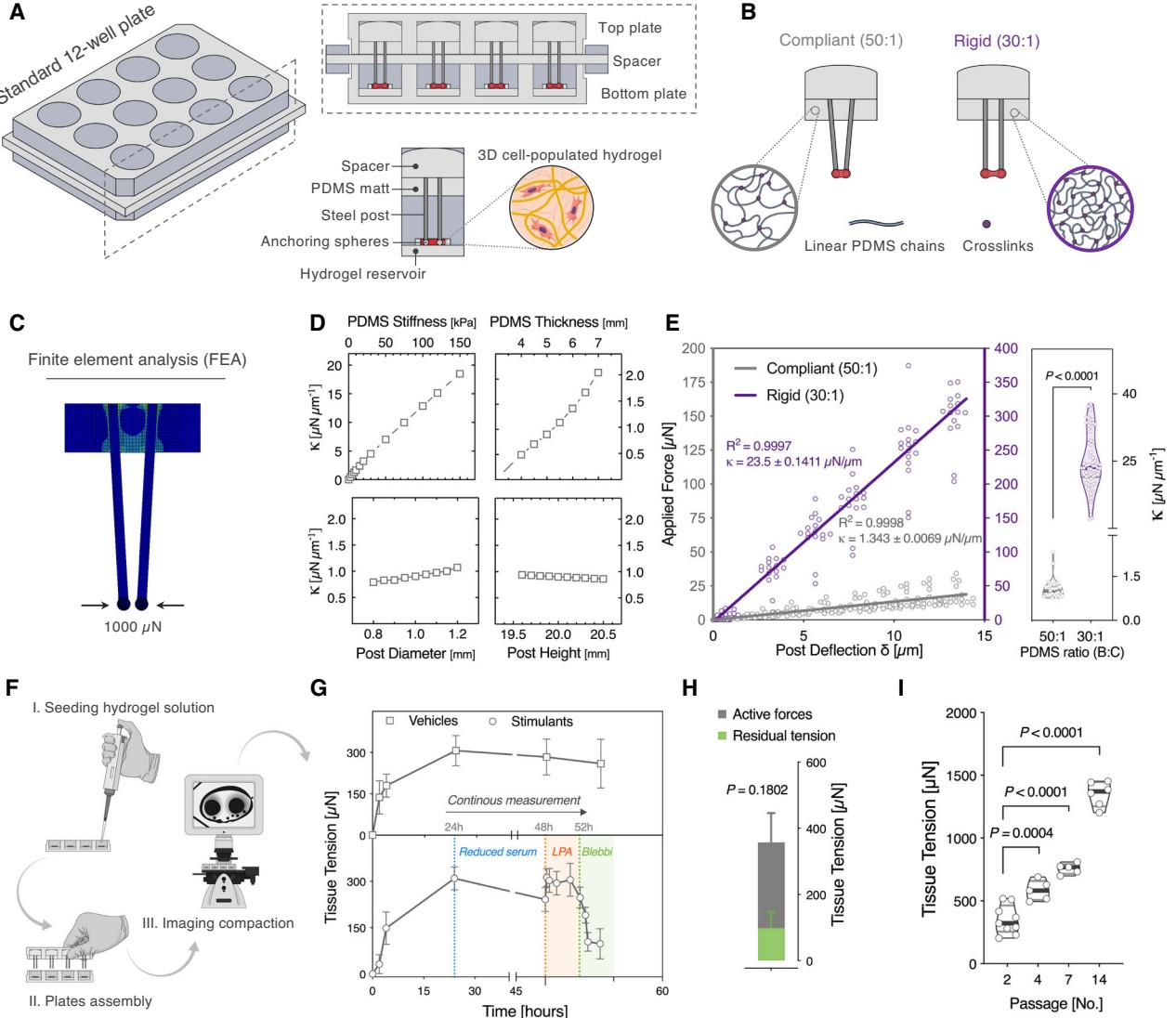

**Fig. 1 | Design and characterization of a modular, mechanically-tunable force sensing platform. A** Schematic illustration of two spaced 12-well plates interlocked on top of each other. Top: Sagittal sectional views of steel pillars arrangement in the top plate and hydrogel reservoir within the bottom plate. Bottom: Assembled modular components within each well. **B** Mechanical boundary stiffness of 3D tissue-engineered constructs is tuned by varying the mixing ratios of the PDMS siloxane base and curing agent. **C** Finite element modeling of posts deflection in response to applied horizontal force of 1000 µN. **D** Post spring constant ($k$) plotted as a function of PDMS stiffness (kPa), PDMS mat thickness (mm), post diameter (mm), and post height (mm), as analyzed by FEM modeling. **E** Mechanical characterization of anchoring pillars. Left panel: Force-displacement curves: as measured by a calibrated piezoresistive Femto-Tools™ probe (Solid lines represent linear regression lines of individual data points, $p < 0.0001$). Right panel: Violin plots of spring constants ($k$); ($n = 58$ posts (30:1), 28 posts (50:1). Two-tailed,

Mann–Whitney test. **F** Workflow of tissue formation, plate assembly and imaging of pillars deflection using a standard benchtop microscope. **G** Top: Time-course of cell-generated traction forces of rat tail-derived tendon stromal cells (Vehicle control: $n = 10$ tissues). Lower: Temporal responses of tissue constructs contractile forces to LPA or Blebbistatin stimulation ($n = 11$ tissues). Data points represent the mean ± SEM. **H** Active and residual tension at the end of the 48 h timepoint of vehicle vs stimulants conditions ($n = 6$ tissues (Active forces), $n = 11$ tissues (Residual tension). Bar plot: mean ± SEM, Two-tailed, Mann–Whitney test.
**I** Quantification of tissue tension as a function of the passage number of rat tail-derived tendon stromal cells ($n \geq 5$ tissues, One-way ANOVA with Holm–Sidak post-hoc test. Estimation plots, Cohen's $d$ and permuted $P$ values for (**E**, **G**, **I**) are in (Supplementary Fig. 2). Source data are provided in Supplementary Source Data 1 and Supplementary Source Data 2 files.

Next, we experimentally calibrated the post spring constant ($k$) when embedded in compliant (50:1) or rigid (30:1) PDMS mats. Using capacitive force sensors, we derived the post spring constant, which is the slope of the force-displacement linear relationship (Fig. 1E and Supplementary Fig. 1A, B). Rigid posts were ~20-fold stiffer than the compliant ones, with measured spring constant of 23.5 (±0.1) µN/µm compared to 1.34 (±0.01) µN/µm for compliant posts. These empirically-derived values of spring constants were used to estimate the collective forces exerted by cell-populated, tensioned tissues under different mechanical boundaries.

To generate tendon-like constructs, we embedded tendon-derived stromal cells in type I collagen hydrogel. Cells compacted the polymerized hydrogel into dense fibrous tissue around the post-anchoring spheres in each well (Fig. 1A, F). Within two hours after seeding, tissue tension evidently increased and continued to rise by 3-fold reaching a plateauing maximum of 272 (± 60) µN at 24 h ($p = 0.0014$), (Fig. 1G top, and Supplementary Fig. 1C). In that same experiment, we simultaneously tested the diffusion limits of the system by examining the rapid dynamics of tissue tension forces in response to stimulation with soluble factors.

We found that cell-mediated tissue tension was reduced after serum starvation, then increased or decreased within ~15 min of stimulation with the myosin activator, lysophosphatidic acid (LPA), or with non-muscle myosin II inhibitor, blebbistatin, respectively (Fig. 1G, H).

Next, we examined the sensitivity of the system in measuring relative differences in tissue traction forces of relatively quiescent versus activated stromal cells. We initiated the activation of stromal cells by continuously passaging rat tail-derived tendon cells on ultra-stiff tissue culture plastic which is known to induce long-term mechanical memory[46–48]. As we expected, tissue traction force significantly increased over passages by 67% at passage four (P4) and 117% at passage seven (P7) compared to passage two (P2), reaching a value of ~1353 μN at passage 14 (Fig. 1I). Taken together, we established a 3D experimental platform comprising tunable mechanical boundaries and tissue tensional states while presenting equal bulk material properties to cultured cells.

## Reconstitution of fibrogenesis in tensioned tendon-like constructs

Numerous studies have implicated aberrant transforming growth factor β (TGF-β) signaling in the initiation and progression of SSc[49–54]. Similarly, degenerative tendon diseases are increasingly viewed as fibro-inflammatory pathologies characterized by impaired expression of TGFβ superfamily and dysfunctional scarring[55–57]. TGF-β1 is considered a key profibrotic cytokine that mediates its fibrogenic effects, at least in part, by promoting the activation of "quiescent" resident stromal cells into highly-contractile myofibroblasts[58–61]. Motivated by this, we first sought to examine whether exogenous TGF-β1 stimulation induces activation of naïve tendon stromal cells into a contractile phenotype within tensioned tissue constructs. Untreated stromal cells in reduced serum (i.e., 1% FBS) were used as controls (Fig. 2A). Continuous treatment with TGF-β1 elicited visible deflections of cantilever posts over a 72-h period, with significant two-fold increase in contractile forces reaching a maximum value of $997 \pm 141.1$ μN at 48 h (Fig. 2B, C). Whereas forces continued to rise after 48 h under TGF-β1 treatment, tissue tension plateaued at 24 h at $511 \pm 61$ μN in the non-stimulated controls. However, these changes were not associated with alterations in metabolic activity, as measured by ATP assay (Fig. 2D). To correlate functional changes in cellular contractility to phenotypic activation of stromal cells, we performed immunofluorescence staining for α-smooth muscle actin (α-SMA); a cytoskeletal marker indicative of myofibroblasts. TGF-β1-treated conditions had approximately three-fold significantly higher fluorescence intensities of α-SMA than untreated controls, which is consistent with a myofibroblasts (Fig. 2E, F). Furthermore, quantitative analysis of cytoskeletal alignment and nuclear circularity of embedded cells revealed that TGF-β1 stimulation induced higher degrees of cellular alignment and nuclear elongation along the axis of mechanical tension, reflecting active remodeling processes (Fig. 2G, H). Collectively, these results underscore the utility of the mechano-culture platform in recapitulating the hallmarks of pathological fibrosis in tensioned tendon-like tissue constructs.

## Sustained mechanical tension recapitulates key aspects of pro-fibrotic activation in tendon-derived stromal cells

Progressive matrix remodeling in fibrosis results in structural changes, such as increased tissue stiffness and tension, which trigger feed-forward, self-sustaining activation loops[13,47,62]. We examined how naïve tendon-derived stromal cells respond to sustained mechanical tension when cultured under variable mechanical boundaries, and how such mechanical feedback can impact the phenotype of tendon-derived fibroblasts (Fig. 3A).

Under rigid mechanical boundaries (i.e., high matrix tension), we observed that rat tail tendon-derived cells generated tissue tensional forces that significantly increased by ~25-fold after two hours, reaching a maximum value of 6904 μN (SEM ± 1113) at 24 h (Fig. 3B, C). At 48 h,

cells embedded within hydrogels tethered to compliant boundaries (Post κ = 1.08 μN/μm) exerted ~5 nN per cell [Mean = 5.6 nN ± 2.6, $n = 6$], while cells anchored to rigid boundaries (Post κ = 24.9 μN/μm) had ~21-fold higher forces [Mean = 117 nN ± 32, $n = 7$] (Fig. 3C). Yet despite these profound differences in cellular contractility, these changes were not associated with alterations in cellular metabolic activity, as measured by total ATP assay (Fig. 3D).

Next, we investigated whether sustained mechanical tension by itself can regulate the phenotype of naïve "quiescent" tendon stromal cells. We first explored the relative impact of boundary stiffness on the activation of myofibroblasts. Rigid boundaries induced a ~50% increase in α-SMA immune staining intensities above compliant controls (Fig. 3E, F), which was consistent at the mRNA levels (Fig. 3G). Moreover, analyses of cytoskeletal alignment and nuclear orientation revealed higher degree of cellular alignment and nuclear elongation along the longitudinal axis in rigidly tethered constructs (Fig. 3H, I). Unexpectedly, we found that mRNA expression of fibrogenic matrix proteins, *Col1a1* and *Col3a1*, were significantly downregulated, whereas decorin (*Dcn*) mRNA was upregulated under rigid conditions in comparison to compliant boundaries (Fig. 3J–L).

Similarly, expression of markers indicative of tenogenic lineage commitment of stromal cells, (*Scx* and *Tnmd*) were significantly downregulated in highly-tensioned constructs (Fig. 3K). There were no differences between the rigid and compliant boundaries in the expression of ECM crosslinking genes (*Lox, Loxl2, Loxl4, Tgm2*) or the mechanosensitive transcription factor *Mrtfa* (Supplementary Fig. 4A, B).

In sum, these results suggest that elevated matrix tension can feedback to independently regulate the pheno-conversion of tendon-derived stromal cells towards myofibroblasts, while appropriate tension is beneficial for tenogenic expression.

## Systemic sclerosis-derived tendon cells are autonomously activated independent of matrix mechanics

Fibrotic involvement in SSc has been historically investigated in either easily accessible tissues e.g., skin, or in internal organs that are associated with high mortality rates, such as lungs. Tendons are often involved early during the disease course, and tendon friction rubs have high prognostic value in predicting SSc progression over time[63]. Nonetheless, knowledge about how tendons are affected at the molecular level is scarce. This is primarily attributed to the considerable ethical concerns in obtaining research biopsies from a tissue with limited healing capacity, which may further risk harming patients. Here, we give a snapshot on SSc-derived tendon tissue and stromal fibroblasts from a rare autopsy that has been donated to our laboratories after post-mortem examination (only one donor was available during the course of this study).

First, we examined the baseline gene expression in SSc tendon tissue samples using a panel of key matrisome-related genes and established markers of fibroblast activation implicated in fibro-inflammatory diseases[18,64–66]. We compared mRNA expression levels in tendons from the SSc donor with healthy hamstring tendons and age cohort-matched diseased tendinopathic tissues from biceps and Achilles tendons (Fig. 4A, Supplementary Fig. 7A, B). SSc donor samples had baseline expression levels of *COL1A1* transcripts that were significantly lower than age cohort-matched, non-SSc tendinopathic samples ($P = 0.0265$), (Fig. 4A). Expression of *COL1A1* mRNA in the SSc group was comparable to healthy tendons ($p > 0.9999$), whereas *VCAM1* transcripts were significantly higher in SSc samples compared to healthy hamstring controls (Supplementary Fig. 7A, B). In contrast, there were no statistical differences between the SSc samples and comparison groups in the expression of *COL3A1, DCN*, or *ACTA2* or fibroblast activation markers (*PDPN, TLR4, ICAM-1*), (Supplementary Fig. 7A, B).

We then explored the relative contribution of total collagen amounts and mature trivalent collagen crosslinks to tendon fibrogenesis by quantifying hydrolyzed collagen hydroxyprolines and

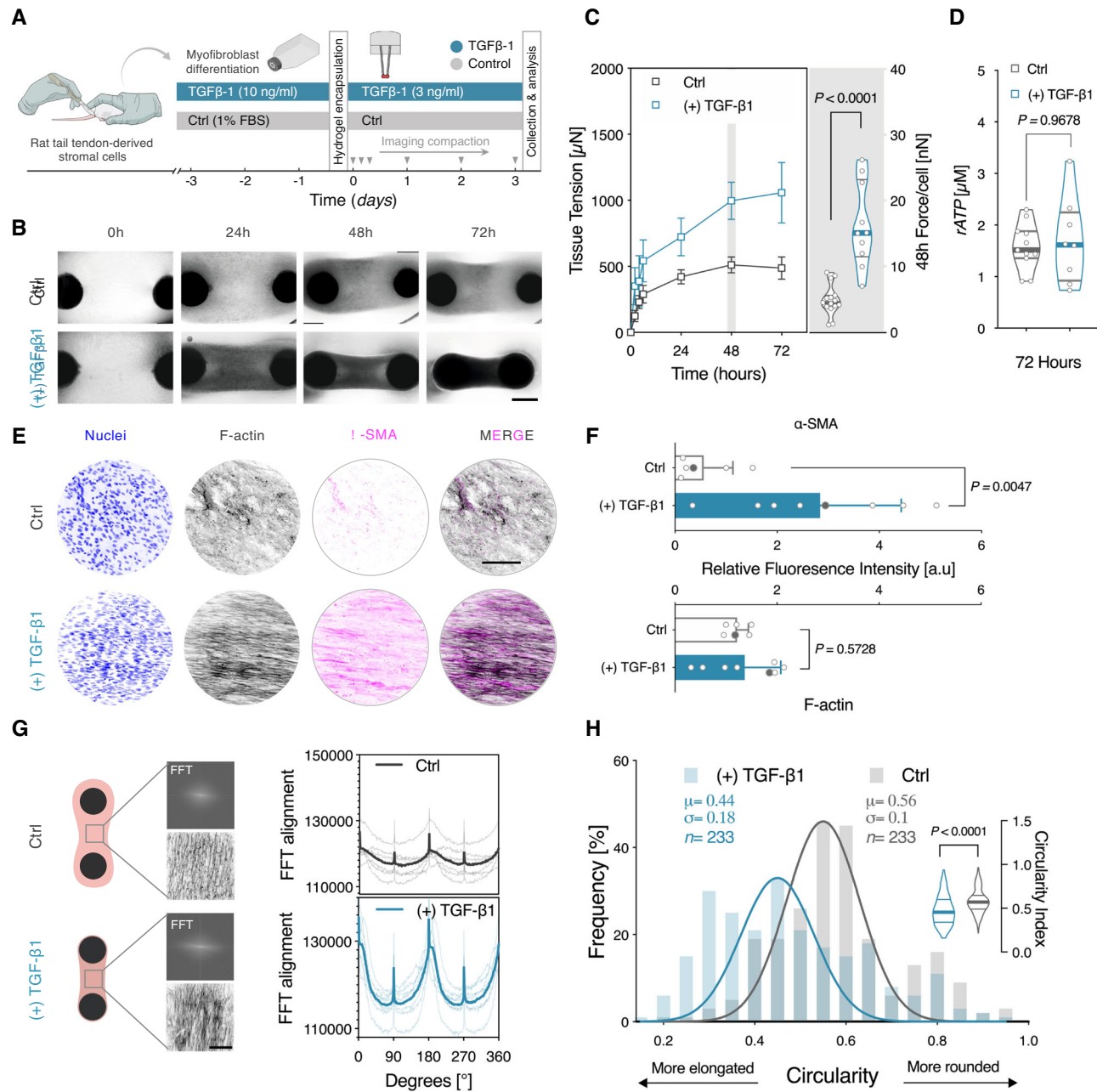

**Fig. 2 | Reconstitution of fibrogenesis in tensioned tendon-like constructs.**
**A** Schematic representation of the experimental protocol. Rat tail tendon-derived stromal cells were pre-treated with 10 ng/ml of TGF-β1 or vehicle control (Ctrl) for 72 h before encapsulation in 3D collagen hydrogels. **B** Representative images showing the time-course of tissue compaction. Scale bar = 1 mm. **C** Quantitative analysis of tissue traction forces. Left: Temporal evolution of tissue traction forces. $n \geq 18$ tissues from 3 biologically independent experiments. Data points indicate the mean ± SEM. Right: Violin plots of forces per cell, following normalization to initial seeding density. Horizontal lines indicate the median and interquartile range ($n = 15$ tissues (Ctrl), $n = 10$ tissues (TGF-β1). Two-tailed, Mann–Whitney test. **D** Analysis of cellular metabolic activity, as a measure of viability. $n = 11$ tissues (Ctrl), $n = 8$ tissues (TGF-β1) from 2 biologically independent experiments. Two-tailed, Mann–Whitney test. **E** Representative fluorescence images of cytoskeletal and pro-fibrotic activation marker, smooth muscle alpha-actin (α-SMA). Scale bar = 100 μm. **F** Quantification of fluorescent intensity levels of F-actin and (α-SMA)

in TGF-β1 stimulated and vehicle control. ($n = 6$ tissues (Ctrl), $n = 8$ tissues (TGF-β1) from 2 biologically independent experiments, Bar plots indicate the mean ± SD. Gray symbols indicate values of the representative images in (**E**). Two-tailed, Mann–Whitney test. **G** Cytoskeletal alignment in TGF-β1-treated and untreated controls. Left: Schematic of approximate positions of image sampling. Insets show representative FFT frequency patterns and corresponding F-actin staining images. Right: Analysis of overall cytoskeletal orientation from FFT graphs (Spaghetti plots: $n = 6$ samples (Ctrl), $n = 7$ samples (TGF-β1); solid lines represent the mean values). Scale bar = 100 μm. **H** Quantification of nuclear circularity of tendon-derived stromal cells. ($n = 233$ nuclei, solid lines represent Gaussian non-linear fit of frequency values). Inset violin plots represent all individual data points used for the frequency distribution analysis. Two-tailed, Mann–Whitney test. Scale bar = 100 μm. Estimation plots, Cohen d, and permuted $P$ values for (**C**, **D**, **F**, and **H**) are in (Supplementary Fig. 3). Source data are provided in Supplementary Source Data 1 and Supplementary Source Data 2 files.

mature hydroxyallysine-derived crosslinks: deoxypyridinoline (DPD) and pyridinoline (PYD), respectively, in tissue homogenates.

Results revealed no differences in total collagen content between the three tendon groups (Fig. 4B). In contrast, SSc tendons had slightly

higher amounts of PYD/DPD mature trivalent crosslinks ($p > 0.069$) and higher ($T_{onset}$) denaturation temperatures ($p = 0.0552$) compared to diseased tendinopathic tendons and healthy controls, respectively. While these differences were not statistically significant, they reflected

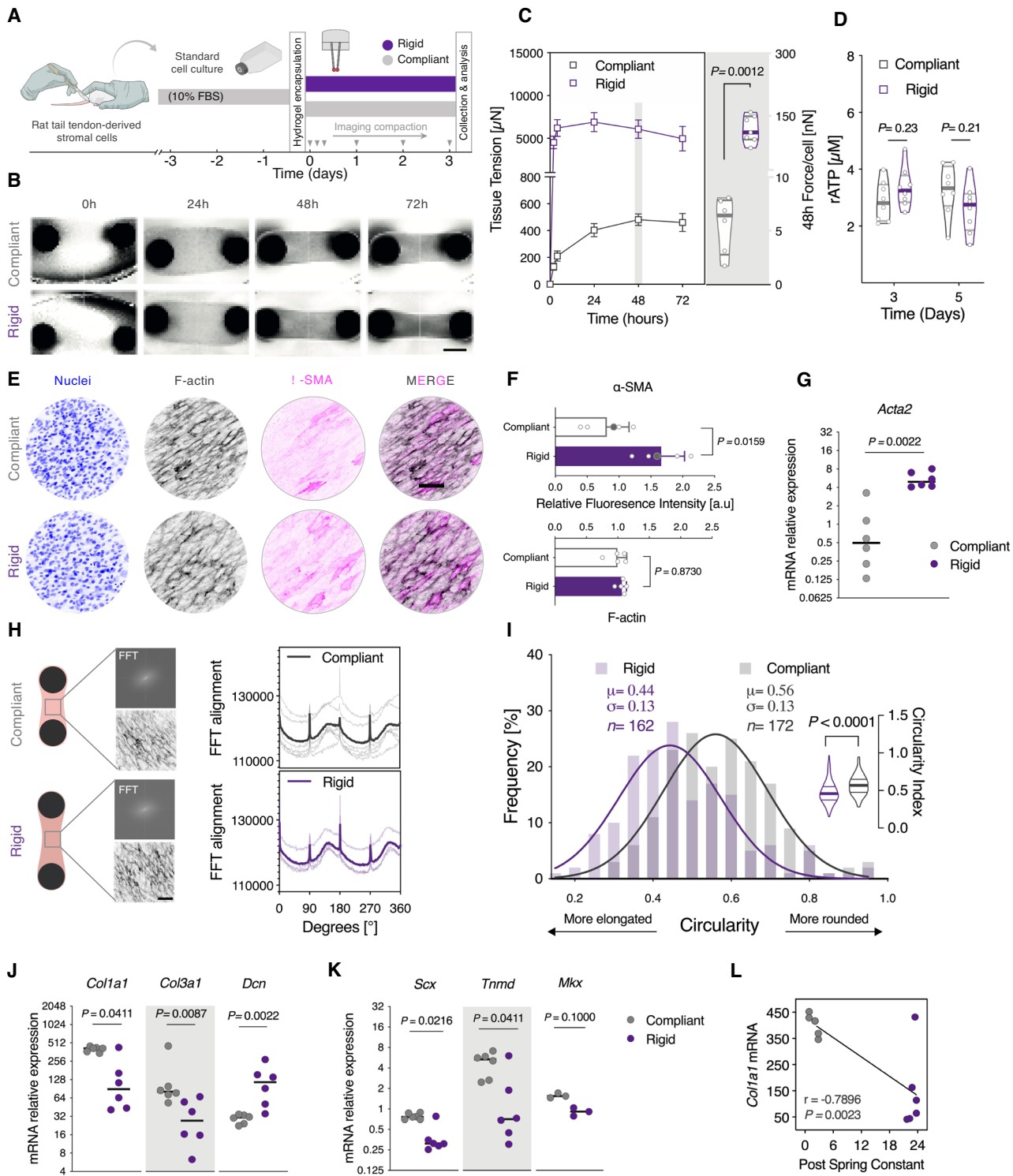

large effect sizes (Cohen's *d*) between SSc versus control tendons (Fig. 4C, Supplementary Figs. 7C, D, 8).

Having examined the baseline pro-fibrogenesis characteristics of SSc tendons, we next assessed the behavior of pathologically activated SSc tendon fibroblasts using our mechano-culture platform. We quantified tissue tensional forces of cell-populated, tendon-like constructs as a proxy measure of cellular global traction forces. For this purpose, we cultured anatomically-matched tendon fibroblasts derived from the SSc donor or healthy subjects under mechanically-compliant boundary conditions (Fig. 4D, E). Over a 48-h period of culture, we observed that SSc-derived fibroblasts showed progressive increase in tissue traction forces, akin to TGF-β1-stimulated normal

tendon fibroblasts (Figs. 4F and 2C). At 24 h time-point, SSc-derived tendon cells exerted on average 9 nN per cell [Mean = 8.9 nN ± 1.2, *n* = 15 tissues], while healthy controls had approximately a two-fold lower forces [Mean = 5 nN ± 0.8, *n* = 15 tissues]. To assess the generalizability of these findings to other fibroblast populations, we further extended this analysis to the extensively studied SSc-derived dermal fibroblasts. Indeed, SSc dermal fibroblasts progressively exerted tissue tensional forces over a 48-h period that were approximately four-times higher than healthy dermal controls (Fig. 4F). SSc-derived dermal fibroblasts applied forces of ~23 nN per cell [Mean = 22.9 nN ± 7.5, *n* = 24], while healthy control cells had ~2-fold lower forces [Mean = 12.3 nN ± 7.6, *n* = 24]. Interestingly, in that same time frame, healthy

**Fig. 3 | Sustained mechanical tension recapitulates key aspects of pro-fibrotic activation in tendon stromal cells. A** Schematic representation of the experimental protocol. Rat tail tendon-derived stromal cells were encapsulated in 3D collagen hydrogels tethered to compliant or rigid posts. **B** Representative images of tissue compaction time-course. Scale bar = 1 mm. **C** Quantification of tissue traction forces. Left: Temporal evolution of tissue traction forces. $n = 14$ tissues (Compliant), $n = 15$ tissues (Rigid) from 2 biologically independent experiments. Data points indicate the mean ± SEM. Right: Violin plots of forces per cell normalized to initial seeding density. Horizontal lines indicate the median and interquartile range. $n = 6$ tissues (Compliant), $n = 7$ tissues (Rigid). Two-tailed, Mann–Whitney test. **D** Analysis of cellular metabolic activity, as a measure of viability. ($n = 8$ tissues/group from 2 biologically independent experiments, Two-way ANOVA (boundary stiffness, time) with Holm–Sidak *post-hoc* test. **E** Representative fluorescence images of cytoskeletal and pro-fibrotic activation marker, smooth muscle alpha-actin (α-SMA). Scale bar = 100 μm. **F** Quantification of fluorescent intensity levels of F-actin and α-SMA. $n = 5$ tissues from 2 biologically independent experiments. Bar plots indicate the mean ± SD. Gray symbols indicate values for the representative images in (**E**). Two-tailed, Mann–Whitney test. **G** mRNA expression of *Acta2* gene. ($n = 6$ replicates/group from 3 biologically independent experiments, with each

data point representing a ΔCt value of 2–3 pooled tissues. Horizontal lines indicate the median. Two-tailed, Mann–Whitney test. **H** Cytoskeletal alignment of tendon-derived stromal cells tethered to compliant or rigid posts. Left: Representative FFT frequency patterns and corresponding F-actin staining images. Right: Cytoskeletal orientation from FFT graphs. Spaghetti plots: $n = 6$ samples (Compliant), $n = 4$ samples (Rigid); solid lines represent the mean. Scale bar = 100 μm. **I** Quantification of nuclear circularity. $n = 172$ nuclei (Compliant), $n = 162$ nuclei (Rigid). Solid lines represent Gaussian non-linear fit of frequency values. Inset violin plots show individual data points used for the frequency distribution analysis. Two-tailed, Mann–Whitney test. Scale bar = 100 μm. **J** mRNA expression of ECM-related genes, and **K** tendon lineage-related genes. ($n = 6$ replicates/group (except *Mkx* $n = 3$) from 3 biologically independent experiments, with each data point representing a ΔCt value of 2–3 pooled tissues. Horizontal lines indicate the median, Two-tailed, Mann–Whitney test. Expression values were normalized to *Eif4a2* and *Gapdh* reference genes. **L** Scatterplot of expression values for *Col1a1* are plotted against Post spring constants for linear regression and correlation analysis (Pearson's $r$ and Two-tailed $p$-value are indicated). Estimation plots, Cohen's $d$ and permuted $P$ values for (**C, D, F, I, J, K**) are in (Supplementary Figs. 5, 6). Source data are provided in Supplementary Source Data 1 and Supplementary Source Data 2 files.

and SSc-derived tendon cells exerted substantially less tissue traction forces compared to healthy and SSc-derived dermal fibroblasts, respectively. While the difference in the order of magnitude between SSc and healthy controls was similar in skin and tendon-derived fibroblasts, SSc dermal fibroblasts exerted higher traction forces compared to SSc tendon fibroblasts across all timepoints. This difference was more pronounced at 48 h; where SSc dermal cells applied approximately 5-fold higher forces compared to SSc-derived tendon cells. In comparison, healthy dermal fibroblasts exerted 3.5-fold higher forces relative to healthy tendon fibroblasts (Fig. 4F).

Next, we characterized the fibro-inflammatory phenotypes of tendon and dermal resident stromal fibroblasts and examined whether SSc-derived cells respond to changes in boundary rigidity. Analysis of mRNA transcripts revealed that SSc-derived tendon cells populating the tethered constructs showed a gene expression signature typical of fibro-inflammatory activation in SSc, with altered expression of *FN^(EDA)*, *IL6, IL8, ACTA2, PDPN* and *TLR4* mRNA relative to healthy, non-SSc compliant controls (Fig. 4G–I). In comparison, SSc-derived dermal fibroblasts showed similar trends with upregulation of *FN^(EDA)*, *IL6, IL8, ACTA2*, and *TLR4* transcripts. However, except for *TLR4*, these differences were not statistically significant in dermal fibroblasts, (Supplementary Fig. 8A–D). Strikingly, we found that *COL1A1* and *COL3A1* mRNA were significantly downregulated in SSc-derived tendon cells compared to anatomically-matched healthy controls, irrespective of the boundary rigidity (Fig. 4G). Moreover, *COL1A1* mRNA was significantly downregulated in healthy controls under rigid boundaries reaching transcriptional levels comparable to SSc conditions. *IL6, IL8*, and *ICAM1* transcripts were significantly increased in SSc-derived cells in response to dysregulated boundary mechanics. Consistent with our previous observations (in Fig. 3L), *COL1A1* expression in healthy tendon stromal cells showed strong dependency on boundary rigidity, which was not the case both in diseased SSc tendon cells and in dermal fibroblasts (Fig. 4J, Supplementary Fig. 8). This suggests that SSc tendon fibroblasts may have reduced baseline expression of *COL1A1*, or that altering matrisome-related transcription in response to tensional cues is an intrinsic property of healthy tendon stromal fibroblasts. Together, these results demonstrate that SSc-derived tendon cells are intrinsically activated in a cell-autonomous manner, and may possess altered tension-sensitive transcription of matrisome and stromal-immune interactions markers.

## Dysregulated mechanics characterize tendon involvement in Fosl-2^(Tg) mouse model of SSc

To extend our findings from the limited clinical samples of SSc tendons, we turned to Fosl-2^(Tg) transgenic mouse model. Unlike

bleomycin-inducible models of fibrosis, Fosl-2^(Tg) mice display spontaneous SSc-like multi-organ inflammatory and autoimmune phenotypes[67–69]. We have reported that Fosl-2^(Tg) mice develop spontaneous systemic inflammation around the age of 13–17 weeks[70]. The disease course in these mice closely approximates what is observed in human SSc cases with regards to the disease onset and progression[71]. Whereas early stages of the disease in mice are dominated by dysregulated T cell activation, late stages are characterized by vasculopathy and extensive multiorgan inflammation and fibrosis (Fig. 5A)[70].

To investigate tendon involvement in Fosl-2^(Tg) mice, we focused our analyses on two cohorts: 6-8 week-old and older than 14 week-old mice; hereafter termed early and established phenotype, respectively. We first characterized the micromechanical properties of tail tendon fascicles (the functional unit of tendons) by performing ramp-to-failure tests using a custom-built horizontal uniaxial testing device (Fig. 5B)[72]. Structural and material properties of tendon fascicles were significantly different between Fosl-2^(Tg) mice compared to wild-type (WT) littermate controls. While tendon CSA, stiffness, failure stress, failure strain, failure force and stiffness were comparable to WT during early stages of the phenotype, Fosl-2^(Tg) mice with established phenotype displayed a significant average increase in elastic moduli of 23% relative to WT controls, [$P = 0.0302$] (Fig. 5C, Supplementary Fig. 13).

Next, we sought to explore potential contributing factors to these differences in tissue mechanics. First, we profiled the mRNA expression of a focused set of key matrisome and ECM regulator genes that are known to be differentially regulated in fibrotic pathologies (Fig. 5D). Relative mRNA levels were similar in Fosl-2^(Tg) and WT early-stage cohorts. However, expression levels of *Col1a1, Col3a1, Lox*, and *Loxl2* were significantly reduced in Fosl-2^(Tg) tendons with established phenotypes compared to WT controls.

Moreover, *Col1a1* expression showed strong and significant negative correlation with tendon *E*. moduli in the established phenotype group ($r = 0.75$, $P = 0.002$) (Fig. 5E). In contrast, quantification of total collagen content with hydroxyproline assay revealed that Fosl-2^(Tg) tendons at the established phase had significantly higher collagen content relative to the early phenotype groups: WT ($P = 0.0222$) and Fosl-2^(Tg) ($P = 0.0071$) (Fig. 5F). However, there were no differences in average collagen amounts between the Fosl-2^(Tg) and WT groups within each phenotype cohort: Early ($P = 0.7007$) *vs.* Established ($P = 0.2806$) (Fig. 5F).

Taking this into consideration, we reasoned that the biomechanical differences in the stiff Fosl-2^(Tg) tendons could emerge from the formation of ECM crosslinks, rather than collagen accumulation. To test this hypothesis,

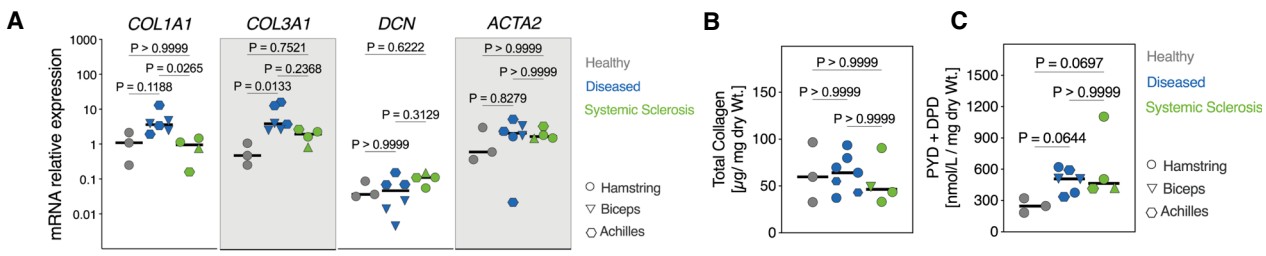

we first analyzed the molecular thermal stability of Fosl-2(Tg) tendons using differential scanning calorimetry (DSC) endotherms. Fosl-2(Tg) tendons showed significantly higher ($T_{onset}$) denaturation temperatures compared to the established, age-matched WT controls ($P < 0.0263$), and to the early Fosl-2(Tg) phenotypes ($P < 0.018$), (Fig. 5G–I). This likely reflects higher degree of thermal stabilization of collagen molecules in Fosl2(Tg) tendons.

Since the high thermal stability suggests that elevated collagen crosslinking may underlie the observed mechanical differences, we next quantified mature hydroxyallysine-derived crosslinks DPD and PYD in Fosl-2(Tg) tendons. We found that there were significantly higher amounts of PYD/DPD mature trivalent crosslinks in the established Fosl-2(Tg)

tendons compared to WT controls ($P < 0.047$), (Fig. 5J). Similarly, quantification of fluorescence intensities of a LOX-activatable collagen peptide-sensor revealed that Achilles tendons of Fosl-2(Tg) tendons with established fibrosis had considerably higher in-situ LOX enzymatic activity compared to WT controls ($p = 0.0077$), (Fig. 5K, L)[73]. These insights, taken together with the quantification of collagen amounts and DSC measurements, suggest that increased crosslinking in Fosl-2(Tg) tendons contributes to tissue-level changes in bulk mechanics (i.e., stiffening).

Overall, these findings support the premise of disrupted mechano-homeostasis in Fosl-2(Tg) tendons and point to potential

**Fig. 4 | Systemic sclerosis-derived tendon stromal cells are autonomously activated independent of matrix mechanics. A** Baseline mRNA expression of matrisome and fibro-inflammatory genes in healthy (hamstring), diseased tendinopathic tendons, and different tendons from the systemic sclerosis (SSc) donor. Horizontal lines indicate median. Expression values were normalized to *RPL13A* and *GAPDH* reference genes. **B** Quantification of hydroxyproline in snap-frozen tendons. **C** Total amounts of mature trivalent crosslinks (PYD and DPD). Panels (**A–C**): Horizontal lines indicate median. Data points represent independent donors (*n* = 3 donors (healthy), 6 donors (diseased), except for the SSc condition where each datapoint is a different tendon sourced from the SSc donor (*n* = 4 different anatomical sites). Kruskal–Wallis test with Dunn's multiple comparisons *post-hoc* test. **D** Schematic of the experimental protocol. Human tendon-derived stromal cells from healthy or SSc donor were encapsulated in collagen hydrogels tethered to compliant or rigid posts. **E** Representative images of tissue compaction time-course. Scale bar = 1 mm. **F** Quantification of tissue traction forces of tendon and dermal fibroblasts. Left: Temporal evolution of tissue traction forces. Tendon: Cells

from 3 anatomically different tendons of the SSc donor *vs.* 3 healthy young controls. *n* = 22 tissues (healthy), *n* = 24 tissues (SSc). Skin: Cells from 3 different SSc donors *vs.* 3 healthy controls. *n* = 24 tissues (healthy), *n* = 24 tissues (SSc). Data points represent the mean ± SEM. Right: Violin plots of forces per cell normalized to initial seeding density. Horizontal lines indicate the median and interquartile range (*n* = Tendon: 15 tissues, Skin: 22 tissues). Two-tailed, Mann–Whitney test. **G** mRNA expression of ECM-related genes, **H** immune-inflammatory genes, and **I** stromal activation markers. (*n* = 6 tissues/group from 3 biologically independent experiments. Data points depict ΔCt, horizontal lines indicate the median. Two-way ANOVA (boundary stiffness, disease stage) with Holm–Sidak *post-hoc* test. Expression is normalized to *RPL13A* and *GAPDH* reference genes. **J** Linear regression and correlation analysis of *COL1A1* expression values and post-spring constants (Pearson's *r* and two-tailed *p*-value). Estimation plots, Cohen's *d*, and permuted *P* values for (**C–F**) are in (Supplementary Figs. 9–12). Source data are provided in Supplementary Source Data 1 and Supplementary Source Data 2 files.

transcriptional feedback between the expression of ECM-related genes and dysregulated tendon tissue mechanics.

## Transcriptome of fibrotic Fosl-2(Tg) tendons is enriched in extracellular matrix and macrophages activation signatures

Having established the fibrotic nature of Fosl-2(Tg) tendons and that tissue stiffening is a feature of this phenotype, we next sought to explore the transcriptional landscape of tendon fibrosis in Fosl-2(Tg) mice in an unbiased fashion using bulk RNA-sequencing (Fig. 6A). Principal component analysis (PCA) demonstrated that the largest source of variation between Fosl-2(Tg) transgenic and WT genotypes was in the established group (Fig. 6B). Supporting this, pairwise differential expression analysis showed progressive increase in the number of differentially expressed genes (DEGs) between the Fosl-2(Tg) *vs.* WT comparison in the established group compared to the cohort in the early phase (Fig. 6C).

We next searched for the biological pathways that are enriched in the DEGs using *Enrichr*. We found that the DEGs in the early and established groups were functionally enriched in processes related to Epithelial–mesenchymal transition (EMT), Inflammation (e.g., Inflammatory Response, Interferons response, etc.) and ECM (e.g., ECM organization, Collagen Biosynthesis, etc.) pathways (Fig. 6D, E). To profile the biological processes (BP) underlying these pathways, we performed gene set enrichment analysis (Fig. 6F–I). Interestingly, ECM-related processes were positively enriched in the early cohort and negatively enriched in the established phenotype. Conversely, processes related to inflammation and immune responses were positively enriched in the established fibrotic cohort (Fig. 6H, I, Supplementary Fig. 15A, B). Next, we examined which genes were driving the ECM and immune signatures. We found that 12 genes of the collagen family of extracellular matrix proteins were downregulated in the Fosl-2(Tg) established fibrotic group compared to WT controls, including *Col1a1*, *Col1a2, Col2a1, Col4a5, Col6a2, Col12a1* (Supplementary Fig. 15C). Eight gene surrogate of macrophage activation (e.g., *Cd163, Retnla, Ccl9, Ccl7, Tlr4, Myd88*) were upregulated in the Fosl-2(Tg) established fibrotic tendons (Supplementary Fig. 15D).

To further assess the macrophage activation signatures, we used CIBERSORT digital cytometry tool to computationally deconvolute constituent cellular fractions in the bulk RNA-sequencing dataset. We mapped the proportion of 13 cell types that were previously established to be resident in mouse tendons (Fig. 6 J-L and Supplementary Fig. 16). Fractions of immune cells and Nerve cells 3 were significantly enriched in Fosl-2(Tg) fibrotic tendons relative to WT controls (Fig. 6L).

Given that previous investigations of Fosl-2(Tg) transgenic mice have established that macrophage infiltration and polarization characterize fibrotic phenotype in these mice, we explored the generalizability of these transcriptional processes across different connective tissues[69,74]. We performed comparative analysis of the transcriptome

of Fosl-2(Tg) fibrotic tendons and lung. We used a previously published dataset that closely demonstrated lung involvement in Fosl-2(Tg) mice (Fig. 6M). While there was moderate overlap of 341 genes between the DEGs of Fosl-2(Tg) tendon *vs.* lungs transcriptome (Fig. 6N), the two datasets significantly overlapped in Gene Ontology (GO) terms related to fibrosis, macrophage activation, inflammation processes (Fig. 6O, P). Interestingly, predicted key upstream regulatory transcription factors (TFs) included *Nfkb1* and *Sp1* TFs, which are known regulators of inflammation and fibrosis, respectively (Fig. 6Q). Similarly, we then performed a comparative analysis of the whole bulk transcriptome of the Fosl-2(Tg) tendons and capsular tissues from human patients with adhesive capsulitis[75]. We found strong and significant overlap between the two datasets (Supplementary Fig. 18A–D). Of interest was that eight out of the 20 overlapped GO terms and signatures were related to matrisome, cell-ECM interaction and inflammatory processes (Supplementary Fig. 18B). The TF-gene interactions predicted that only four transcription factors were enriched in the overlapped features between the two datasets. Interestingly, these TFs are known to be involved in tendon inflammation (*RELA*, *NFKB1*)[76], and in the cross-talk between fibroblasts and macrophages in fibrosis (*SP1*, *SP3*)[77], (Supplementary Fig. 18C).

Collectively, this comprehensive profiling of the transcriptional landscape of Fosl-2(Tg) fibrotic tendons has concretely established that repression of ECM transcription and activation of inflammation and macrophages signatures are key features of tendon involvement in Fosl-2(Tg) model of SSc.

## Dysregulated mechanics and macrophages co-culture differentially regulate stromal fibroblasts activation

Recently, we have reported that Fosl-2(Tg) mice develop multiorgan systemic inflammation phenotype, which is associated with increased secretion of pro-fibrotic cytokines IL-10, IL-13, IL-5 and IL-6[70]. In the current work, we have further observed that the transcriptome of fibrotic Fosl-2(Tg) tendons is positively enriched in "macrophage activation" and "inflammatory responses" signatures, and that rigid boundary mechanics boosted mRNA expression levels of markers surrogate of stromal-immune interactions (i.e., *IL6, IL8*, and *ICAM1*) in human SSc-derived tendon stromal cells (Fig. 4). Given these findings, we sought to investigate whether and how altered boundary mechanics impact the crosstalk between Fosl-2(Tg) tendon stromal fibroblasts and bone marrow-derived macrophages (BMDM), (Fig. 7A).

To do so, we used our mechano-culture platform to condition tendon-derived fibroblasts from mice with established fibrosis to variable boundary rigidity, either as monocultures or in direct co-culture with bone marrow-derived macrophages (BMDM) in a mix-and-match approach (Fig. 7B, C, Supplementary Fig. 19). We then assessed the mRNA expression of key tendon matrisome and ECM regulator genes in Fosl-2(Tg) or wild-type (WT) cells. In monocultures, WT tendon

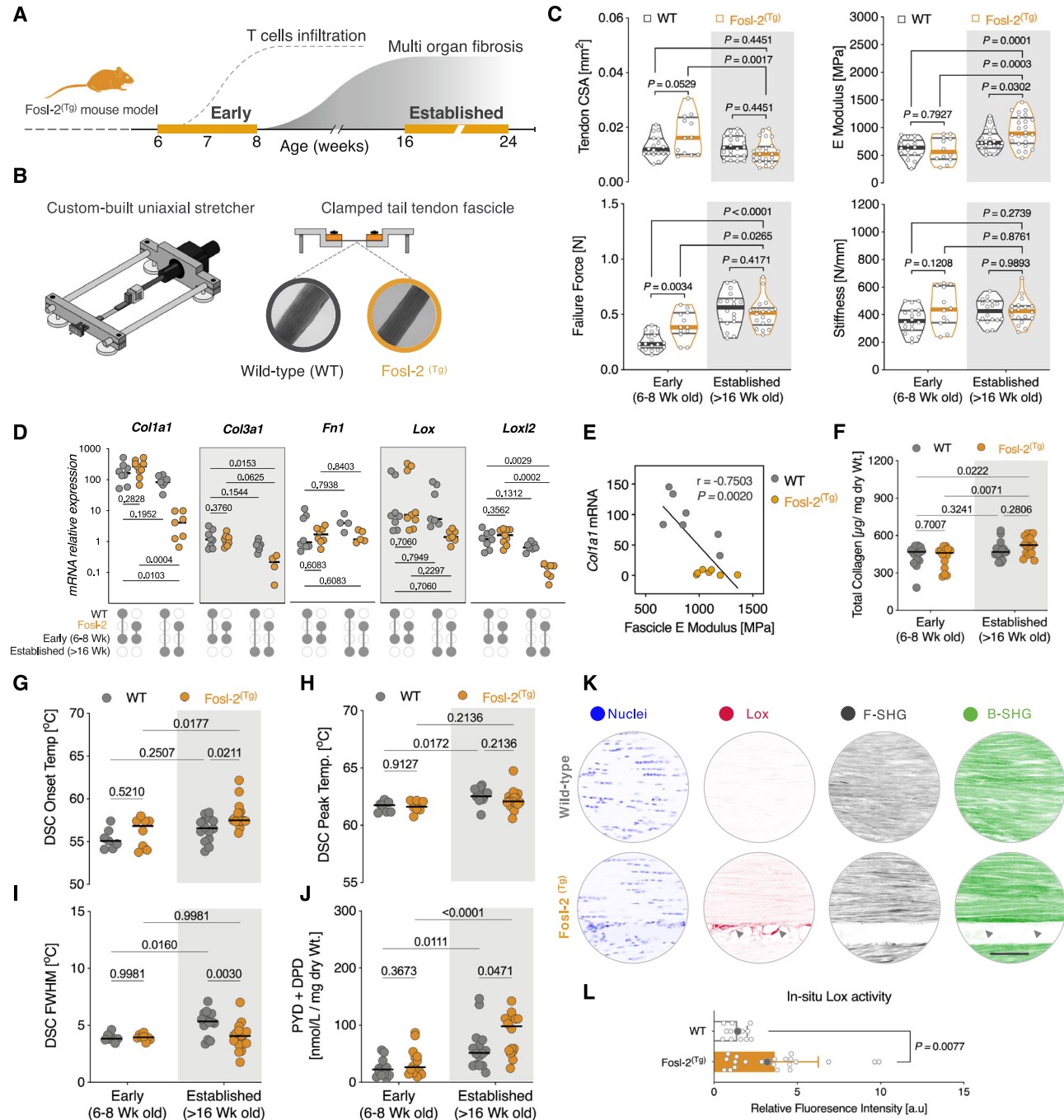

**Fig. 5 | Dysregulated mechanics characterize tendon involvement in Fosl-2(Tg) mouse model of SSc. A** Graphical timeline of the phenotype development in Fosl-2(Tg) transgenic mice. **B** Experimental setup of the uniaxial mechanical testing. **C** Biomechanical characterization of Fosl-2(Tg) tendons and their WT littermates. Violin plots depict tail tendon cross-sectional area (CSA), E modulus, failure force, and stiffness. Each data point represents independent sample (i.e., fascicle). Horizontal lines indicate the median and interquartile range ($n = 18$ (Early WT), $n = 12$ (Early Fosl-2(Tg)), $n = 23$ (Established WT), $n = 24$ (Established Fosl-2(Tg)) fascicles from ≥6 mice/group. Two-way, ANOVA (genotype, disease stage) with Holm–Šídák's *post-hoc* test. **D** Baseline mRNA expression of ECM-related genes in fresh frozen Fosl-2(Tg) tail tendons and WT controls. Data points depict ΔCt, horizontal lines indicate the median. Two-way, ANOVA (genotype, disease stage) with Holm–Šídák's *post-hoc* test. Gene expression is normalized to *Anxa5* and *Gapdh* reference genes. **E** Linear regression and correlation of *Col1a1* expression in established phenotype with fascicles E moduli (Pearson's *r* and two-tailed *p*-value). **F** Quantification of hydroxyproline in tendons. $n = 14$ mice/genotype (Early), $n = 16$

mice/genotype (Established). Two-way ANOVA with Holm–Šídák's *post-hoc* test. **G** Endothermic onset temperature (°C), **H** peak temperature, and **I** Full-width at half-maximum (FWHM) of thermally-denatured tendons measured by DSC. $n = 7$ (Early WT), $n = 8$ (Early Fosl-2(Tg)), $n = 14$ (Established WT), $n = 18$ (Established Fosl-2(Tg)) mice/group. Two-way ANOVA with Holm–Šídák's *post-hoc* test. **J** Quantification of mature trivalent crosslinks (PYD and DPD). Black horizontal lines indicate median. $n = 13$ (Early WT), $n = 14$ (Early Fosl-2(Tg)), $n = 16$ (Established WT), $n = 16$ (Established Fosl-2(Tg)) mice/group. Two-way ANOVA with Holm–Šídák's *post-hoc* test. **K** Representative images of in-situ Lox activity and second harmonic generation (SHG). Scale bar = 100 μm. **L** Quantification of fluorescent intensity values of in-situ Lox collagen peptide-sensors in Established phenotype. ($n = 13$ replicates (WT), 22 replicates (Fosl-2(Tg)). Bar plots indicate the mean ± SD. Gray symbols: representative images in (**K**). Two-tailed, Mann–Whitney test. Estimation plots, Cohen's *d* and permuted *P* values are in (Supplementary Figs. 13, 14). Source data are provided in Supplementary Source Data 1 and Supplementary Source Data 2 files.

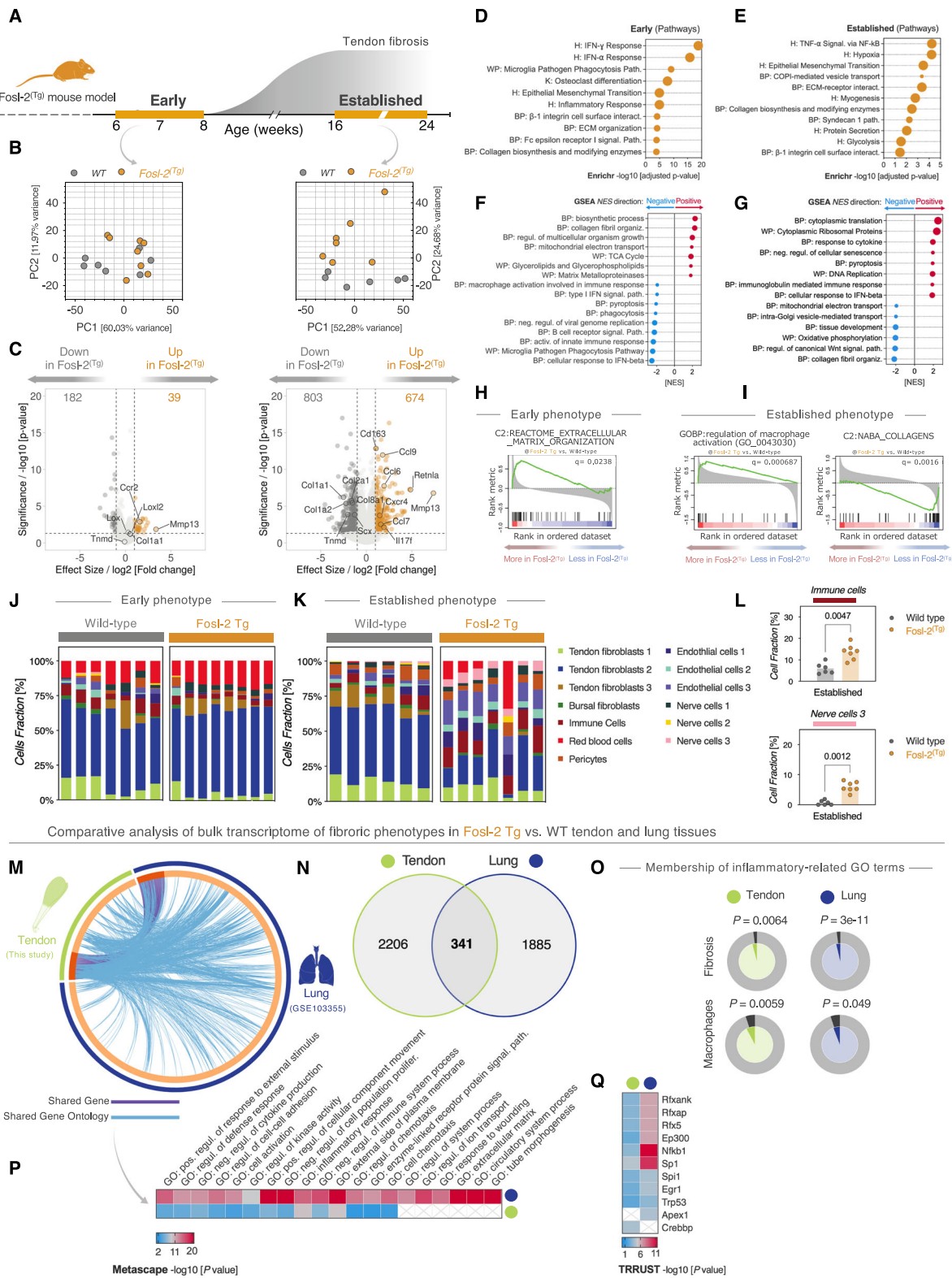

stromal cells showed modest reduction in *Col1a1* expression under rigid boundary relative to compliant controls. In contrast, total mRNA of *Col1a1* in Fosl-2(Tg) fibroblasts remained unchanged in rigid boundary compared to compliant conditions (Fig. 7D, Supplementary Fig. 20). Similarly, there was no change in *Col3a1* or *Dcn* expression in fibroblast monocultures. Interestingly, co-culture of WT fibroblasts

with BMDM increased the expression of *Col1a1*, which was more pronounced under rigid mechanical boundaries (*P* = 0.76 in compliant, *P* = 0.42 in rigid), and when WT fibroblasts were co-cultured with Fosl-2(Tg) BMDM (*P* = 0.04 in compliant, *P* = 0.03 in rigid), (Fig. 7D, Supplementary Figs. 20–22). While similar trends were observed when Fosl-2(Tg)-derived fibroblasts were co-cultured with BMDM, expression of

**Fig. 6 | Transcriptome of fibrotic Fosl-2(Tg) tendons is enriched in extracellular matrix and macrophages activation signatures. A** Graphical timeline illustrating the time window for developing spontaneous tendon fibrosis in Fosl-2(Tg) mice. **B** Principal component analysis (PCA) of feature expression counts representing the variance between wild-type (WT) and Fosl-2(Tg) at the Early and Established phenotypes. Each dot depicts an independent biological sample (i.e., mouse). **C** RNA-seq volcano plot of differential expressed genes (DEGs) of Fosl-2(Tg) vs. WT tendons of the Early (left) and Established (right) groups. Colored dots highlight the significantly expressed genes. Horizontal line corresponds to FDR ≤ 0.05 and vertical lines are at a cutoff of log$_2$[Fold change] ±1. **D, E** Enriched pathways analysis of DEGs subset using Enrichr against BP, R, WP, and K Human databases. Adjusted p-values < 0.05. **F, G** Pre-ranked Gene Set Enrichment Analysis (GSEA) of positively and negatively enriched biological processes (BP) by Normalized Enrichment Score (NES). **H, I** GSEA enrichment plots of the differentially expressed ECM and inflammation gene sets. **J, K** Distribution of inferred composition of 13 cell fractions which

was estimated from the transcriptome-wide RNA-seq using CIBERSORT algorithm. **L** Dot plots show inferred percentage of Immune Cells and Nerve Cells 3 in WT vs. Fosl-2(Tg) tendons from the Established phenotype. Each dot represents an independent biological unit (mouse). Unpaired, two-tailed Mann–Whitney test. $n = 6$ mice (WT), $n = 7$ mice Fosl-2(Tg) **(M–Q)** Comparative analysis of Fosl-2(Tg) vs. WT RNA-Seq datasets of fibrotic tendons (Established group) and lungs. **M** Circos plot of overlapped features at gene and GO terms levels. **N** Venn diagram highlights overlapped DEGs in the tendon and lung RNA-Seq datasets. **O** Enrichment of genes matching membership of fibro-inflammatory GO terms: "fibrosis", "Macrophages". Significance value indicates whether the term membership is statistically significantly enriched in the RNA-seq dataset. **P** Heatmap of top significantly enriched GO clusters, and **Q** Top 10 significantly enriched transcriptional regulators of shared targets in (**P**) using TRRUST module. Color code denotes −log10(p-value) of hypergeometric enrichment test. R Reactome, WP WikiPathways, K KEGG, H Hallmark, GO-BP Gene Ontology - Biological Processes.

Cola1 was highly variable and statistically insignificant irrespective of the mechanical boundary conditions.

mRNA expression of Col3a1 and Lox followed similar trends to Col1a1, with one difference that Col3a1 expression was only statistically significant when WT fibroblasts were in co-culture with Fosl-2(Tg) BMDM under rigid boundaries ($P = 0.03$) (Fig. 7D, Supplementary Figs. 20–22). Similarly, co-culture with BMDM induced significant shifts in the expression of cross-linking enzyme Loxl2 in the conditions where Fosl-2(Tg) fibroblasts were in co-culture with WT or Fosl-2(Tg) BMDM under rigid boundaries ($P = 0.05$ and 0.03, respectively) compared to mono-cultures (Fig. 7E, Supplementary Figs. 23–26).

Next, we quantified how boundary rigidity and co-cultures with BMDM influence tissue tensional forces of Fosl-2(Tg) and WT tendon stromal cells in vitro (Fig. 7F). Fosl-2(Tg)-derived tendon cells exerted significantly higher traction forces as early as four hours ($P = 0.0006$), which reached a maximum value of 640.2 μN (±104.83) at 24 h ($p < 0.0001$). In comparison, basal tensional forces generated by WT-derived fibroblasts were approximately four-fold lower than Fosl-2(Tg) cells, which plateaued after six hours of monoculture at 169.3 μN (±29.5) (Fig. 7F, left). Direct cocultures with BMDM significantly enhanced traction forces generated by WT tendon fibroblasts, with WT-derived BMDM boosting traction forces by four folds, whereas co-culture with Fosl-2(Tg) BMDM significantly increased WT fibroblasts contractility to levels comparable to activated Fosl-2(Tg) fibroblasts at 24 h timepoint (Fig. 7F, G, Supplementary Fig. 27).

Collectively, these findings suggest that interaction between tendon stromal fibroblasts and macrophages may underpin the induction and progression of fibrogenesis and crosslinking-mediated stiffening in Fosl-2(Tg) mice. It underscores the powerful utility of our mechano-culture platform in unwinding the complex interplay between the mechanical boundary rigidity, evolved ECM tension and the stromal-macrophage interactions in models of tendon fibro-inflammatory diseases.

## Discussion

We have engineered a modular mechano-culture platform that enables one to easily vary static mechanical tension of tendon-like hydrogel constructs independently of bulk material properties (e.g., stiffness, crosslinking density, or composition). We have used this platform to reveal a complex interplay between sustained mechanical tension, ECM remodeling and activation of tendon-derived stromal cells in models of fibro-inflammatory pathologies.

Fibrotic progression in SSc is accompanied by substantial changes in tissue mechanics, particularly heightened ECM tension and rigidity. Until now, a myriad of 3D in vitro tissue mimetics has been developed to model states of static tension in connective tissues. However, the artisanal craftsmanship required to engineer and manufacture these systems has limited its widespread adoption by biomedical and translational research labs. Our mechano-culture platform described

here addresses this need. We propose a straightforward platform that can easily be replicated; allowing to study the roles that matrix tension play in pathological fibrosis while simultaneously providing a highly relevant online (non-invasive) readout of cellular forces in situ.

In the present study, we show that elevated matrix tension in 3D tissue constructs mediates pheno-conversion of naïve tendon-derived stromal fibroblasts towards activated myofibroblasts. By anchoring collagen hydrogels to cantilever posts with varying degrees of boundary rigidity, we sought to create ECM tissue mimetics with states of high or low tension, independently of altering bulk stiffness of the provided collagen matrix or application of exogenous TGF-β1. In general, tissue resident stromal cells sense different degrees of matrix tension through a sophisticated mechanotransduction apparatus that connects the extracellular matrix to the contractile cytoskeleton and other mechanically sensitive subcellular structures. In highly tensioned environments, this process culminate in incorporation of α-SMA into contractile actin stress fibers; a hallmark of myofibroblasts activation[78].

However, most of these observations are based on experiments conducted on 2D planar substrates where mechanotransduction dynamics are substantially different from 3D counterparts. Nonetheless, our findings are consistent with the role of matrix tension in myofibroblast activation[78,79].

How elevated matrix tension contributes to myofibroblast activation and fibrosis progression is an area of intense research. Elegant work from Wipff et al. and Wu and colleagues have clearly demonstrated that excessive mechanical tension sets resident stromal cells on a fibrotic trajectory by mediating the activation of latent TGF-β signaling loops[80,81]. Because myofibroblast activation is associated with a phenotypic switch and enhanced production of extracellular matrix, we focused our analysis on the transcription of key matrisome-related molecules and markers of tendon lineage specification. Strikingly, dysregulated tissue stiffening was associated with downregulation of collagens mRNA transcription in 3D tensioned cultures in vitro. This finding is in contrast with the current consensus using 2D planar substrates where progressive matrix stiffening strongly promotes fibroblasts matrix synthesis[46–48,80,82–85].

In this study, we extended the utility of this mechano-culture platform to investigate the impact of mechanical stress on the pathological activation of tendon fibroblasts in SSc. We provide limited evidence from proof-of-principle experiments using tendons from multiple anatomical locations of a single donor with confirmed diagnosis of SSc. We found that tendon stromal cells from healthy human controls exhibited comparable transcriptional responses to rodent-derived, rigidly-anchored quiescent fibroblasts in vitro and to the stiff Fosl-2(Tg) tendons in vivo. Specifically, healthy tendon-derived cells showed significant downregulation of collagens mRNA (COL1A1 and COL3A1) in response to the high mechanical tension in rigid mechanical boundaries. In contrast, SSc-derived tendon cells had significantly

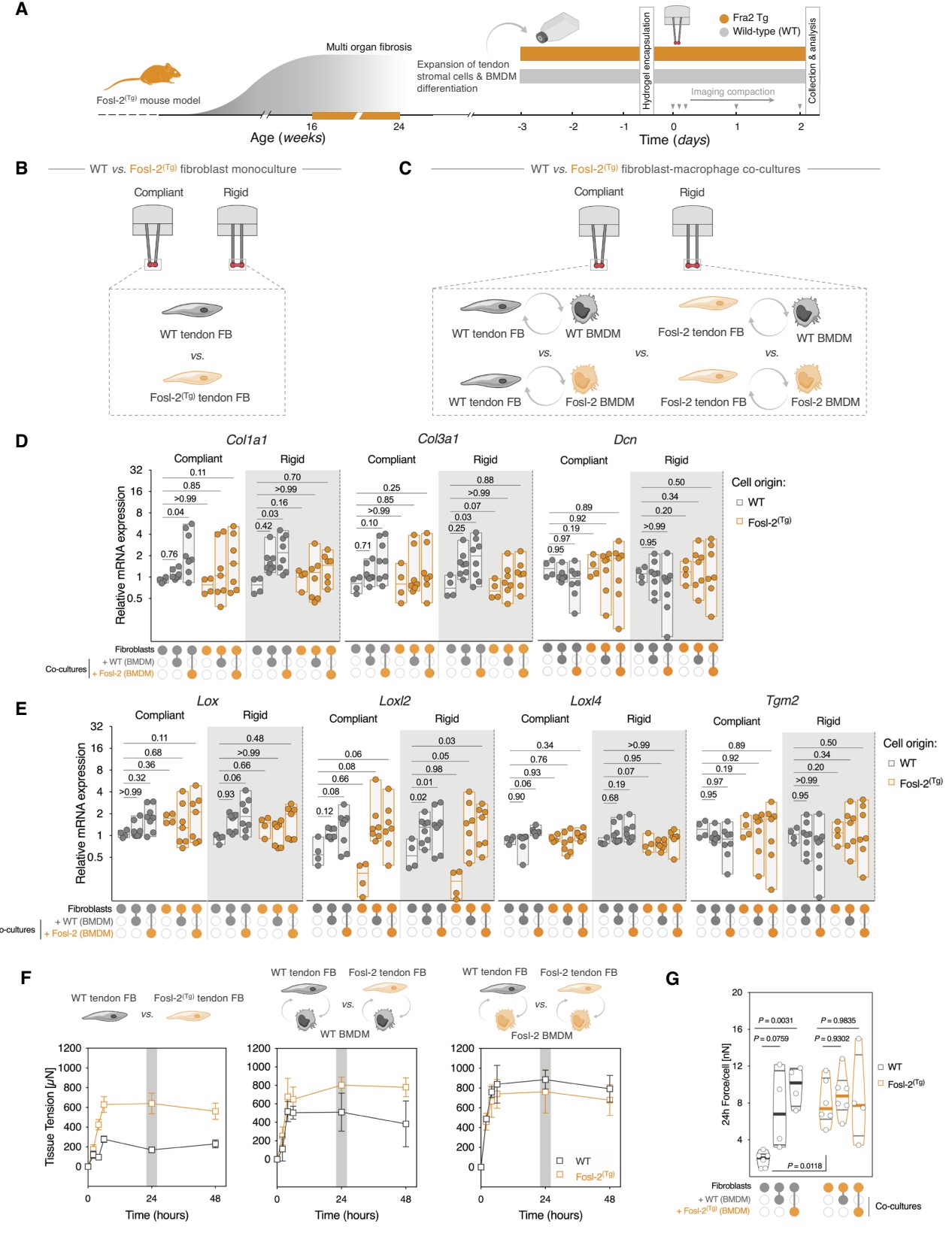

lower expression of *COL1A1* and *COL3A1* transcripts compared to healthy controls, and irrespective of the mechanical condition.

Our findings are in agreement with published reports showing that dense, mature fibrotic niches were associated with decreased transcription of *Col1a1* and *Col1a2*[86,87]. This decreased transcription was further reported in a multi-omics single cell profiling of the aging lung, which attributed the dysregulated matrix remodeling in aged lungs to the increase in transcriptional noise and aberrant epigenetic control[88]. Although the aforementioned studies attributed these age-dependent transcriptional repression (at least in part) to cell-mediated processes, the contribution of ECM stiffening in aging cannot be ruled out[89]. Based on our findings, it is tempting to speculate that elevated

**Fig. 7 | Macrophages enhance stromal fibroblasts activation and expression of ECM remodeling genes. A** Schematic illustration for experimental design and timeline. **B** Experimental design for monocultures and **C** "mix-and-match" direct cocultures. **D** mRNA expression of ECM-related genes, and **E** ECM crosslinking enzymes in monoculture and direct cocultures of Fosl-2[(Tg)] and WT cells tethered to different mechanical boundary rigidities. (Mono-cultures: $n = 4$ replicates/group, with cells derived from 4 different mice. Co-cultures: $n = 8$ replicates/group, with tendon cells derived from 4 different mice. BMDM cells were from two different pools of cells sourced from $n = 2$ different mice and pooled together. Each data point represents fold-change value. Floating bars depict min. and max. values with horizontal line indicating the median. Two-way ANOVA (Genotype, co-culture mix) with Dunnett's *post-hoc* test. **F** Time-course quantification of tissue traction forces

of stromal fibroblasts monocultures (left: $n = 12$ tissues (WT), $n = 16$ tissues (Fosl-2[(Tg)]); stromal fibroblasts co-cultured with WT-derived BMDM (middle: $n = 5$ tissues (WT), $n = 6$ tissues (Fosl-2[(Tg)]); stromal fibroblasts co-cultured with Fosl-2[(Tg)]-derived BMDM (right: $n = 4$ tissues (WT), $n = 4$ tissues (Fosl-2[(Tg)]). Data points represent means ± SEM. **G** Violin plots of traction forces per cell, following normalization to initial seeding density of stromal fibroblasts. Each data point represents a biological replicate. Horizontal lines indicate median values and interquartile ranges (Two-way ANOVA (Genotype, co-culture mix) with Tukey's post-hoc test). Estimation plots, Cohen's *d* and permuted *P* values for (**C**) are in (Supplementary Figs. 20–27). Source data are provided in Supplementary Source Data 1 and Supplementary Source Data 2 files.

matrix stiffness is a key driver of dysregulated matrix remodeling in tendons, as was recently shown to be the case in brain tissues[90].

One plausible mechanism by which matrix stiffening alters transcription of collagen and crosslinking-related genes is by increasing the physical confinement of tissue resident cells. Several lines of evidence from tendon and other biological systems corroborate this hypothesis. Work by Zhou, Franklin and colleagues using mouse embryonic fibroblasts revealed that proliferation of fibroblasts and expression of matrisome-associated genes (e.g., *Serpine1* and *Col2a1*) is significantly limited by spatial confinement[91]. The authors of this work used the term "physical space availability"; a synonymous term for spatial confinement. They attributed this space availability/confinement-dependent rewiring of gene expression programs to the differential activation of Hippo-YAP and TGF-β signaling pathways. Spatially confined fibroblasts (i.e., high-density culture) displayed higher fluorescent intensity of cytoplasmic YAP1 and downregulation of *Serpine1* and *Col2a1* mRNA transcription, whereas less confined cells (i.e., low-density culture) exhibited increased fluorescent intensity of nuclear YAP1 and transcription of ECM genes[91].

Additionally, Work by Grinstein and colleagues demonstrated that the global transcription of *Col1a1*, *Col3a1*, and *Mki67* (proliferation marker) is significantly repressed when tendons transition from cell proliferative-driven embryonic growth to ECM expansion homeostasis postnatally[86]. This phase of tendon development is accompanied by significant mechanical stress with increased accumulation of collagen matrix and tissue tensioning resulting in spatial confinement of tendon cells.

As tendons increase in length and ECM density, the cellular proliferation rate and transcription of ECM genes significantly decreased between postnatal day P0 and P35[86]. Studies by Podolsky et al. showed that although total collagen content was significantly higher in aged and fibrotic lungs, *Col1a1* and *Col1a2* transcripts were universally suppressed in comparison to young non-fibrotic controls[87]. Similarly, encapsulating tendon progenitor cells within stiff, highly crosslinked alginate niches spatially confined tendon cells in 3D and restricted the degree of cellular spreading[92]. While these studies did not mechanistically explain the potential mechanical factors driving these observations, we speculate that the downregulation of cellular proliferation and *Col1a1*, *Col1a2*, and *Col3a1* transcripts is driven, at least in part, by the spatial confinement of tendon cells during the ECM expansion and/or stiffening.

The precise molecular mechanism(s) by which tension-mediated mechanical confinement regulates global transcription rates still need to be worked out. However, one avenue that warrants further investigations is the potential crosstalk between mechanical confinement cues and the RNA Polymerase II (RNAPII) enzyme. Active RNAPII is a hallmark of ongoing mRNA transcriptional elongation. It has been previously shown that stiffness-mediated mechanical stress in models of skin aging correlates with notable reduction in cellular proliferation as well as in the phosphorylation RNAPII-S2P[93]. ChIP–seq experiments in these studies have confirmed a reduced genome occupancy of RNAPII in aged skin compared to young controls. It is indeed

challenging to decouple the effect of aging processes from the aging-mediated tissue stiffening and increased confinement in in vivo mouse models of aging. However, supporting evidence from alternative mouse models of tissue stiffening (i.e., age-matched Collagen XIV[-/-] and Tnx[-/-]) and from 3D organoid confinement have demonstrated the essential role of mechanical stress in reducing RNAPII occupancy and eventually in attenuating global transcription[93,94].

We acknowledge that we prioritized comparing the cellular traction forces and gene expression of SSc-derived cells to healthy cells originated from young donors, limiting our ability to exclude the potential confounding effect of ageing. However, in strong support of the valuable but limited human data we present, we found that tendons of Fosl-2[(Tg)] mice exhibit dysregulated mechanical homeostasis, with significantly higher failure forces and elastic moduli.

We showed that Fosl-2[(Tg)] overexpressing transgenic mice develop T cell-mediated spontaneous systemic inflammation with many features reminiscent of SSc[70]. Moreover, sera from Fosl-2[(Tg)] mice had elevated levels of profibrotic Th2 cytokines (IL-5, IL-6 and IL-10) as well as other inflammatory cytokine/chemokine mediators (IL-1b, TNF-a and CCL2). Biochemical and structural analyses revealed that tendon ECM stiffening in Fosl-2[(Tg)] mice likely results from matrix crosslinking rather than excessive ECM production. These findings both mirror and contrast with previous reports examining the role of Fra-2 in tissue fibrosis (Fra-2 and Fosl-2 are synonyms aliases for the same gene). In agreement with our findings, Ucero et al. showed that Fosl-2[Tg] mice develop spontaneous lung fibrosis, which was associated with increased expression of matrix crosslinking enzyme *Loxl2* (Fosl-2[Tg] mice in Ucero et al. were derived from different founder animals than the ones used in this work)[69]. Similarly, Georges and colleagues demonstrated that tissue stiffening preceded collagen deposition in an inducible model of liver fibrosis through a Lox-mediated crosslinking activity that ultimately results in myofibroblast activation[25]. Although the spontaneous fibrosis in Fra-2 lungs was also associated with enhanced expression of ECM collagen, this was not the case in our tendon characterization suggesting tissue-specific differences in the mechanism of fibrosis in Fosl-2/Fra-2[(Tg)] mice.

Whether tendon stiffening in Fosl-2[(Tg)] mice is mediated through intrinsic cellular activation mechanisms, profibrotic Th2 paracrine signaling, or both remains elusive and is ground for future work in our laboratories. However, evidence from our in vitro experiments suggests that Fosl-2[(Tg)] tendon-derived fibroblasts and macrophages cultured under tension are highly contractile and express higher levels of *Loxl2* under rigid mechanical boundaries. This suggests that tendon stiffening in Fosl-2[(Tg)] mice is potentially mediated, at least in part, by a complex interplay between tissue mechanics and fibroblast-macrophages crosstalk. These findings echo observations from other fibro-inflammatory conditions such as adhesive capsulitis and arthritis[95]. Akbar et al. mapped the immune landscape in adhesive capsulitis, and discovered that the immune milieu in diseased shoulder capsules is dominated by T cells with a clear shift from macrophages to T cells-enriched populations compared to the resting control tissues[95]. Furthermore, work by Ng and colleagues reported similar observations

where they further revealed that T cells and macrophages populate distinct microanatomical niches in the capsular tissues of adhesive capsulitis[96].

Adhesive capsulitis tissues were enriched in T cells in the sublining layer and MERTK⁺LYVE1⁺MRC1⁺ macrophages subpopulation in the lining layer. While the authors attributed these differences to the potential activation of developmental cellular programs, it remains to be seen if the underlying biophysical cues of the capsular matrix contribute to this differential homing of macrophages and T cells in adhesive capsulitis[96].

We conclude that tension-regulated positive feedback loops are likely to be a central feature of tendon fibrosis in SSc. Our study demonstrates how mechanovariant tissue-engineered models provide a powerful platform for unwinding the mechanisms of cell-cell and cell-matrix crosstalk. We suggest that these experimental models can and should play an important role in future research, including the identification of therapeutic targets to potentially mitigate the onset and progression of fibrosis.

## Methods

### Device assembly

The device was assembled by combining two standard 12-multiwell plates interlocked on top of each other. Plates were separated by a Poly(methyl methacrylate) spacer. The top plate contained two arrays of steel pin pairs (A20007120, inox-schrauben.de) embedded within a thick mat of PDMS. Each well in the top plate's upper (A1–A4 wells) and lower (C1–C4 wells) arrays contains one pair of pins (i.e., 8 pairs in total per plate), while the central wells were left empty. The PDMS layer was cast on top of a polycarbonate disc. This allows for controlling the PDMS thickness as well as the depth of pins into the bottom plates. Pins were $(1 \pm 0.006)$ mm in diameter, $(20 \pm 0.5)$ mm in length and spaced ~4 mm apart from each other. To provide a better gel anchoring point, a ceramic bead (116913050CF: Lysing Matrix D, MP Biomedical) with an average diameter of 1.4 mm was glued on top of each post using medical grade, low-viscosity ethyl-based instant adhesive (LOCTITE® 4011 Prism Medical CA: Ellsworth Adhesives, UK). The corresponding bottom plate encompasses a PDMS well which serves as reservoir for casting cell-laden hydrogels. The 5 mm × 8 mm PDMS reservoir wells were punched out of ~2 mm thick layer using a metal puncher and glued to a ~10 mm mat of 10:1 PDMS.

### Calibration of steel pillars spring constants

Post spring constants were measured experimentally using a capacitive piezoresistive MEMS force sensor mounted on an XYZ micromanipulating arm in a horizontal configuration (FT-RS1002 Microrobotic System: FemtoTools AG, Switzerland), as described elsewhere[97,98]. The posts were individually calibrated by bringing the tip of the force sensor into contact with approximately the center of the glued spherical beads. Displacement force is measured while simultaneously pushing against the sphere to deflect the post. For each measurement, the post was displaced until the sensor force reached a value of 400 μN or 4000 μN for the compliant or rigid configurations, respectively. Hooke's law is used to calculate the spring constant (κ):

$$k = \frac{\Delta F}{\Delta \delta} \qquad (1)$$

### Quantification of tissue traction forces

To quantify tissue traction forces, cell-populated gels were cultured under tension for the indicated time periods outlined in figures legends. Timepoint "zero" was defined as the moment the growth media is added to the polymerized collagen gels. Briefly, tissue tension was quantified by monitoring the post deflection every 2 h for the first 4–6 h and then every 24 h for up to 3 days. Images of the beads capping

the posts were acquired with an inverted, brightfield EVOS™ XL Core Imaging System (AMEX1000, Thermo Fisher Scientific) using a 2× or 4× objectives. These images were processed with a custom-written Matlab script (Matlab® R2018b, MathWorks, Inc.) to segment the two beads and calculate the distance between the centroids (in pixels). The deflection of the posts was determined for each time point and the distance between the two beads was rescaled to the initial inter-post distance (in μm) that was experimentally measured with the Femto-Tools during the calibration spring constants. Finally, tissue traction forces were calculated with Hooke's law formula using the measured deflections ($\Delta \delta$) and the average of spring constant of each post (κ).

### Finite element simulation of microtissue contractility

Posts and the embedding PDMS mat model were implemented in Ansys Workbench (V. 16.2., ANSYS, Inc). The posts were modeled as stainless-steel material using the default settings of the software. Spherical beads were simulated as a linear elastic material with a density of 2400 kg/m³, a Young's modulus of 185 GPa, and the PDMS was modeled as a linear elastic material with a fixed Poisson's ratio of 0.499 and a Young's modulus of 5 kPa. A hexahedral mesh was used for the post and PDMS substrate, while tetrahedral mesh was fitted for the capping bead. To save computational power, only one post was modeled, and a symmetry region was added to create the second post. For the boundary conditions, fixed constrains on all sides were chosen. As loading condition, a force acting on the surface of the beads in direction of the plane of symmetry was set. The magnitude of the loading force was set to 1000 μN. This force represents the force generated by the contracting cells in the collagen gel. The simulations were also performed using a Neo-Hookean model for the material.

### Fosl-2$^{(Tg)}$ mice experiments

Fosl-2$^{(Tg)}$ mice were generated by Sanofi-Genzyme as described elsewhere[70]. Briefly, a vector containing the murine Fosl2 gene (Exons 1–4, corresponding introns, and truncated UTRs) was randomly inserted into the genome under the control of the MHCI promotor H2Kb. The eGFP sequence was inserted in frame after the Fosl2 transgene as a marker with a self-cleaving peptide T2A (Thosea asigna virus 2A) inserted between the two coding sequences[70]. Mice were housed under specific pathogen-free conditions at the University of Zurich. Nontransgenic mice from the same breeding were used as controls.

### Biomechanical testing of tendons

For biomechanical testing of tendons, mice were anesthetized and sacrificed according to the guidelines of the Swiss Animal Welfare Ordinance (TSchV) University of Zurich. Samples were then stored in a freezer until further analysis. On the day of mechanical testing, fascicles were extracted by holding the tail from its posterior end with surgical clamps, and gently pulling off the skin until bundles of fascicles were exposed. Each fascicle was examined under microscope (Motic AE2000, 20× magnification) for visible signs of damage, and for measuring the external diameter. Specimens with frayed ends, visible kinks or diameters less than 90 μm were discarded. Micromechanical tensile testing was performed using a custom-built horizontal uniaxial test device to generate load-displacement and load-to-failure curves (10 N load cell, Lorenz Messtechnik GmbH, Germany). Briefly, two-centimeter fascicle specimens were carefully mounted and kept hydrated in PBS, as previously described (Snedeker et al.). Each fascicle underwent the following protocol: pre-loading to 0.015 N (L0: 0% strain), 5 cycles of pre-conditioning to 1% L0, an additional 1% strain cycle to calculate the tangential elastic modulus, and then ramped to failure to 20% strain under a predetermined displacement rate of 1 mm/s. The load-displacement data were processed using a custom-written Matlab script (Matlab® R2018b, v. 9.5.0.944444, MathWorks, Inc.). Tangent elastic moduli were calculated from the linear region of

stress-strain curves (0.5–1%). Nominal stress was estimated based on the initial cross-sectional area. Cross-sectional area was calculated from microscopic images assuming fascicles have perfect cylindrical shape.

## Cell culture

For all experiments, cells were maintained in Dulbecco's Modified Eagle's Medium with L-glutamine, sodium pyruvate, and sodium bicarbonate (DMEM high glucose - D6429, Sigma-Aldrich), supplemented with 10% FBS, 1% Penicillin-Streptomycin, 1% MEM Non-essential Amino Acid Solution (M7145, SAFC Sigma-Aldrich) and 200 μM L-ascorbic acid phosphate magnesium salt (013-19641, FUJI-FILM Wako Chemicals).

## Isolation of tail tendon-derived stromal cells

Tail tendon-derived stromal fibroblasts were isolated from sexually mature 12–14 week-old Wistar rats, or Fosl-2[(Tg)] transgenic mice and their wild-type controls. Rat tail tenocytes were chosen as a model in proof-of-concept experiments because they originate from the same population of tendon somatic progenitor cells as load-bearing tendons[99], and are convenient for obtaining large number of cells at low passages. After euthanasia, tail tendon fascicles were extracted under sterile conditions as described above in the biomechanical testing section[100]. Tissue was washed once in PBS, minced into small pieces, and digested overnight in 0.2% (wt/v) Collagenase D (11088866001, Roche) in DMEM-F12 medium supplemented with 1% (v/v) Penicillin-Streptomycin. The same isolation protocol was followed for isolating tendon-derived stromal fibroblasts from Fosl-2[(Tg)] mice.

## Isolation and differentiation of bone marrow-derived macrophages

Bone marrow-derived macrophages were prepared as detailed elsewhere[101]. In brief, bone marrow cells were harvested by flushing femurs and tibias of sex-matched, littermate mice. Bone-marrow-derived macrophages (BMDMs) were generated in DMEM-high glucose media supplemented with 10% (v/v) heat-inactivated FBS, 1% (v/v) Penicillin-Streptomycin, and 50 ng/ml of recombinant murine M-CSF (315-02, PeproTech). Half of culture medium was replenished every 3–4 days. At day 7 post-induction, macrophages were collected and used in the co-culture experiments with tendon-derived stromal fibroblasts.

## Isolation of tendon-derived stromal cells from healthy and SSc tendons

Hamstring tendons (Semitendinosus and Gracilis) were collected from otherwise healthy donors undergoing surgical autograft repair of their anterior cruciate ligaments[102]. SSc tendons were collected from a single donor postmortem at the time of autopsy. Hamstring, supraspinatus, infraspinatus, subscapularis, patellar and Achilles tendons were collected by autopsy technicians under the supervision of a senior pathologist. The patient fulfilled the American College of Rheumatology (ACR)/European League Against Rheumatism (EULAR) criteria for SSc[103]. The female patient was in the age range of 75–80 years old, and had a disease duration of from first non-Raynaud-symptom of 21 years and was suffering from diffuse cutaneous SSc with a modified Rodnan skin score as a measure of skin fibrosis of 14/51 two days before her death. The patient was positive for high-titer anti-nuclear Abs on immunofluorescence with an AC-3 pattern. Consistently, she was positive for anti-centromere Abs on ELISA. She refused taking any disease-modifying anti-rheumatic agents including immunosuppressive drugs during the course of her disease. Donor information and demographics are summarized in (Supplementary Table 1). Tendons were immediately placed in (DMEM)/F12 medium (D8437, Sigma-Aldrich) in the operating room and subsequently processed in the cell culture lab. Human tendon-derived stroma cells were isolated by collagenase digestion as described elsewhere[104].

Briefly, tendon samples were washed in PBS to remove blood or tissue debris. Surrounding fat, fascia or muscles were excised, and tendon tissue was cut into ~5 mm × 5 mm pieces followed by 6–12 h digestion with Collagenase, Type I (17018029, Gibco™) at 37 °C. Isolated cells were allowed to grow in DMEM/F12 supplemented with 20% heat-inactivated fetal bovine serum (FBS – 10500, Gibco™), 1% (v/v) Penicillin-Streptomycin (P/S, P0781, Sigma-Aldrich) and 1% (v/v) Amphotericin B (15290018, Gibco™). Cells were maintained in a culture incubator set at 37 °C and 5% $CO_2$, and fresh media were replenished every 3–4 days until cells reached 80% confluency in T75 culture flasks. Cells were routinely sub-cultured once during initial expansion and were used between passages 2 and 5 for all experiments.

## Fabrication of cell-laden, tendon-like constructs

Tendon-derived cells were encapsulated in hydrogels by mixing with neutralized collagen solutions prior to gelation, and cultured for 2–5 days. The two time-points were chosen to allow for full compaction of constructs (day 2) and additional days for tissue maturation (day 5). Collagen gels were produced using rat tail collagen type I (High Concentration type I collagen – 354249, Corning®). A 2× acidic collagen solution was neutralized on ice to pH 7.2–7.4 with precooled 10× PBS (70011044, Gibco™) and 1M sodium hydroxide (S5881, Sigma-Aldrich). Next, this solution was mixed at a ratio of 1:1 with a 2× concentrated cell solution to reach a final collagen concentration of 1.7 mg/ml. A 90 μl of the liquified cell-laden gel solution was added per PDMS reservoirs and was allowed to polymerize for 45 min, after assembling the plates. Plates were then carefully removed from the incubator and 1 mL of supplemented culture medium was added to each well. When indicated, cells were treated with recombinant TGF-β1 (580702, BioLegend®), as specified in figure legends, and were cultured in growth media containing 1% FBS.

**Tendon fibroblasts bone marrow-derived macrophages direct co-culture experiments.** For co-culture experiments, M-CSF pre-differentiated BMDM were pooled from 2 to 3 mice per genotype (WT or Fosl-2[(Tg)]). Pooled differentiated BMDM were mixed with WT or Fosl-2[(Tg)] tendon fibroblasts at 1:1 mixing ratio, in a mix-and-match fashion, and at a density of $4 \times 10^6$ cell/ml. Pre-mixed macrophage-fibroblast populations were encapsulated in collagen hydrogels and cultured under variable tension at a final seeding density of $1 \times 10^6$ cell/ml, as described above. Co-cultures were maintained in DMEM - high glucose media supplemented with 10% FBS, 1% Penicillin-Streptomycin, 1% MEM Non-essential Amino Acid Solution (M7145, SAFC Sigma-Aldrich) and 200 μM L-ascorbic acid phosphate magnesium salt. Tissue tension was quantified by monitoring the post deflection for up to 48 h.

## RNA extraction from human and Fosl-2[(Tg)] mice tendons

Tail fascicles were placed in 2 mL Eppendorf Safe-Lock tubes, snap frozen in liquid nitrogen and stored at −80 °C until further assayed. On the day of isolation, tubes were placed immediately in dry ice. Samples were transferred to supercooled Spex microvial cylinder (6757C3, SPEX™ SamplePrep) containing 150 μl of GENEzol™ reagent (GZR200, Geneaid). Samples were cryo-grinded in a bath of liquid nitrogen using FreezerMill (6870, SPEX™ SamplePrep) until fascicles were completely disrupted. Pulverized samples were collected by rinsing the tubes with 900 μl GENEzol™, transferred to a clean 1.5 ml tube and kept in dry ice. To purify RNA, tissue homogenate was thawed and mixed well with 200 μl of Chloroform (102445, Merck) at a mixing ratio of 1:5. Samples were then spun down for 15 min (at $15,000 \times g$, 4 °C). The upper RNA-containing aqueous phase was transferred to a clean 1.5 ml tube and mixed with one part of 70% ethanol. RNA cleanup was subsequently performed using PureLink™ RNA Micro Scale Kit (12183016, Invitrogen™), including an on-column DNA digestion step with DNase I (DNASE70, Sigma-Aldrich). For collagen gels, constructs were collected in Qiagen RLT Plus lysis buffer (1053393, Qiagen), and were

snap-frozen in liquid nitrogen and stored at −80 °C. Samples were disrupted by vertexing for 1 min. followed by homogenization using QIAshredder columns (79654, Qiagen). Total RNA was isolated using RNeasy Plus Micro Kit (74034, Qiagen), including a step to remove genomic DNA (Qiagen gDNA Eliminator spin columns). Cleaned RNA concentration and quality were determined by spectrophotometric determination of $A_{260}/A_{280}$ ratio. Samples were stored at −80 °C, and only samples with $A_{260}/A_{280}$ ratio of 1.8–2 were used for downstream RT-qPCR analysis.

### Real-time quantitative PCR (RT-qPCR)

Real-time quantitative PCR were performed using StepOnePlus Real-Time PCR System (4376600, Applied Biosystems™) and TaqMan Assays. 90 ng of total RNA were reverse transcribed to cDNA using High-Capacity RNA-to-cDNA Kits (4387406 or 4368814, Applied Biosystems™). RT-qPCR reactions were run with 2 µl cDNA and 8 µl of TaqMan® Mastermix (containing 5 µl Universal PCR Master Mix, 0.5 µl of TaqMan primer, 2.5 µl of ultrapure water) adding up to a total volume of 10 µl. Primer details and assay IDs are listed in the online supplementary material (Supplementary Table 2). Reactions were carried out in technical duplicates. Relative expression was calculated using the comparative $2^{-\Delta CT}$ method[105].

### Bulk RNA sequencing

Total RNA was isolated using RNeasy Plus Micro Kit (74034, Qiagen), including a genomic DNA removal step with Qiagen gDNA Eliminator spin columns. RNA-Seq library preparation was carried out by GENE-WIZ (Leipzig, Germany). Briefly, RNA quantity was measured using Qubit 4.0 Fluorometer (Life Technologies, Carlsbad, CA, USA), and its RNA integrity was checked with RNA Kit on Agilent 5300 or 5600 Fragment Analyzer (Agilent Technologies, Palo Alto, CA, USA). Ribosomal RNA (rRNA) was depleted using NEBNext rRNA Depletion Kit (Human/Mouse/Rat). RNA-seq libraries were constructed using NEBNext Ultra RNA Library Prep Kit for Illumina, following the manufacturer's guidelines (NEB, Ipswich, MA, USA). Enriched RNAs were fragmented and first-strand and second-strand cDNA were subsequently synthesized. cDNA fragments were end repaired and adenylated at 3'ends, universal adapter was ligated to cDNA fragments, followed by index addition and library enrichment with limited cycle PCR. Sequencing libraries were validated using NGS Kit on the Agilent Fragment Analyzer, and quantified by using Qubit 4.0 Fluorometer. The sequencing libraries were multiplexed and loaded on the flow cell on the Illumina NovaSeq 6000 instrument (or equivalent) according to manufacturer's instructions. Libraries were sequenced using a 2 × 150 Pair-End (PE) configuration v1.5. to a sequencing depth of at least 20 million reads. Image analysis and base calling were conducted by the NovaSeq Control Software v1.7 on the NovaSeq instrument. Raw sequence data (.bcl files) generated from Illumina NovaSeq was converted into fastq files and de-multiplexed using Illumina bcl2fastq program version 2.20. One mismatch was allowed for index sequence identification.

### Data preprocessing and bioinformatics analysis

Raw sequencing data were processed using R package ezRun, which is implemented within the SUSHI framework of the Functional Genomics Center Zurich (ETH Zurich and the University of Zurich)[106]. Sequencing quality control was performed with MultiQC (version 1.9)[107]. Raw.fastq files were mapped to the Mus musculus genome reference (GENCODE GRCm39 – Release_M26-2021-04-20) with STAR aligner software[108]. Reads counts per gene were calculated using featureCounts function against the reference genome (GENCODE GRCm39).[49] Differential expression analysis was imputed by using the DESeq2 R package (version 1.36.0) with default parameters[109]. Pathway enrichment analysis was carried out using Enrichr[110]. Gene set functional enrichment

was performed using eVITTA package in RStudio[111], and the gene set overrepresentation analysis (ORA) implemented within clusterProfiler package [v3.18.0][112]. Cellular deconvolution was performed using the CIBERSORTx tool[113,114], and cell-type signature matrix of De Micheli et al.[115]. Volcano plots were visualized using VolcaNoseR package[116]. Overlayed comparison of RNAseq dataset was carried out with Metascape software[117].

### Pharmacological treatments of tissues with agonists and inhibitors

To test the dynamic contractility and temporal response to stimulation with soluble factors, cell-laden collagen hydrogels were formed and cultured in standard growth media for 24 h as described above. Then constructs were serum-starved in 1% FBS for another 24 h to reduce the basal levels of cytoskeletal tension. Cellular contractility was modulated by incubating tissues with Oleoyl-L-α-lysophosphatidic acid (LPA) at a final concentration of 20 µM (L7260, Sigma-Aldrich), or (−)-Blebbistatin at a final concentration of 50 µM (B0560, Sigma-Aldrich). Vehicle-alone controls were: PBS for LPA and DMSO for Blebbistatin in serum-reduced media.

### Immunostaining and confocal fluorescence microscopy

Collagen constructs were fixed while still under tension with pre-warmed 4% neutral buffered formaldehyde solution (ROTI®Histofix 3105.2, Carl Roth) for 45 min at room temperature. Subsequently, tissues were permeabilized with 0.5% Triton X-100 (93418, Sigma) in PBS for 30 min, and blocked overnight with 3% bovine serum albumin (BSA - P6154, Biowest) in a humidified chamber at 4 °C. Samples were immersed in Image-iT FX Signal Enhancer blocking solution (I36933, Invitrogen™) for at least 30 min before overnight incubation with primary antibody solution at 4 °C. After washing, tissues were incubated with the appropriate fluorescently-conjugated, isotype-specific secondary antibody for 4 h, and protected from light. Nuclei and actin were counterstained with NucBlue and AlexaFluor-conjugated phalloidin, respectively. Samples were mounted in 120 µm Secure-Seal™ adhesive spacers and embedded in Agarose, low melting point (V3841, Promega). Immunofluorescence images were acquired with the iMic spinning disk confocal microscope, fitted with a Hamamatsu Flash 4.0 sCMOS camera and a SOLE-6 Quad laser (Omicron), using 10× (N.A. 0.4) and 20× (N.A. 0.75) objectives (Olympus UPLSAPO).

### Multiphoton microscopy and second harmonic generation (SHG) imaging

Multiphoton microscopy and SHG imaging were performed on unfixed, 8 µm thick mouse Achilles tendon frozen sections. Frozen sections were equilibrated to room temperature for 10 min. LOX-inducible probe was added to the unfixed sections and allowed to incubate at 37 °C for 3 h in a humidified chamber[73]. Slides were washed twice in PBS for a total of 5 min, and subsequently fixed for 10 min in ice-cold acetone at −20 °C. Following fixation, sections were washed in PBS three times (each for 5 min), counterstained with DRAQ5 (62251, Thermo Scientific™) at a concentration of 1:1000. Slides were mounted in Immu-Mount medium and covered with coverslips. SHG imaging was performed on sections using a Leica TCS SP8 microscope equipped with a ×25 0.95 numerical aperture (NA) L Water HCX IRAPO objective and a Mai Tai XF (Spectra-Physics) MP laser tunable from 720–950 nm. The LOX-inducible probe excitation was generated at ~760 nm, and detected at the emission wavelength with an external detector equipped with a filter for DAPI (460 nm). Sequentially, the SHG signal was acquired with an aligned condenser (0.55NA) using an 880 nm excitation laser, and detected around half the incident wavelength (435–485) with an opened pinhole. Laser power, photomultiplier tube (PMT) voltage, gain and offset were monitored and kept constant throughout each acquisition[118].

## Differential scanning calorimetry (DSC)

Mouse tail tendons were extracted as described in the "Biomechanical testing" method section and were equilibrated in PBS during testing. Human tendon tissues were collected by cryo-sectioning from unfixed, OCT-embedded tissue blocks that were stored at −80 °C (8–10 sections of 20–30 μm thickness). Tissue sections were washed extensively with PBS to remove the water-based OCT embedding medium, and were allowed to equilibrate in PBS. For DSC measurements, each mouse or human sample was blotted in Kimwipes® to remove excess PBS, and were subsequently weighed, and then hermetically sealed in a 40 μl aluminum crucible pan (30389221, Mettler-Toledo). Samples heat flow was scanned to 100 °C at 5 °C/min. After scanning, pan were punctured, dried and reweighted. Endotherms were analyzed to determine onset temperature ($T_{onset}$), peak temperature ($T_{peak}$), full-width at half-maximum (FWHM) using Origin 2023.

## Quantification of mature cross-links

Tendon samples were thawed in ice and equilibrated in PBS. Tissues were reduced with 0.22 mM potassium borohydride (438472, Sigma-Aldrich) for one hour at room temperature, collected by centrifugation, washed twice with Milli-Q water, and lyophilized to dryness overnight. Next, samples were resuspended to 10–50 mg/ml and hydrolyzed in 6 M hydrochloric acid (258148, Sigma-Aldrich) for 20 h at 95 °C. One-fourth of the hydrolyzed volume was used for the collagen quantification assay. HCl was evaporated to dryness from the rest. Dry hydrolysate was reconstituted in water to a final concentration of 30 mg/ml to measure mature pyridinium cross-links (Pyridinoline PYD and Desoxypyridinolin DPD). PYD and DPD cross-links were quantified using competitive ELISA immunoassay (8010, Quidel Corporation, San Diego, USA) following the manufacturer's protocol and including the rapid vigorous washing step. Results were analyzed with 4-parameter calibration curve fitting equation using GainData® online software (Arigo Biolaboratories, https://www.arigobio.com/ELISA-calculator).

## Total collagen content

Collagen content was quantified using QuickZyme total collagen assay (QZBtotcol, QuickZyme Biosciences) according to the manufacturer's protocol. Briefly, tendon fascicle hydrolysate was first diluted with Milli-Q water to a final concentration of 4 M HCl (1 volume hydrolysate: 0.5 volume water). Samples were further diluted 1:100 with 4 M HCl to be in the linear range of the collagen standard curve. Diluted samples were pipetted to 96-well plates and incubated with Assay Buffer and Detection Buffer as specified in the manufacturer's protocol. Colorimetric absorbance was measured at 570 nm using microplate spectrophotometer (Epoch, Biotek). Collagen content and the amount of mature cross-links were normalized to tissue dry weight (mg).

## Statistics

All statistical analyses were performed using Prism 10 (Version 10.3.0, GraphPad Software) and DABEST-Matlab for the estimation statistics (Matlab® R2018b, MathWorks, Inc.). Samples were checked for normality distribution with Shapiro–Wilk test. Statistical differences were evaluated with Unpaired Student's $t$ test with Welch's correction, Ordinary one-way or two-way ANOVA (followed by Holm–Šídák's multiple comparisons test), for normally distributed data. Kruskal–Wallis or Mann–Whitney tests were used in cases where the normal distribution criterion was not met. Non-parametric: Wilcoxontest was used for paired data. Exact $P$-values and post-hoc tests are reported in the figure legends. Whenever applicable, we also report the magnitude of the effect (Cohen's d effect size) and CIs. All figures for estimation statistics, permutation $p$-values, and further discussion on how we defined technical vs. biological replication units are included in the Supplementary Information "see Supplementary Discussion". We note that variability in replicate numbers (n) across some

experiments is due to multiple factors. In mechano-culture experiments, this variability resulted from rupture or degradation of collagen hydrogels during the experiments. In Fosl-2$^{(Tg)}$ experiments, it arose from low RNA yield in Fosl-2 tendons, as tissue crosslinking makes RNA extraction particularly challenging. Additionally, Fosl-2$^{(Tg)}$ mice often develop a severe phenotype that necessitates euthanasia for welfare reasons, leading to a relatively high dropout rate compared with WT controls within each litter. When sample numbers were limited, we prioritized the most important readouts.

## Ethical statement

Human samples were collected with written informed donor consent with voluntary participation, in compliance with the requirements of Declaration of Helsinki, Swiss Federal Human Research Act (HRA), and Zurich Cantonal Ethics Commission (Approval numbers: 2015-0089, 2020-0119). Tissue harvesting from Wistar rats and animal experiments with the Fosl-2$^{Tg}$ mice were approved by Zurich Cantonal Veterinary Office (Permits: ZH007/2019 Fosl-2$^{Tg}$, ZH265/14 - ZH239/17 Wistar rats).

## Reporting summary

Further information on research design is available in the Nature Portfolio Reporting Summary linked to this article.

## Data availability

Source data files are provided with this paper. The RNA-sequencing data generated in this study have been deposited in the Gene Expression Omnibus (GEO) database under accession number GSE319654. The raw data that support the findings of this study are available from the corresponding author upon request. Source data are provided with this paper.

## Code availability

The code developed in this manuscript is publicly available at Zenodo under accession: 18455078 (https://doi.org/10.5281/zenodo.18455077).

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

## Acknowledgements

We are grateful for the donors who generously consented to use their tissue in this work. We acknowledge the support of clinical teams, research nurses, administrative (Helen Strebel) and the cleaning staff at Balgrist University Hospital and the Swiss Center for Musculoskeletal Biobanking. We would like to thank: Silvio Broder for his assistance with the biomechanical testing, Dr. Astrid Jüngel and Dr. Stefan Dudli (University Hospital of Zürich) for the constructive discussion and feedback, and Prof. Raffaele Mezzenga and Dr. Yang Yao (ETH Zurich) for accessing the differential scanning calorimeter, Prof. Mark Jones (University of Southampton) for the PYD/DPD ELISA protocols. We are grateful for the graphic design support we have received from Simone Zaugg (ETH Zurich). This work has been funded by the Cariplo Foundation [2016–0481], the Vontobel Foundation, and institutional funding of ETH Zurich and Balgrist University Hospital (to J.G.S.).

## Author contributions

A.A.H. and J.G.S. conceived the study and designed the experiments. R.K., J.F., and A.A.H. prototyped and tested the mechano-culture platform. A.A.H. performed the experiments, analyzed and interpreted the data and wrote the manuscript. F.R. and A.H. assisted with Fosl-2(Tg) mouse experiments. S.L.W. performed biomechanical tests. M.A.R. synthesized the Lox activity probes and performed multiphoton imaging. B.N. assisted with experimental work and performed some of the RT-qPCR experiments. H.W. and O.D. provided resources. J.G.S. acquired funding, supervised the project, interpreted the data, and commented on the manuscript. All authors approved the final version of the manuscript.

## Funding

## Competing interests

O.D. has/had consultancy relationship with and/or has served as a speaker for the following companies in the area of potential treatments for systemic sclerosis and its complications in the last three calendar years: 4P-Pharma, Abbvie, Acceleron, Acepodia Biotech, Alcimed, Altavant, Amgen, AnaMar, Aera, Argenx, AstraZeneca, Blade, Bayer, Boehringer Ingelheim, Calluna (Arxx), Cantargia AB, Catalyze Capital, Corbus, CSL Behring, Galderma, Galapagos, Glenmark, Gossamer, Horizon, Janssen, Kymera, Lupin, Medscape, MSD Merck, Miltenyi Biotec, Mitsubishi Tanabe, Nkarta Inc., Novartis, Orion, Pilan, Prometheus, Quell, Redxpharma, Roivant, EMD Serono, Topadur and UCB. Patent issued "mir-29 for the treatment of systemic sclerosis" (US8247389, EP2331143). Co-founder of CITUS AG. Research Grants: BI, Kymera, Mitsubishi Tanabe, UCB. The remaining authors declare no competing interests.
