## [Transparent Peer Review file · Nature Communications]

Mechanical Gating of Tendon Fibrogenic Transcription in Systemic Sclerosis

Corresponding Author: Professor Jess Snedeker

Version 0:

Reviewer comments:

Reviewer #1

(Remarks to the Author)

The authors have conducted an elegant study, designing a novel 3D mechano-culture system that allows culture of cells under variable static tension, to mimic fibrotic conditions. The mechanisms leading to fibrosis are poorly understood, and therefore development of physiologically relevant systems in which to investigate these mechanisms are paramount to better appreciate the interplay between ECM tension, fibroblast activation and fibrotic remodelling.

In this study, the authors report development and validation of a simple culture system in which stiffness experienced by cells can be altered while keeping collagen content constant. They then use this system to demonstrate increased mechanical tension induces fibrosis in tendon-derived cells, similar to the response seen when cells are treated with the pro-fibrotic mediator TGF-beta. They then compared the response of tendon cells from healthy donors with those derived from a patient with systemic sclerosis, demonstrating persistent fibrotic activation in SSc cells. Finally, they investigate factors influencing the altered tendon mechanics observed in a mouse model of systemic fibrosis, demonstrating this may be driven by alterations in collagen crosslinking, although this has not been confirmed. They also show that co-culture of fibroblasts and macrophages increased WT fibroblast traction forces, when cultured with Fra2 derived macrophages.

While many aspects of this study are novel and have been investigated thoroughly, I have some concerns/comments regarding the validity and robustness of the study which need to be addressed – see comments below:

Major comments:

1. A consistent, and rather unexpected finding, is that col1 gene expression is decreased in pro-fibrotic conditions, and this requires further investigation. Is it possible to measure collagen synthesis at the protein level in the hydrogels? The authors do report reduction in col1a1 gene expression in tendons from fra2 mice, providing some confidence that these results are not an artefact of the in vitro culture system, however this decrease in collagen gene expression is not associated with decreased hydroxyproline content (Fig.5f – it is not clear if this data is from the early or established group). Providing data showing expression of col1a1 in healthy and SSc human tendon samples, as well as hydroxyproline content would provide further confidence in these results.
2. The authors present some limited evidence that the differences observed in mechanical properties in tail tendons from WT and Fra2 mice are due to alterations in collagen crosslinking, however these results need to be explored in more detail to be able to conclude that altered crosslinking drives the differences in mechanics seen. I strongly recommend that the authors measure specific crosslinks in the WT and Fra2 mice tails to establish if it is indeed altered crosslinking that is driving the differences in mechanical properties seen. If human samples are available for similar analysis that would strengthen the results further. It also seems counterintuitive that levels of cross-linking associated genes are decreased in fibrotic conditions which would imply decreased, rather than increased cross-linking. Please comment on this important point. How do the authors think that altered crosslinking may ultimately result in the pro-fibrotic activation of fibroblasts observed?
3. It is unfortunate that fibrotic samples could only be obtained from 1 human donor, and the methods do not describe how this sample was obtained, or if donor characteristics were matched with healthy tendon samples. However, the authors have attempted to address this limitation by investigating response of fibrotic cells derived from a mouse model of systemic sclerosis. Please show representative images of hydrogel compaction comparing WT and Fra2 cells in Fig. 5. Did the authors establish if there were differences in α -SMA/F-actin and cytoskeletal alignment between cells from normal and fibrotic tendons (both in human and mouse experiments)? Were the same panel of genes assessed for both human and tendon experiments? Supplementary table 1 seems to be missing from the supplementary information.
4. The co-culture experiments are not described in the methods, and the results of these experiments are not discussed in

light of the other findings of this study, nor in regard to the relevant literature.

5. Please check figure legends carefully – in Figure 4, panels are described that are absent from the figure itself. Please clarify if immunofluorescence was performed on these samples.

6. Please check the manuscript carefully for typos, particularly the methods section

Reviewer #2

(Remarks to the Author)

Interesting study looking into how dysregulated matrix mechanics affect cell phenotype towards a pro-fibrotic phenotype using tendon-derived stromal cells and systemic sclerosis (SSc) as a model. Elegant methodology with the development of a new mechanoculture platform to support the investigation of the effect of matrix stiffness on the activation of fibroblasts, with validation of dysregulated matrix mechanics in the tendons of the Fra2 mouse model of systemic sclerosis, and use of tendon-derived stromal cells from Fra2 transgenic mice and wild type mice to identify functions affected by dysregulated matrix mechanics. The authors succeed to recreate a pro-fibrotic phenotype similar to the one induced by TGF β through tuning of matrix stiffness and report increased matrix stiffness negatively correlate with major pro-fibrotic ECM genes expression and positively correlate with the expression of markers of stromal-immune interactions and that co-culture with bone marrow-derived macrophages can regulate gene expression of tendon-derived stromal cells independently of matrix mechanics. In addition, the developed mechanoculture platform has great potential for future studies in the mechanism of fibroblast activation through dysregulated matrix mechanics.

Major points

1. How does the matrix stiffness generated by the compliant and the rigid PDMS mats relate to physiologic tendon matrix stiffness? How did you determine the rigid PDMS creates an “abnormal” or higher than physiologic matrix stiffness and the compliant substrate recreates the physiologic tendon microenvironment closer?
2. Were technical replicates used for statistical analysis instead of biological replicates (except for the SSc where this was unavoidable)? Was a statistician consulted for the use of technical replicates instead of biological for the statistical analysis?
3. Line 272 and Figure 5C – Since you are measuring the fascicles CSA and reporting differences in CSA between wild type and Fra2Tg fascicles why not calculate and plot failure stress instead of force to facilitate the comparison of properties between wild type and Fra2Tg fascicles? Or at least mention it in the text if there was no significant difference in failure stress measured between wild type and Fra2Tg fascicles.
4. Line 472 – Isolation of tendon-derived stromal cells - Please include donor age and gender for SSc and healthy tissue. Were the healthy controls age-matched to the SSc donor? If not, please include in the limitations as ageing can also have an impact on cell phenotype and expression.

Minor points

Line 144 – Within hours after seeding, tissue tension evidently increased after two hours, and ... - change to avoid repetition of within hours.

Figure 1A legend – Sagittal section might be more accurate than cross section. A cross section view of the well could be included to show the initial set up view from the top/bottom of the well.

Figure 1A – “PDMS matt” - mat is misspelled in the actual figure.

Figure 1G or Line 148 – Was the increase and decrease statistically significant?

Figure 1G – Was the provenance of these cells also rat tail tendon? Maybe add in the legend.

Line 154 – “in in” repetition.

Figure 1I – Is the “active forces” column representing the vehicle tissue tension after 48 h? As the magnitude does not appear to correspond to the magnitude of the data point in 1G.

Line 177 – “consistent with” appears that part of the sentence missing.

Line 190 – “the phenotype”

Line 200 – Maybe better to specify immune staining intensities (rather than immune intensities).

Line 207 – The way the sentence about the tenogenic markers is formulated it would seem Mx is also significantly decreased which does not seem to be the case from Figure 3K – please amend accordingly.

Figure 4C – why does line graph has n=24/group and the violin plot has 16/group – isn't the source of the data the same presented in a different way?

Figure 4D and E legend – The legend says immunostaining quantification and the text says mRNA expression and panels appear mislabelled/the legend includes panels that do not appear in the figure. Please amend appropriately.

Figure 4 D-G – The x axis labelling is complex when the subcategories could just be labelled

“ | compliant | rigid | ” since you have already colour-coded the healthy and SSc – just a suggestion. Same for Figure 5D.

Line 244 – You are not specifying whether the dysregulation is observed under compliant or rigid boundary mechanics conditions which gives the impression that this dysregulation is being observed under non-dysregulated mechanics/compliant boundary mechanics conditions. This is not the case for IL6 and IL8 though which are not significantly different between healthy and SSc tendon-derived stromal cells under compliant conditions. Please specify which conditions you are referring to or remove IL6 and IL8 if they are not significantly regulated.

Line 252 – Is it an inability of SSc cells to sense boundary rigidity or expression of COL1A1 is already lower in SSc cells so it does not get significantly lower in rigid conditions? As for other genes eg IL6, IL8, and ICAM a significant difference in expression is noted between healthy and SSc cells under rigid conditions but not compliant – appearing to show a response to boundary rigidity in SSc cells. You also mention in the next sentence SSc-derived tendon cells show stiffness-sensitivity of stromal-immune interactions markers – unless that was “and (absence of) stiffness-sensitivity of stromal-immune interactions markers”.

Line 263 – “We have recently reported...” reference (65) is missing. Also, reference 65 – is missing a title, journal volume

and pages.

Figure 5D – Are the p values for Fn1 and Lox missing?

Line 317 – “mirroring our observations in Fra2 tendons in vivo and SSc-derived tendon fibroblasts” but you report no significant changes in Col1a1 expression under rigid conditions in SSc cells (and Lox expression is not reported) and Lox expression in Fra2 tendons does not appear to be significantly different from WT unless there are some significant p values missing from the graphs. Please clarify sentence if needed.

Line 334 – “Elevated rigidity” – compared to what? Physiologic matrix stiffness in tendons?

Line 368 – As you are mentioning in line 370 the referenced paper in line 368 does not really investigate fibrosis and cell environments of increased stiffness so maybe using “This was further reported/This decreased transcription was further reported” or similar could be a better representation than “This was further confirmed”.

Line 404 – Diameter and length details are missing.

Line 405 – “Each well in the top plate’s upper and ...”

Line 412 – “mm” repetition

Device assembly paragraph (line 402) – The setup of the anchoring spheres is not included but would aid in the overall understanding of the setup of the spheres in relation to the posts.

Line 437 – “software.” comma instead?

Line 441 – to create

Line 541 – 90 ng – space between number and unit

Line 551 – “The then”

Line 570 – Statistics

- Two-way ANOVAs are also mentioned in the figure legends but not in the statistics paragraph.

- (from major points) Were technical replicates used for statistical analysis instead of biological replicates (except for the SSc where this was unavoidable)? Was a statistician consulted for the use of technical replicates instead of biological for the statistical analysis?

- Some experiments had a wide variation of sample numbers per group eg Fig 5D n=4-9. Was that due to loss of samples in the process resulting in some groups having less samples than others or sample additions where it was deemed to be needed or cell numbers/availability?

Reviewer #3

(Remarks to the Author)

This manuscript describes a substantial body of work that addresses mechanical factors that may be highly relevant to the tendon abnormalities which are a hallmark of systemic sclerosis patients. The authors have developed a culture technique that allows mechanics tension to be applied in vitro and using this methodology have defined effects on fibroblasts and extended previous work that shows how mechanical tension may promote the contractile phenotype and profibrotic properties of fibroblasts. A key over-arching question is the extent to which these findings may apply to a broader population of fibroblasts that those derived from tendons.

Specific comments:

1. It would be helpful to explain more clearly how the mechanical forces are generated and different between cultures. Whilst the authors obviously have a very clear understanding it is less apparent to readers unfamiliar with the concepts of stiffness and elasticity and this should be explained to help with later interpretation of the data, especially for mutant mice at different time points.
2. More extensive comparison of SSc fibroblasts should be possible even though tendon samples may not be available routinely. It would be an important control experiment to explore whether dermal fibroblasts show the same or different responses to those observed for tendon derived cells.
3. It is unreliable to draw conclusions from a single patient sample. It would be helpful if additional material could be obtained from a range of SSc patients and perhaps compared with other fibrosing disorders such as Dupuytren's contracture that may be more available. Although 3 biologically independent experiments are performed it is not clear how this was achieved with a single donor.
4. I am somewhat confused by references to the *fosl2* mutant mice as well as Fra2 and the authors should clarify this and how the different mouse strains were examined (page 29 line 449)
5. The transcription factor MRTFA is considered critical in mechano-sensing in fibroblasts, and it would be helpful to have this included in the relevant gene expression and protein assays to explore the impact on myofibroblasts differentiation and contractility.
6. The relative importance of motility, contractility and matricellular interaction in determining the contraction in this culture system should be considered as it is unlikely that cell contractility is a major determination of tendon biomechanics that is driven more by extrinsic skeletal muscle contraction. This may be a limitation of the current approach and should be clarified.
7. The findings reported here concerning inflammatory cells interfering with normal tension induced down regulation of ECM may be relevant to other conditions such as adhesive capsulitis and this could be discussed.

Version 1:

Reviewer comments:

Reviewer #1

(Remarks to the Author)

The authors have addressed all my comments and concerns thoroughly and I have no further comments on the revised manuscript.

Reviewer #3

(Remarks to the Author)

The authors have carefully and comprehensively addressed the points raised at review. This is appreciated and the rebuttal document includes significant new data and analysis. The revised paper is clearer and more complete. No further comments.

REVIEWER COMMENTS

Reviewer #1 (Remarks to the Author):

The authors have conducted an elegant study, designing a novel 3D mechano-culture system that allows culture of cells under variable static tension, to mimic fibrotic conditions. The mechanisms leading to fibrosis are poorly understood, and therefore development of physiologically relevant systems in which to investigate these mechanisms are paramount to better appreciate the interplay between ECM tension, fibroblast activation and fibrotic remodelling.

In this study, the authors report development and validation of a simple culture system in which stiffness experienced by cells can be altered while keeping collagen content constant. They then use this system to demonstrate increased mechanical tension induces fibrosis in tendon-derived cells, similar to the response seen when cells are treated with the pro-fibrotic mediator TGF-beta. They then compared the response of tendon cells from healthy donors with those derived from a patient with systemic sclerosis, demonstrating persistent fibrotic activation in SSc cells. Finally, they investigate factors influencing the altered tendon mechanics observed in a mouse model of systemic fibrosis, demonstrating this may be driven by alterations in collagen crosslinking, although this has not been confirmed. They also show that co-culture of fibroblasts and macrophages increased WT fibroblast traction forces, when cultured with Fra2 derived macrophages.

While many aspects of this study are novel and have been investigated thoroughly, I have some concerns/comments regarding the validity and robustness of the study which need to be addressed – see comments below:

Major comments:

1. A consistent, and rather unexpected finding, is that col1 gene expression is decreased in pro-fibrotic conditions, and this requires further investigation. Is it possible to measure collagen synthesis at the protein level in the hydrogels? The authors do report reduction in col1a1 gene expression in tendons from fra2 mice, providing some confidence that these results are not an artefact of the in vitro culture system, however this decrease in collagen gene expression is not associated with decreased hydroxyproline content (Fig.5f – it is not clear if this data is from the early or established group). Providing data showing expression of col1a1 in healthy and SSc human tendon samples, as well as hydroxyproline content would provide further confidence in these results.

First, we thank the reviewer for their careful reading of the work. Several critical points were raised, and we appreciate the opportunity to elaborate on these findings.

Fig.5f in the original submission included only mice from the established group. To address this point, we have additionally quantified total collagen amounts in tendons from $N=14$ mice of the early group (6-8 week-old) and added 9 mice to the established (>16 week-old) group ($N=17$), **[Figure 5F, Rebuttal figure 1 A-B]**. Confirming the observations in the original submission, there was no statistical difference in total collagen amounts between the Fosl-2^(Tg) and WT controls in the early ($p=0.7007$), or established ($p=0.2806$) groups. Interestingly, tendons from the established Fosl-2^(Tg) group had significantly higher collagen amounts compared to the Early groups: WT controls ($p=0.0222$) and Fosl-2^(Tg) ($p=0.0071$). Furthermore, there was no statistical difference in collagen amounts between the early and established wild-type groups ($p=0.3241$), suggesting that the observed difference in collagen amounts in Fosl-2^(Tg) tendons are likely mediated by the Fosl-2 transgene, and likely not by confounding effects from the aging process. **(Figure 5F, Rebuttal figure 1 A-B)**.

This point is addressed in the results **[Lines: 327 – 332]**, and discussion **[Lines 550 – 561]** sections.

Next, we performed a preliminary experiment where we isolated tendon stromal cells from the Fosl-2^(Tg) and WT established phenotype and seeded these cells onto silicone mechano-variant substrates with elastic modulus (E) of 35 kPa. We chose this elasticity range based on our previous work where we showed that ≈ 35 kPa stiffness reinstates a tendon-like transcriptomic signature and favors minimizing fibro-inflammatory activation of tendon stromal cells in vitro (in comparison with tissue culture plastic).(1) We reasoned that the downregulation of *Col1a1* in established Fosl-2^(Tg) tendons is not a cell-autonomous process, and that removing cells from their native, mechanically-stressed environment will alter *Col1a1* mRNA expression. Confirming our hypothesis, we found that Fosl-2^(Tg)-derived tendon stromal cells expressed significantly higher levels of *Col1a1* mRNA transcripts compared to WT controls (**Rebuttal figure 1 C**). This suggests that the suppression of collagen expression in Fosl-2^(Tg) is potentially mediated, at least in part, by the stiffening of the extracellular matrix and the associated changes in its biophysical cues (i.e. elasticity, tension, confinement, etc..). For more discussion addressing the potential processes mediating this response, we refer the reviewer to our response to their point no. 2 below. (Section: Mechanical confinement and RNA polymerase II repression of transcription).

Next, to get a more global overview on the expression of collagens in Fosl-2^(Tg) tendons (and to exclude that the observed negative correlation between tissue stiffening and fibrogenic transcription might have risen from “bias” in our selection of *Col1a1* gene), we further performed bulk RNA-seq on tendon fascicles from 28 mice [early phenotype $N= 15$ (7 WT + 8 Tg) and established phenotype $N= 13$ (6 WT + 7 Tg)]. The findings of the RNA-seq analysis are discussed in detail elsewhere, however, here we only highlight the findings that are relevant to the reviewer’s questions:

- A) Over-representation analysis (ORA) of the differentially expressed genes showed that tendons from the established phenotype had significant over-representation of Gene Ontology (GO) terms related to extracellular matrix processes (GO:0030199 collagen fibril organization, GO:0030198 extracellular matrix organization) in the upregulated genes of the Early cohort and in the downregulated genes of the established phenotype (**Rebuttal figure 1 D-E, New Figure 6 D-I, New supplementary figure 15 A-D**).
- B) A closer look at the genes behind these GO terms revealed that genes related to the Collagen family of matrix proteins are mostly overrepresented in the GO terms. Specifically, *Col1a1*, *Col1a2*, *Col2a1*, *Col4a5*, *Col4a4*, *Col8a1*, *Col11a1* were all significantly downregulated in Fosl-2^(Tg) tendons compared to WT controls in the Established phenotype (**Rebuttal figure 1 F, New supplementary figure 15 C**).

Overall, this finding further strengthens our original observations based on the transcription of *Col1a1*.

“Providing data showing expression of col1a1 in healthy and SSc human tendon samples, as well as hydroxyproline content would provide further confidence in these results.”

As suggested by the reviewer, we assessed the mRNA expression of *Col1a1* and *Col3a1* and quantified total collagen amounts in healthy hamstring tendon controls, non-SSc diseased tendinopathic (biceps and Achilles) and SSc human tendon samples (tendons of different anatomical origins from the single SSc donor). The diseased tendinopathic samples were all from a cohort group of more advanced age (65-80 year-old), in order to better match the age of the SSc donor.

Results showed that SSc tendon samples had baseline expression levels of *Col1a1* transcripts that are significantly lower than the age-matched, non-SSc tendinopathic samples ($p= 0.0265$).

Interestingly, baseline tissue expression of *Col1a1* mRNA in the SSc donor was similar to the healthy tendons group ($p > 0.9999$) (Rebuttal figure 1 G, New Figure 4 A, New supplementary figure 7 A and B).

Next, we measured total collagen amounts by quantifying hydroxyprolines in hydrolyzed tissues. Results showed no differences in collagen content between the three tendon groups (Rebuttal figure 1 H, New Figure 4 B). In contrast, SSc and tendinopathic tendons had higher amounts of mature PYD/DPD trivalent crosslinks compared to healthy controls (albeit not statistically significant, $p > 0.06$), (Rebuttal figure 1 I, New Figure 4 C). It is important to mention that while the tendinopathic and SSc samples were age-matched, all SSc samples were derived from a single biological donor, and as such these results should be interpreted with caution.

This point is addressed in the results [Lines: 239 – 259], and in the discussion sections.

Rebuttal figure 1 | Analysis of collagen expression and total amounts in Fosl-2^{Tg} mice and human tendons.

(A) Baseline *Colla1* mRNA expression in fresh frozen Fosl-2^{Tg} and wild-type control. ($N = 6-9$ mice/group, with each data point representing ΔCt value, horizontal lines indicate the median, Two-way ANOVA (genotype, disease stage) with Holm-Šidák's multiple comparisons post-hoc test). All individual gene expression is shown normalized to *Anxa5* and *Gapdh* reference genes. (B) Quantification of hydroxyproline content in tendons ($N = 14$ mice/genotype (Early), $N = 16$ mice/genotype (Established)). Ordinary Two-way ANOVA with Holm-Šidák's multiple comparisons test. (C) mRNA expression of *Colla1* in freshly isolated tendon stromal cells seeded on 35 kPa PDMS mechanovariant substrates. ($N = 4$ replicates/group from 2 experiments), with each data point representing ΔCt , horizontal lines indicate the median. (D-E) Pathway bubble plots depicting over-representation analysis (ORA) of multiple DEGs comparison

between Fosl-2^(Tg) vs. WT tendons in early and established phenotypes. (D) Upregulated genes. (E) Downregulated genes. (F) Normalized expression of collagen genes in bulk RNA-sequencing of established Fosl-2^(Tg) vs. wild-type tendons. (G) Baseline *Coll1a1* mRNA expression in snap frozen human tendons. Each data point representing Δ Ct value, horizontal lines indicate the median. Kruskal–Wallis one-way ANOVA with Dunn's multiple comparisons post-hoc test). All individual gene expression is shown normalized to *RPL13A* and *GAPDH* reference genes. (H) Quantification of hydroxyproline content in human tendons. Horizontal lines represent the median. Kruskal–Wallis one-way ANOVA with Dunn's multiple comparisons post-hoc test. (I) Total amounts of mature trivalent crosslinks (PYD and DPD). Black horizontal lines indicate median values. ($N=$ 13-15 mice/genotype (Early), $N=$ 16 mice/genotype (Established)). Ordinary Two-way ANOVA with Holm–Šidák's multiple comparisons test. For (G-I): ($N=$ 3-6 independent donors/group, except for systemic sclerosis where different anatomical tendons were derived from a single donor.

Abbreviations: (WT): Wild-type, (GO-BP) Gene Ontology – Biological Processes.

2. The authors present some limited evidence that the differences observed in mechanical properties in tail tendons from WT and Fra2 mice are due to alterations in collagen crosslinking, however these results need to be explored in more detail to be able to conclude that altered crosslinking drives the differences in mechanics seen. I strongly recommend that the authors measure specific crosslinks in the WT and Fra2 mice tails to establish if it is indeed altered crosslinking that is driving the differences in mechanical properties seen. If human samples are available for similar analysis that would strengthen the results further. It also seems counterintuitive that levels of crosslinking associated genes are decreased in fibrotic conditions which would imply decreased, rather than increased cross-linking. Please comment on this important point. How do the authors think that altered crosslinking may ultimately result in the pro-fibrotic activation of fibroblasts observed?

We thank the reviewer for raising another important point. We agree that measuring collagen cross-links provides a more direct readout of crosslinking activity, and thus may better explain the mechanical phenotype.

To address this point, and as requested by the reviewer, we optimized an enzyme-linked immunosorbent assay (ELISA) to quantify levels of total mature trivalent pyridinium cross-links (Pyridinolin PYD and Deoxypyridinolin DPD) in Fosl-2^(Tg) mice tails and human tendons. The assay optimizes a commercial kit that provides a highly sensitive method to quantify PYD and DPD crosslinks for evaluation Pyd in urine (as a surrogate for bone turnover) using a specific anti-PYD antibody.(2-5) Furthermore, we directly measured the *in-situ* enzymatic activity of lysyl-oxidase (LOX) using a fluorescent LOX-activatable collagen peptide-sensor probe and two-photon confocal microscopy.(6)

Crosslinks (mouse): We performed the pyridinium cross-links ELISA on ($N=$ 13-15 mice/genotype (Early), $N=$ 16 mice/genotype (Established)). Results showed Fosl-2^(Tg) tail tendons with established fibrosis had indeed significantly higher PYD + DPD crosslinks compared to age-matched, wild-type controls ($p=$ 0.0471) and to Fosl-2^(Tg) with early phenotype ($p<$ 0.0001), (**Rebuttal figure 2A, New Figure 5J**). Similarly, quantification of fluorescence intensity of the LOX-activatable collagen peptide-sensor revealed that Achilles tendons of Fosl-2^(Tg) tendons with established fibrosis had considerably higher *in-situ* LOX enzymatic activity compared to WT controls ($p=$ 0.0077), (**Rebuttal figure 2B, New Figure 5K and L**). Taken together, these additional insights, together with the quantification of collagen amounts and DSC measurements, suggest that increased crosslinking in Fosl-2^(Tg) tendons contribute tissue-level changes in bulk mechanics (i.e. stiffening).

We have now updated the results section [Lines: 340 – 351] to reflect these new insights.

“If human samples are available for similar analysis that would strengthen the results further.”

Crosslinks (human): Furthermore, as briefly discussed in point no. 1, we quantified total collagen amounts and mature crosslinks in the limited human SSc samples (4 different anatomically distinct tendons from a single donor), and compared them to tendinopathic tendons (biceps and Achilles, N= 6) and non-diseased controls (hamstring controls, N=3), (**Rebuttal figure 1 I, New Figure 4 C**). There was no difference in total collagen amounts between the different tendon groups. However, similar to Fosl-2^(Tg) tissues, SSc and tendinopathic tendons had higher amounts of PYD/DPD crosslinks compared to non-diseased controls (albeit not statistically significant; $P= 0.0697$ and $P= 0.0644$, respectively).

Additionally, we performed DSC on the human samples. SSc tendons showed an increase in DSC Onset temperature (Median temp= 66.17 °C) compared to diseased (Median temp= 62.46 °C, $p= 0.0552$) and non-diseased controls (Median temp= 63.67 °C, $p= 0.7701$), although again these differences were not statistically significant (**Rebuttal figure 2 G and H, New Supplementary figure 7 C and D**).

Overall, the results suggest that human SSc tendons showed trends comparable to Fosl-2^(Tg) tendons with established fibrosis, however we interpret these observations with caution, also within the manuscript, given the very small sample size.

We have now updated the results section [**Lines: 250 – 259**] in light with these new insights.

Rebuttal figure 2 | Analysis of collagen expression and total amounts in Fosl-2^(Tg) mice and human tendons.

(A) Total amounts of mature trivalent crosslinks (PYD and DPD). Black horizontal lines indicate median values. ($N= 13-15$ mice/genotype (Early), $N= 16$ mice/genotype (Established)). Ordinary Two-way ANOVA with Holm-Šidák's multiple comparisons test, Cohen's $d = 0.771$ [95.0%CI -0.353, 1.94]. (B) Quantification of fluorescent intensity levels of in-situ Lox collagen peptide-sensor in Established phenotype, Grey symbols indicate values for the representative images in (B). Two-tailed Mann Whitney test. (C) Endothermic onset temperature (°C), (D) peak temperature, and (E) Full-width at half-maximum (FWHM) of thermally-denatured tendons as measured by DSC ($N= 14$ mice/genotype (Early), $N= 16$ mice/genotype (Established)). Ordinary Two-way ANOVA with Holm-Šidák's multiple comparisons test. (F) Total amounts of mature trivalent crosslinks (PYD and DPD). (G) Endothermic onset temperature (°C), (H) Peak temperature of thermally-denatured human tendons as measured by DSC. Ordinary Two-way ANOVA with Holm-Šidák's multiple comparisons test. For (F-H): ($N = 3-6$ independent donors/group, except for systemic sclerosis where different anatomical tendons were derived from a single donor).

"It also seems counterintuitive that levels of cross-linking associated genes are decreased in fibrotic conditions which would imply decreased, rather than increased cross-linking. Please comment on this important point."

This is a good point. Fosl-2^(Tg) mice are genetically engineered whereby Fosl2 transgene was randomly inserted under the control of the major histocompatibility complex class I antigen H2Kb (MHCI) promotor. These mice develop spontaneous phenotype (e.g. dermatitis and skin fibrosis) with variable onset of inflammation that depends on the mouse sex and age.

Against the backdrop of this knowledge, we speculated that the low expression of *Lox* and *Lox2* could be due to the inherent variability of the onset of the inflammation/fibrosis phenotype. To begin to address this point, we checked our bulk transcriptomic dataset (Fra2^(Tg) vs. WT) for the mRNA expression of all cross-linking associated genes that are known to be involved in connective tissue fibrosis and/or tendon pathologies, including: Lysyl oxidase (*Lox*), Lysyl oxidase-like family (*Lox1*, *Lox2*, *Lox3*, *Lox4*) and Tissue transglutaminase-2 (*Tgm2*).

We performed the sequencing experiment with tendon fascicles from 28 mice [early phenotype $N= 15$ (7 WT + 8 Tg) and established phenotype $N= 13$ (6 WT + 7 Tg)] that are different from the ones used in the RT-qPCR data in (Figure 5D). Interestingly (and confirming the RT-qPCR data), there was no statistical difference in the mRNA differential expression of all cross-linking associated genes between the Fosl-2^(Tg) and WT controls in the early phenotype (Rebuttal figure 3 A, New Supplementary figure 17). In contrast, total *Lox2* and *Lox3* transcripts were significantly downregulated in established Fosl-2^(Tg) tendons compared to the WT controls. Established Fosl-2^(Tg) tendons had significantly lower expression of *Lox*, *Lox2*, *Lox3*, *Lox4*, *Tgm2* genes compared to the Fosl-2^(Tg) early phenotype (Rebuttal figure 3 A, New Supplementary figure 17). This data increased our confidence that the decreased expression levels of cross-linking associated genes in fibrotic Fosl-2^(Tg) is not bias from the inherent variability of the phenotype onset.

Given that the LOX-activatable collagen peptide-sensor revealed that Fosl-2^(Tg) tendons with established fibrosis had considerably higher in-situ LOX enzymatic activity, it is plausible that:

1. the peak for the expression of cross-linking associated genes was somewhere in between the two time-points that we examined in this study; possibly between 9-13 week. Furthermore, the regulation of collagen crosslinking is a multi-step process involving both intracellular and extracellular phases. The initial steps of lysine oxidation to form reactive aldehydes occur intracellularly, while subsequent crosslink formation happens extracellularly. This separation of processes adds layers of regulation and potential points of modulation that can dissociate gene expression from actual enzymatic activity.(7)

and/or

- the downregulation of cross-linking associated genes is part of a tissue stiffening-mediated global transcriptional repression of mechanosensitive and matrisome-dependent gene programs in Fosl-2 fibrotic tendons. (Discussed in detailed below. Section: Mechanical confinement and RNA polymerase II repression of transcription).

Expression of matrix crosslinking enzymes in (Fosl-2^{Tg}) vs. WT) tendons

Rebuttal figure 3 | Expression of matrix crosslinking enzymes in Fosl-2^{Tg} mice.

A) Normalized RNA-Seq expression values of matrix crosslinking enzymes in tail tendons of (Fosl-2^{Tg}) vs. Wild-type) mice. (Fosl-2^{Tg}): $N = 7$ mice – WT: $N = 6$ mice). Horizontal lines indicate the mean. ANOVA Tukey's multiple comparisons test.

“How do the authors think that altered crosslinking may ultimately result in the pro-fibrotic activation of fibroblasts observed?”

We hypothesize that elevated mechanical tension/stiffening (due to altered crosslinking) mediates the pro-fibrotic activation of fibroblasts. While we acknowledge that our study did not delve into establishing direct causal relationship between elevated matrix tension, pro-fibrotic activation of tendon fibroblasts, and transcriptional repression of matrisome and cross-linking associated genes, we speculate on several possibilities that may explain these counterintuitive observations.

1. Extracellular matrix and mechanical activation of growth factors:

It is widely appreciated that the ECM plays key roles in regulating cellular behaviour by tuning the availability and bioactivity of bioactive growth factors (GFs).(8) ECM - through its organization, mechanical properties, and proteolytic activity – tunes the growth factors activity serving as a reservoir for morphogens that orchestrate tissue patterning during development or wound healing responses.(9, 10) Many GFs, including TGF- β 's, VEGFs, FGFs, IGFs and HGF, bind with high affinity to different ECM components.(11-14) While GFs can be liberated from the ECM through the proteolytic degradation of the ECM proteins, matrix mechanical tension and cell-generated forces also contribute to the regulation of GFs.(15)

Several studies have shown that the mechanical state of extracellular matrix primes resident fibroblasts towards fibro-inflammatory activation by controlling the activation efficacy of ECM-bound growth factors.(16, 17) Of most relevance to this work is the mechanical activation of the pro-fibrotic growth factor TGF- β 1. TGF- β 1 is a potent growth that drives activation of stromal cells into pro-fibrogenic myofibroblasts. It is secreted by cells in the extracellular space in a complex, inactive form as a homodimer that is non-covalently associates with its Latency Associated Pro-peptide (LAP).(17-19) This process serves as a “safety” mechanism that prevents

an unintended TGF- β 1-mediated pro-fibrotic activation of fibroblasts, thus temporally uncoupling the release of TGF- β 1 from its activation. In this work, cells encapsulated within rigid boundaries exerted higher traction forces, and it is plausible to speculate that cell traction forces might have led to the dissociation of latent TGF- β 1 from its LAP domain.

This might lead to a feed-forward signaling loop that propagates the α -SMA-mediated cellular contractility and eventually contributes to the highly tensioned matrix state.

However, while this hypothesis may explain the increase in α -SMA signal secondary to mechanical rigidity in our work, it does not explain the transcriptional repression of *Col1a1* in the rigid boundary conditions. Recent work focusing on collagen-producing stromal cells in lung fibrosis revealed a counterintuitive duality between *Acta2* (i.e. α -SMA gene) and *Col1a1* transcription. Using single-cell RNA-seq and bleomycin-induced mouse model of lung fibrosis, Tsukui *et. al.* showed that cells characterized by high *Acta2* expression demonstrated low *Col1a1* expression.(20) Additionally, the authors analyzed human clinical samples from idiopathic pulmonary fibrosis and systemic sclerosis lungs and found that *ACTA2* transcripts poorly correlated with *COL1A1* transcription at single-cell level.(20) Taken together (with the findings of our study), this suggests that α -SMA expression does not exclusively identify pathogenic fibroblasts that produce the highest level of collagen type I. It raises the possibility that the simultaneous incorporation of α -SMA and the increase in expression of *Col1a1* is potentially not a universal phenomenon in all fibrotic diseases, or may reflect a prioritization of different cellular tasks during the natural course of disease progression.(21)

2. Mechanical confinement and RNA polymerase II repression of transcription:

Another potential explanation for the observed pro-fibrotic activation of fibroblasts and the reduced transcription of collagen and crosslinking-related genes is that these processes are mediated through two distinct mechanisms, and that the spatial confinement of cells in the stiff, highly tensioned tissues mediates transcriptional repression of matrisome genes.

Several lines of evidence from tendon and other biological systems corroborate this hypothesis. Experiments by Zhou, Franklin and colleagues using mouse embryonic fibroblasts revealed that proliferation of fibroblasts and expression of matrisome-associated genes (e.g. *Serpine1* and *Col2a1*) is significantly limited by spatial confinement.(22) The authors of this work used the term “physical space availability”; a synonymous term for spatial confinement. They attributed this space availability/confinement-dependent rewiring of gene expression programs to the differential activation of Hippo-YAP and TGF- β signaling pathways. Spatially confined fibroblasts (i.e. high density culture) displayed higher fluorescent intensity of cytoplasmic YAP1 and downregulation of *Serpine1* and *Col2a1* mRNA transcription, whereas less confined cells (i.e. low density culture) exhibited increased fluorescent intensity of nuclear YAP1 and transcription of ECM genes.(22)

Additionally, Work by Grinstein and colleagues demonstrated that the global transcription of *Col1a1*, *Col3a1*, and *Mki67* (proliferation marker) is significantly repressed when tendons transition from cell proliferative-driven embryonic growth to ECM expansion homeostasis postnatally.(23) This phase of tendon development is accompanied by significant mechanical stress with increased accumulation of collagen matrix and tissue tensioning resulting in spatial confinement of tendon cells. As tendons increase in length and ECM density, the cellular proliferation rate and transcription of ECM genes significantly decreased between postnatal day P0 and P35.(23) Similarly, Podolsky *et. al.* showed that although total collagen content was significantly higher in aged and fibrotic lungs, and *Col1a1* and *Col1a2* transcripts were universally suppressed in comparison to young non-fibrotic controls.(24) While these studies did not mechanistically explain the potential mechanical factors driving these observations, we speculate that the downregulation of cellular proliferation and *Col1a1*, *Col1a2*, and *Col3a1*

transcripts is driven, at least in part, by the spatial confinement of tendon cells during the ECM expansion and/or stiffening.

In support of this argument, Dudaryeva *et. al.* engineered 3D micro niches with tunable confinement volume.(25) They subsequently showed that human mesenchymal stromal cells (MSCs) demonstrate marked decrease in 1) proliferation rates, and 2) YAP1 nuclear activation upon 3D confinement in comparison with 2D planar cultures.(25) Similarly, encapsulating tendon progenitor cells within stiff, highly crosslinked alginate niches spatially confined tendon cells in 3D and restricted the degree of cellular spreading.(26) Interestingly, spatial confinement of tendon cells (in stiff niches) correlated with marked repression of *Col1a1* transcripts compared to less-confined cells in softer niches. Such phenotype of *Col1a1* repression was also observed when kidney-like organoids are confined in 3D matrices and compared to non-confined cultures in air–liquid interface.(27) While we think all these studies provide circumstantial evidence, it collectively points to a unifying principle by which spatial mechanical confinement might mediate the global transcriptional repression of mechanosensitive and matrisome-dependent gene programs.

The precise molecular mechanism(s) by which mechanical confinement regulates global transcription rates still need to be worked out. However, one avenue that warrants further investigations is the potential crosstalk between mechanical cues and the RNA Polymerase II (RNAPII) enzyme. Active RNAPII is a hallmark of ongoing mRNA transcription elongation. It has been previously shown that stiffness-mediated mechanical stress in models of skin aging correlates with notable reduction in cellular proliferation as well as in the phosphorylation RNAPII-S2P.(28) ChIP–seq experiments in these studies have confirmed a reduced genome occupancy of RNAPII in aged skin compared to young controls. It is indeed challenging to decouple the effect of aging processes from the aging-mediated tissue stiffening and increased confinement in *in vivo* mouse models of aging. However, supporting evidence from alternative mouse models of tissue stiffening (i.e. age-matched Collagen XIV–/– and Tnx–/–) and from 3D organoid confinement have demonstrated the essential role of mechanical stress in reducing RNAPII occupancy and eventually in attenuating global transcription.(28, 29)

Collectively, we contemplate that the myofibroblastic/fibrogenic activation and the transcriptional repression of matrisome-related genes are likely mediated by two distinct mechanisms.

An abridged version of these points is included in the revised manuscript in lines [485-542].

3. It is unfortunate that fibrotic samples could only be obtained from 1 human donor, and the methods do not describe how this sample was obtained, or if donor characteristics were matched with healthy tendon samples. However, the authors have attempted to address this limitation by investigating response of fibrotic cells derived from a mouse model of systemic sclerosis. Please show representative images of hydrogel compaction comparing WT and Fra2 cells in Fig. 5. Did the authors establish if there were differences in α -SMA/F-actin and cytoskeletal alignment between cells from normal and fibrotic tendons (both in human and mouse experiments)? Were the same panel of genes assessed for both human and tendon experiments? Supplementary table 1 seems to be missing from the supplementary information.

Human systemic sclerosis donor: We apologize for the oversight in documenting the sample provenance and procedure. Further, we acknowledge that including one human donor is a limitation of our study. Nonetheless, we consider that these specimens are exceptionally rare, and despite our best effort, only one systemic sclerosis autopsy had been available for research at our specialized tertiary center over a period of six years.

To avoid any overstatement, we now emphasized the confounding effect of ageing and the single donor as a limitation in the discussion [Line: 542-546], which reads: “We acknowledge that we prioritized comparing the cellular traction forces and gene expression of SSc-derived cells to healthy cells originated from young donors, limiting our ability to exclude the potential confounding effect of ageing.”

We have also updated the Methods sections to include details about the collection process of the SSc donor. Line [685-694] reads as following:

“SSc tendons were collected from a single donor postmortem at the time of autopsy. Hamstring, supraspinatus, infraspinatus, subscapularis, patellar and Achilles tendons were collected by autopsy technicians under the supervision of a senior pathologist. The patient fulfilled the American College of Rheumatology (ACR)/European League Against Rheumatism (EULAR) criteria for SSc.(104) The female patient was 76 years old, and had a disease duration of from first non-Raynaud-symptom of 21 years and was suffering from diffuse cutaneous SSc with a modified Rodnan skin score as a measure of skin fibrosis of 14/51 two days before her death. The patient was positive for high-titer anti-nuclear Abs on immunofluorescence with an AC-3 pattern. Consistently, she was positive for anti-centromere Abs on ELISA. She refused taking any disease-modifying anti-rheumatic agents including immunosuppressive drugs during the course of her disease.”

WT and Fosl-2^(Tg) in vitro experiments: We have now included the representative images of hydrogel compaction in supplementary figure [Supplementary figure 19 A-D]. Furthermore, we attempted to perform immunostaining on α -SMA WT and Fosl-2 cells, as suggested by the reviewer. Unfortunately, despite our best effort we were unable to obtain reliable immunofluorescence data on mouse-derived cells due to the non-specific binding of primary antibodies.

“Were the same panel of genes assessed for both human and tendon experiments?”

RT-qPCR gene panels: We used a limited panel of genes which included key matrisome-related and cross-linking associated genes that are most relevant to tendon biology (i.e. *Col1a1*, *Col3a1*, *Lox*, *Loxl2*, *Loxl4*, *Tgm2*). However, depending on the goal of the experiment, we have expanded the basic panel to include: 1) known markers indicative of tenogenic lineage commitment of stromal cells (*Scx*, *Tnmd*, *Mkx*) as in (Figure 3K: Rat tendon-derived cells), or 2) established markers of fibroblast activation implicated in fibro-inflammatory diseases as in (Figure 4H and I: Human SSc cells).

To facilitate the comparability across figures, tissues and cell types, we have included the expression of *Lox*, *Loxl2*, *Loxl4*, *Tgm2* genes in rat tendon-derived cells in (Rebuttal figure 4A, New supplementary figure 4A). We have also plotted the normalized expression values of matrix crosslinking enzymes and fibroblast activation markers in Fosl-2^(Tg) and wild-type control mice, which is collated in (Rebuttal figures 3 and 4B, New supplementary figures 17 and 18A).

Supplementary table 1 is now included in the Supplementary information file.

A

B

Rebuttal figure 4 | Expression of matrix crosslinking enzymes and fibroblast activation markers.

(A) mRNA expression of ECM crosslinking enzymes in rat tail tendon stromal cells tethered to different mechanical rigidities. (n = 8 replicates/group from 3 independent experiments, with each data point representing a Δ Ct value of 2-3 pooled tissues, horizontal lines indicate the median, Two-tailed Mann Whitney test). All individual gene expression is shown normalized to *Eif4a2* and *Gapdh* reference genes. (B) Normalized RNA-Seq expression values of fibroblast activation markers in tail tendons of (*Fosl-2*^(Tg) vs. Wild-type) mice. (*Fosl-2*^(Tg): N = 7 mice – WT: N = 6 mice). Horizontal lines indicate the mean. ANOVA Tukey's multiple comparisons test.

- The co-culture experiments are not described in the methods, and the results of these experiments are not discussed in light of the other findings of this study, nor in regard to the relevant literature.

Thank you for the catch. We have updated the Methods section with the co-culture experiments, which reads:

“Tendon fibroblasts bone marrow-derived macrophages direct co-culture experiments:
 For co-culture experiments, M-CSF pre-differentiated BMDM were pooled from 2-3 mice per genotype (WT or *Fosl-2*(Tg)). Pooled differentiated BMDM were mixed with WT or *Fosl-2*(Tg) tendon fibroblasts at 1:1 mixing ratio, in a mix-and-match fashion, and at a density of 4x10⁶ cell/ml. Pre-mixed macrophage-fibroblast populations were encapsulated in collagen hydrogels and cultured under variable tension at a final seeding density of 1x10⁶ cell/ml, as described above. Co-cultures were maintained in DMEM - high glucose media supplemented with 10% FBS, 1% Penicillin-Streptomycin, 1% MEM Non-essential Amino Acid Solution (M7145, SAFC Sigma-Aldrich) and 200 μ M L-ascorbic acid phosphate magnesium salt. Tissue tension was quantified by monitoring the post deflection for up to 48 hours.”

This point is now addressed in the Methods section Line: [718- 726].

To be able to make reliable statements, we have repeated the experiments with tendon stromal cells and BMDM macrophages that are derived from additional new mice to increase the number of the biological units and therefore increase the confidence in our results (Rebuttal figure 5,

Figure 7). We have now toned down the statements and re-interpreted the results according to the new findings.

We expanded the Results section to include the new data from the co-culture experiments. Line [419 - 434] reads:

“In monocultures, WT tendon stromal cells showed modest reduction in Col1a1 expression under rigid boundary relative to compliant controls. In contrast, total mRNA of Col1a1 in Fosl-2^(Tg) fibroblasts remained unchanged in rigid boundary compared to compliant conditions (Figure 7D). Similarly, there was no change in Col3a1 or Dcn expression in fibroblast monocultures. Interestingly, co-culture of WT fibroblasts with BMDM increased the expression of Col1a1, which was more pronounced under rigid mechanical boundaries (p= 0.76 in compliant, p= 0.42 in rigid), and when WT fibroblasts were co-cultured with Fosl-2^(Tg) BMDM (p= 0.04 in compliant, p= 0.03 in rigid), (Figure 7D). While similar trends were observed when BMDM were co-cultured with Fosl-2^(Tg)-derived fibroblasts, expression of Col1a1 was highly variable and statistically insignificant irrespective of the mechanical boundaries. mRNA expression of Col3a1 and Lox followed similar trends to Col1a1, with one difference that Col3a1 expression was only statistically significant when WT fibroblasts were in co-culture with Fosl-2^(Tg) BMDM under rigid boundaries (p= 0.03) (Figure 7D). Similarly, co-culture with BMDM induced significant shifts in the expression of cross-linking enzyme Loxl2 which was most pronounced in the conditions where Fosl-2^(Tg) fibroblasts were in co-culture with WT or Fosl-2^(Tg) BMDM under rigid boundary conditions (p= 0.05 and 0.03, respectively) compared to mono-cultures (Figure 7E).”

And discussed in light of relevant literature in Lines [562- 578].

Rebuttal figure 5 | Analysis of gene expression in macrophage-fibroblast co-culture experiments

(A) Experimental design for monocultures and (B) “mix-and-match” direct co-cultures. (C) mRNA expression of ECM-related genes, and (D) ECM crosslinking enzymes in monoculture and direct cocultures of Fosl-2^(Tg) and WT cells tethered to different mechanical boundary rigidities. (Mono-cultures: $n = 4$ replicates/group, with cells derived from 4 different mice. Co-cultures: $n = 7-8$ replicates/group, with tendon cells derived from 4 different mice. BMDM cells were from two different pools of cells derived from 2 different mice and pooled together. Each data point represents fold-change value, horizontal lines indicate the median. Two-way ANOVA (Genotype, co-culture mix) with Dunnett’s *post-hoc* test).

Abbreviations: (WT): Wild-type. (BMDM) Bone marrow-derived macrophages.

5. Please check figure legends carefully – in Figure 4, panels are described that are absent from the figure itself. Please clarify if immunofluorescence was performed on these samples.

Thank you for the catch. We updated Figure 4 legend and removed the mislabeling.

6. Please check the manuscript carefully for typos, particularly the methods section

Multiple typos have been corrected.

Reviewer #2 (Remarks to the Author):

Interesting study looking into how dysregulated matrix mechanics affect cell phenotype towards a pro-fibrotic phenotype using tendon-derived stromal cells and systemic sclerosis (SSc) as a model. Elegant methodology with the development of a new mechanoculture platform to support the investigation of the effect of matrix stiffness on the activation of fibroblasts, with validation of dysregulated matrix mechanics in the tendons of the Fra2 mouse model of systemic sclerosis, and use of tendon-derived stromal cells from Fra2 transgenic mice and wild type mice to identify functions affected by dysregulated matrix mechanics. The authors succeed to recreate a pro-fibrotic phenotype similar to the one induced by TGF β through tuning of matrix stiffness and report increased matrix stiffness negatively correlate with major pro-fibrotic ECM genes expression and positively correlate with the expression of markers of stromal-immune interactions and that co-culture with bone marrow-derived macrophages can regulate gene expression of tendon-derived stromal cells independently of matrix mechanics. In addition, the developed mechanoculture platform has great potential for future studies in the mechanism of fibroblast activation through dysregulated matrix mechanics.

Major points

1. How does the matrix stiffness generated by the compliant and the rigid PDMS mats relate to physiologic tendon matrix stiffness? How did you determine the rigid PDMS creates an “abnormal” or higher than physiologic matrix stiffness and the compliant substrate recreates the physiologic tendon microenvironment closer?

Stiffness of 3D hydrogels can be modified by changing the intrinsic mechanical properties (i.e. elastic modulus), or by manipulating the extrinsic boundary rigidity – i.e. by constraining the edges of isotropic fibrillar hydrogels (e.g. collagen biopolymers). Here, our strategy involved tuning the elasticity of the PDMS mats to control the rigidity of each cantilever post. Post rigidity determines the degree to which it deflects in response to collective (horizontal) cellular traction forces of tendon cells that are encapsulated within the gel. The two posts anchor and uniaxially constrain the contracting hydrogel, thus restricting its deformation and increasing the overall resistance and *effective stiffness* of the gel.

Indeed, the tissue-level stiffness of natural biopolymer-based 3D gels is several orders of magnitudes softer than the bulk stiffness of tendon tissues.(30-33) It is noteworthy to mention that our goal here was not to tissue-engineer a physiologic tendon tissue, but rather to examine the role that cell-level (matrix) mechanical tensional cues play in regulating cellular behavior. Despite our best efforts, and although multi-scale mechanical testing is one of our laboratory’s core expertise, we could not reliably apply a measurement approach that we felt could meaningfully measure the elastic moduli of the collagen hydrogels in tension. The methodological work to meaningfully connect cell/matrix-level to tissue-level mechanics is substantial, and remains an ongoing effort for our lab.

“How did you determine the rigid PDMS creates an “abnormal” or higher than physiologic matrix stiffness and the compliant substrate recreates the physiologic tendon microenvironment closer?”

Our rationale for creating a uniaxial anisotropic (i.e. aligned) tendon-like constructs is informed by our previous work where we showed that microtissues with uniaxial constraints appear to be beneficial for tenogenic expression (*Scx* and *Mkx*).⁽³⁴⁾ Based on this, we further benchmarked the tenogenic expression rigid vs. compliant tethered constructs which revealed that compliant constructs had significantly higher expression of *Scx* and *Tnmd* genes in comparison with rigid boundary conditions (Figure 3K).

We have now better clarified these points in the introduction [Lines: 87-102], and results sections [221-224].

2. Were technical replicates used for statistical analysis instead of biological replicates (except for the SSc where this was unavoidable)? Was a statistician consulted for the use of technical replicates instead of biological for the statistical analysis?

Yes. As background, we clarify below how we defined the “biological unit” for replication.

For all experiments involving the molecular characterization of Fosl-2^(Tg) mice, i.e. gene expression and cross-links analysis, we defined single animals as the individual biological unit for replication.

When it comes to the “mechanical” experiments- i.e. the mechano-culture of hydrogels under tension- we find the simplistic intuition behind the conventional dichotomy of “biological” vs. “technical” replication breaks down for multiple reasons. Individual gels compact and develop tension in a way that highly varies between gels, even when cells were originating from the same biological donor (whether human donors or rodents). Our experience consistently showed that the inter-gel/inter-fascicle variability is always higher than the inter-donor/inter-animal variability.(35)

To overcome this conceptual challenge, we followed the guiding statistical principles of Lazic *et. al.* who argued that “*the frequent and common distinction made between ‘biological’ and ‘technical’ replication is unhelpful because they are inconsistently defined, do not capture the important characteristics of an experiment, and do not clarify what to replicate.*”(36, 37) Lazic attributes this shortcoming to oversimplification of the ‘*biological*’ vs. ‘*technical*’ dichotomy which fails to capture the complex, multi-level hierarchy of biological organization. They further argued that it is often that the hypothesis being tested, the experimental interventions, and the technical measurements may each operate at distinct levels of biological organization, thus creating ambiguity regarding which level should dictate the sample size for replication. In an ideal scenario, the hypothesis, interventions, and readouts would all align at the same level of biological organization. However, more often than not, this is not the case as it is the case with our mechano-culture experiments.

Lazic and colleagues propose a hierarchical organization of experimental design differentiating between biological units (BU), experimental units (EU), and observational units (OU).(37) Understanding the distinction between these units clarifies where replication should take place. They recommend there are three criteria that must be fulfilled for an experimental unit to qualify as a genuine replicate: 1) Experimental Units (EUs) must be allocated independently to each experimental condition or intervention. 2) Each experimental intervention must be applied independently to every EU. 3) EUs should not exert influence on each other, particularly concerning the measured variable.

Against the backdrop of this recommendation, we defined the individual gels as the biological units (BU) and experimental (EU) for replication upon which we aim to test the hypothesis, and the boundary rigidity as the experimental intervention. This way the genuine replication at the level of individual gels (rather than the human/animal donors) is more relevant to the hypothesis we are testing which is about differences in the emergent stiffness of tensioned hydrogels as a function of the boundary rigidity. That is to say, we replicated and perturbed the individual gels and boundary rigidity as a biological unit–intervention pair as the basis for the statistical inference. We believe that this reasoning better captures the nuance related to testing our experimental hypothesis. Furthermore, we are aware of the importance of increasing confidence in any biological findings by testing cells originating from multiple donors/cell lines. As such, we performed all experiments with cell from at least 3 different donors with the statistical inference performed at the level of individual gels rather than the average of gels (from a single donor).

We acknowledge this critical nuance was not explained in sufficient details in the original submission. Therefore, we have taken several steps to clarify this point:

1. We removed the reference to “biological replication” which may create some confusion. Line [398]: Sentence “*due to the lack of biological replication*” is removed.
2. We expanded the “Statistical methods” section in lines [851] – [860], and added this discussion in the supplementary information.
3. Line 272 and Figure 5C – Since you are measuring the fascicles CSA and reporting differences in CSA between wild type and Fra2Tg fascicles why not calculate and plot failure stress instead of force to facilitate the comparison of properties between wild type and Fra2Tg fascicles? Or at least mention it in the text if there was no significant difference in failure stress measured between wild type and Fra2Tg fascicles.

In figure 5C, we followed our established practice of focusing on tendon failure force and stiffness as structural properties that reflect biologically relevant tendon function, while providing elastic modulus as a proxy measure of "material quality".

4. Line 472 – Isolation of tendon-derived stromal cells- Please include donor age and gender for SSc and healthy tissue. Were the healthy controls age-matched to the SSc donor? If not, please include in the limitations as ageing can also have an impact on cell phenotype and expression.

Apologies for the oversight. We have now compiled this information in [Supplementary Table S1].

Unfortunately, we were unable to age-match the healthy controls to the single SSc donor from the resourced materials in our biobank. However, to avoid any overstatement, we now emphasized the confounding effect of ageing and the single donor as a limitation in the discussion [Line: 542-544], which reads: “*We acknowledge that we prioritized comparing the cellular traction forces and gene expression of SSc-derived cells to healthy cells originated from young donors, limiting our ability to exclude the potential confounding effect of ageing.*”

Minor points

Line 144 – Within hours after seeding, tissue tension evidently increased after two hours, and ...- change to avoid repetition of within hours.

Line [155] reads: “*Within two hours after seeding, tissue tension evidently increased and continued to rise by 3-fold reaching a plateauing maximum of*”

Figure 1A legend – Sagittal section might be more accurate than cross section. A cross section view of the well could be included to show the initial set up view from the top/bottom of the well.

Corrected to “*Sagittal section views*”.

Figure 1A – “PDMS matt” - mat is misspelled in the actual figure.

Corrected to “*mat*” throughout the manuscript.

Figure 1G or Line 148 – Was the increase and decrease statistically significant?

Yes, the increase in tissue tension was statistically significant at 24 hr compared to the two-hour timepoint (Rebuttal figure 6, New supplementary figure 1C). Line [155] reads: “*Within two hours*

after seeding, tissue tension evidently increased and continued to rise by 3-fold reaching a plateauing maximum of $(272 \pm 60) \mu\text{N}$ at 24 h ($p= 0.00142$), (Figure 1G top, Supplementary figure 1C)."

Rebuttal figure 6 | Quantification of tissue tension of rat tail-derived tendon stromal cells in Vehicle control condition. Kruskal-Wallis test with Dunn's *post-hoc* test.

Figure 1G – Was the provenance of these cells also rat tail tendon? Maybe add in the legend. Yes, that is correct. Legend of Figure 1G now reads: "Time-course of cell-generated traction forces of rat tail-derived tendon stromal cells (Vehicle control)."

Line 154 – "in in" repetition.

Corrected

Figure 1I – Is the "active forces" column representing the vehicle tissue tension after 48 h? As the magnitude does not appear to correspond to the magnitude of the data point in 1G.

Correct, the "active forces" and "residual tension" bars represent the values at the end of Blebbistatin treatment (~ 54 hr). Thank you for spotting the inconsistency – we found incorrectly transposed values in (Figure 1H). We have updated the figure accordingly.

Line 177 – "consistent with" appears that part of the sentence missing.

Corrected. Line [188- 190] reads: "TGF- β 1-treated conditions had approximately three-fold significantly higher fluorescence intensities of α -SMA than untreated controls, which is consistent with state switch to myofibroblasts"

Line 190 – "the phenotype"

Corrected.

Line 200 – Maybe better to specify immune staining intensities (rather than immune intensities).

Corrected. Line [214] reads: "Rigid boundaries induced an approximately 50% increase in α -SMA immune staining intensities above compliant controls"

Line 207 – The way the sentence about the tenogenic markers is formulated it would seem Mxk is also significantly decreased which does not seem to be the case from Figure 3K – please amend accordingly.

Corrected. Line [221] reads "Similarly, expression of markers indicative of tenogenic lineage commitment of stromal cells, (*Scx* and *Tnmd*) were significantly downregulated in highly-tensioned constructs"

Figure 4C – why does line graph has n=24/group and the violin plot has 16/group – isn't the source of the data the same presented in a different way?

We realize that we were not sufficiently accurate in describing the panels. Although we aimed to have N=24 gels in all experiments, it often occurs that some gels rupture or pop off posts at the later timepoints. We have updated the Figure 4 legend, which now reads: *“(F) Quantitative analysis of tissue traction forces of tendon (top) and dermal (bottom) fibroblasts. Left insets: Evolution of tissue traction forces as function of time. Tendon: Cells from 3 anatomically different tendons of the SSc donor and 3 healthy young controls (n = 15-24 tissues/timepoint). Skin: Cells from 3 different SSc donors and 3 healthy controls (n = 22-24 tissues/timepoint). Each data point represent the mean ± SEM. Right inset: Violin plots of forces per cell, following normalization to initial seeding density. Horizontal lines indicate the median and interquartile range (n = Tendon: 15-16 tissues/group | Skin: 22-24 tissues/group)”*

Figure 4D and E legend – The legend says immunostaining quantification and the text says mRNA expression and panels appear mislabelled/the legend includes panels that do not appear in the figure. Please amend appropriately.

Corrected.

Figure 4 D-G – The x axis labelling is complex when the subcategories could just be labelled “ | compliant | rigid | ” since you have already colour-coded the healthy and SSc – just a suggestion. Same for Figure 5D.

Thank you for the suggestion. While we agree that the double labelling is redundant in (Figures 4 G-I) and 5D, we think it is useful in conveying the message of the more complex dataset with multiple comparisons in (Figure 7 D-E). Therefore, we wish to be consistent with this visualization across the whole manuscript.

Line 244 – You are not specifying whether the dysregulation is observed under compliant or rigid boundary mechanics conditions which gives the impression that this dysregulation is being observed under non-dysregulated mechanics/compliant boundary mechanics conditions. This is not the case for IL6 and IL8 though which are not significantly different between healthy and SSc tendon-derived stromal cells under compliant conditions. Please specify which conditions you are referring to or remove IL6 and IL8 if they are not significantly regulated.

We agree with the reviewer on this point. The statement is about the gene expression trend in the SSc conditions compared to the healthy controls (and irrespective of the boundary mechanics). We toned down the statement about the “significant dysregulation”, and specified that we are comparing the healthy, non-SSc compliant controls.

Line [283- 286] reads: *“Analysis of mRNA transcripts revealed that SSc-derived tendon cells populating the tethered constructs showed a gene expression signature typical of fibro-inflammatory activation in SSc, with altered expression of FN(EDA), IL6, IL8, ACTA2, PDPN and TLR4 mRNA relative to healthy, non-SSc compliant controls (Figure 4 G-I).”*

Line 252 – Is it an **inability of SSc cells to sense boundary rigidity** or **expression of COL1A1 is already lower in SSc cells so it does not get significantly lower in rigid conditions**? As for other genes eg IL6, IL8, and ICAM a significant difference in expression is noted between healthy and SSc cells under rigid conditions but not compliant – appearing to show a response to boundary rigidity in SSc cells. You also mention in the next sentence SSc-derived tendon cells show stiffness-sensitivity of stromal-immune interactions markers – unless that was “and (absence of) stiffness-sensitivity of stromal-immune interactions markers”.

We agree that our original conclusion represents strong overgeneralization, especially given that these SSc cells are originating from a single donor. As alluded by the reviewer, our data does not

exclude the possibility that expression of *COL1A1* might have already been lower in SSc cells and that it does not get significantly lower than this baseline.

We have toned down this statement by editing the text in the Results section. Line [297- 301] now reads: *“This suggests that SSc tendon fibroblasts may have reduced baseline expression of COL1A1, or that altering matrisome-related transcription in response to boundary rigidity is an intrinsic property of healthy tendon stromal fibroblasts. Together, these results demonstrate that SSc-derived tendon cells are intrinsically activated in a cell-autonomous manner, and may possess altered tension-sensitive transcription of matrisome and stromal-immune interactions markers.”*

We thank the reviewer for the insightful criticism that helped us to align this conclusion to our data.

Line 263 – “We have recently reported...” reference (65) is missing. Also, reference 65 – is missing a title, journal volume and pages.

Figure 5D – Are the p values for Fn1 and Lox missing?

Corrected, thank you for spotting that. We have updated the reference to the published version of the preprint. Reference (65) is now reference (71).

The *Fn1* and *Lox* P-values have been highlighted in Figure 5D.

Line 317 – “mirroring our observations in Fra2 tendons in vivo and SSc-derived tendon fibroblasts” but you report no significant changes in *Col1a1* expression under rigid conditions in SSc cells (and *Lox* expression is not reported) and *Lox* expression in Fra2 tendons does not appear to be significantly different from WT unless there are some significant p values missing from the graphs. Please clarify sentence if needed.

To be able to make reliable statements, we have now repeated these experiments with tendon stromal cells that are derived from additional new mice. We have now toned down the statements and re-interpreted the results according to the new findings. Line [419- 434] reads:

“In monocultures, WT tendon stromal cells showed modest reduction in Col1a1 expression under rigid boundary relative to compliant controls. In contrast, total mRNA of Col1a1 in Fosl-2^(Tg) fibroblasts remained unchanged in rigid boundary compared to compliant conditions (Figure 7D). Similarly, there was no change in Col3a1 or Dcn expression in fibroblast monocultures. Interestingly, co-culture of WT fibroblasts with BMDM increased the expression of Col1a1, which was more pronounced under rigid mechanical boundaries (p= 0.76 in compliant, p= 0.42 in rigid), and when WT fibroblasts were co-cultured with Fosl-2^(Tg) BMDM (p= 0.04 in compliant, p= 0.03 in rigid), (Figure 7D). While similar trends were observed when BMDM were co-cultured with Fosl-2^(Tg)-derived fibroblasts, expression of Cola1 was highly variable and statistically insignificant irrespective of the mechanical boundaries. mRNA expression of Col3a1 and Lox followed similar trends to Col1a1, with one difference that Col3a1 expression was only statistically significant when WT fibroblasts were in co-culture with Fosl-2^(Tg) BMDM under rigid boundaries (p= 0.03) (Figure 7D). Similarly, co-culture with BMDM induced significant shifts in the expression of cross-linking enzyme Loxl2 which was most pronounced in the conditions where Fosl-2^(Tg) fibroblasts were in co-culture with WT or Fosl-2^(Tg) BMDM under rigid boundary conditions (p= 0.05 and 0.03, respectively) compared to mono-cultures (Figure 7E).”

Line 334 – “Elevated rigidity” – compared to what? Physiologic matrix stiffness in tendons?

We argue it is the “elevated rigidity” in comparison to the compliant boundary rigidity. As discussed in the Major point #1, we showed that the compliant constructs had significantly higher expression of *Scx* and *Tnmd* transcripts compared to the rigid counterparts (Figure 3K). Based on this (and taken together with our previous work where we showed that uniaxially constrained microtissues appear to be beneficial for tenogenic expression(34)), we set the compliant boundary as the baseline for comparison throughout the study/manuscript.

Line 368 – As you are mentioning in line 370 the referenced paper in line 368 does not really investigate fibrosis and cell environments of increased stiffness so maybe using “This was further reported/This decreased transcription was further reported” or similar could be a better representation than “This was further confirmed”.

Thank you, we agree. Line [498- 501] reads *“This decreased transcription was further reported in a multi-omics single cell profiling of the aging lung which attributed the dysregulated matrix remodeling in aged lungs to the increase in transcriptional noise and aberrant epigenetic control.”*

Line 404 – Diameter and length details are missing.

Corrected. Diameter and length details are in line [592], which reads: *“Pins were (1 ± 0.006) mm in diameter, (20 ± 0.5) mm in length and spaced approximately 4 mm apart from each other.”*

Line 405 – “Each well in the top plate’s upper and ...” Corrected.

Line 412 – “mm” repetition

Device assembly paragraph (line 402) – The setup of the anchoring spheres is not included but would aid in the overall understanding of the setup of the spheres in relation to the posts.

Corrected. Line [593] reads: *“To provide a better gel anchoring point, a ceramic bead (116913050CF: Lysing Matrix D, MP Biomedical) with an average diameter of 1.4 mm was glued on top of each post using medical grade, low-viscosity ethyl-based instant adhesive (LOCTITE® 4011 Prism Medical CA: Ellsworth Adhesives, UK).”*

Line 437 – “software.” comma instead? Corrected

Line 441 – to create

Corrected.

Line 541 – 90 ng – space between number and unit

Corrected.

Line 551 – “The then”

Corrected.

Line 570 – Statistics

- Two-way ANOVAs are also mentioned in the figure legends but not in the statistics paragraph.

Corrected.

- (from major points) Were technical replicates used for statistical analysis instead of biological replicates (except for the SSc where this was unavoidable)? Was a statistician consulted for the use of technical replicates instead of biological for the statistical analysis?

This point is addressed in Major point #2

- Some experiments had a wide variation of sample numbers per group eg Fig 5D n=4-9. Was that due to loss of samples in the process resulting in some groups having less samples than others or sample additions where it was deemed to be needed or cell numbers/availability?

The variability in sample numbers is due to multiple factors. In Fosl-2^(Tg) experiments, this often arises from the low RNA yield in Fosl-2 tendons (tissue crosslinking makes RNA extraction very challenging). Additionally, Fosl-2^(Tg) mice often develop strong phenotype which necessitate euthanize the animal(s) for welfare reasons – this led to relatively high dropout rate of Fosl-2^(Tg) mice from each litter relative to WT controls. Whenever we had limited number of samples, we prioritized the most important genes (e.g. *Col1a1*, *Lox*, and housekeeping genes).

Reviewer #3 (Remarks to the Author):

This manuscript describes a substantial body of work that addresses mechanical factors that may be highly relevant to the tendon abnormalities which are a hallmark of systemic sclerosis patients. The authors have developed a culture technique that allows mechanics tension to be applied in vitro and using this methodology have defined effects on fibroblasts and extended previous work that shows how mechanical tension may promote the contractile phenotype and profibrotic properties of fibroblasts. A key over-arching question is the extent to which these findings may apply to a broader population of fibroblasts that those derived from tendons.

Specific comments:

1. It would be helpful to explain more clearly how the mechanical forces are generated and different between cultures. Whilst the authors obviously have a very clear understanding it is less apparent to readers unfamiliar with the concepts of stiffness and elasticity and this should be explained to help with later interpretation of the data, especially for mutant mice at different time points.

We thank the reviewer for their positive comments, and for the opportunity to clarify these concepts.

Tissue mechanical properties, specifically stiffness, describe the relationship between applied loads and resulting deformations. In other words, how the applied forces on a tissue deform its shape. The ratio between the applied load and subsequent deformation defines the stiffness of the tissues.(38) It is important to mention that stiffness/rigidity and elastic modulus are interrelated concepts, and sometimes used interchangeably (particularly when the focus of a work is biological rather than biophysical). However, stiffness/rigidity precisely describes the tissue structural properties (force and length relationships), whereas the elastic modulus defines the properties of the material that constitute that structure.(39, 40)

In the context of this work, with a focus that is both biological and biophysical, it is useful (perhaps imperative) to differentiate between the *intrinsic stiffness* (described above), *global boundary stiffness*, and their reciprocal relationship in regulating the *cell-generated tension* and the *emergent properties* of tissue and cellular behavior.(41, 42) Cellular tension in 3D environments can be modified by changing the intrinsic mechanical properties of the environment within which cells are encapsulated (e.g. hydrogel intrinsic stiffness), or by manipulating the extrinsic boundary stiffness – i.e. by constraining the edges of isotropic fibrillar hydrogels (e.g. collagen biopolymers).

The intrinsic stiffness of hydrogels (bulk elastic modulus) can be tuned by increasing the polymer concentration or by chemically crosslink its polymer backbone. However, this typically alters the porosity (polymer mesh size) and cell binding sites (ligand density and presentation) which may confound any biological findings.(43, 44)

Alternatively, spatially constraining fibrous hydrogels restricts its deformation and increases the overall resistance within the hydrogel and its *effective stiffness*.(45-47) With two constraining pins/cantilevers, hydrogel ECM alignment is directed along the longitudinal axis of the mechanical stress while minimizing the isotropic compaction of the gel. Consequently, cells also align along the axis of the mechanical stress. In addition, by changing the compliance of the constraining pins by making it highly compliant (i.e. deflectable) or minimally compliant (i.e. rigid with minimal deflection), it is possible to tune the overall resistance of the matrix and the *emergent effective stiffness*.(48, 49) In turn, this resistance regulates the cellular contractility and the mechanical forces that the cells can exert on the hydrogel matrix.

Spatially constraining collagen hydrogels represents an attractive strategy to model tendon mechanobiology for multiple reasons: 1) it uses a purely mechanical strategy where cell-generated mechanical forces and boundary constraints guide tissue organization into highly-aligned, tendon-like structures. 2) hydrogel effective stiffness can be tuned independently of the potential confounding factors associated with altering the intrinsic properties of the material. However, it is noteworthy to point out that cell-populated fibrous hydrogels are highly dynamic, self-organizing systems. The effective stiffness of the hydrogel and the cellular responses are mutually conditioned, and emerge from reciprocal interplay between different factors over time and across different length-scales.(42, 50, 51) As such, it is only possible to control for the hydrogel properties (e.g. initial polymer concentration) at the beginning of the experiment.(46)

How the mechanical factors in our 3D mechano-culture system relate to the observations in *Fosl-2^(Tg)* mutant mice can be explained by these emergent properties. As detailed in our response to Reviewer #1 (above), we speculate that the suppression of the matrisome-related transcription is potentially mediated, at least in part, by the emergent stiffening and/or confining effect of the extracellular matrix. For more discussion addressing the potential processes mediating this response, we refer the reviewer to our detailed response to Reviewer #1. (Section: Mechanical confinement and RNA polymerase II repression of transcription).(28, 29, 52)

We updated the introduction and discussion sections, where this point is now addressed in Line [87-102] and [505- 546].

2. More extensive comparison of SSc fibroblasts should be possible even though tendon samples may not be available routinely. It would be an important control experiment to explore whether dermal fibroblasts show the same or different responses to those observed for tendon derived cells.

We thank the reviewer for this important critique that helped us to strengthen our conclusions.

Based on the reviewer's suggestion we performed this control experiment, and we provide additional data where we extensively compared dermal fibroblasts from SSc and healthy controls. We performed these experiments using human dermal fibroblasts originating from the forearm skin of patients with systemic sclerosis (SSc) and healthy control donors. Skin biopsies were obtained from patients who fulfilled the American College of Rheumatology/European League Against Rheumatism (ACR/EULAR) 2013 classification criteria.(53) We hope the reviewer finds this extended evaluation satisfactory.

First, we measured tissue traction forces of SSc dermal fibroblasts and compared it to healthy control donors [**Rebuttal figure 7A**]. Over a two-day period, we observed that SSc-derived dermal fibroblasts displayed progressive increase in tissue traction forces similar to SSc-derived tendon fibroblasts. At 24h, SSc-derived dermal fibroblasts exerted forces of approximately 23 nN per cell [Mean= 22.88 nN \pm 7.52, n= 24], while healthy control cells had ~two-fold lower tensional forces [Mean= 12.26 nN \pm 7.6, n= 24]. While the difference in the order of magnitude between SSc and healthy controls was similar in skin and tendon-derived fibroblasts, SSc dermal fibroblasts exerted higher traction forces compared to SSc-derived tendon fibroblasts at all timepoints. This difference was more pronounced at 48 hrs; where SSc dermal cells applied approximately five-fold higher forces compared to SSc-derived tendon cells. In comparison, healthy dermal fibroblasts exerted 3.5-fold higher forces relative to healthy tendon fibroblasts [**Rebuttal figure 7A**].

We have incorporated this data to (Figure 4F), and updated the results and discussion sections, where this point is now addressed in Line [269- 281].

Next, we performed gene expression analysis to characterize the fibro-inflammatory phenotypes of tensioned dermal fibroblasts, and to assess whether and how SSc-derived dermal fibroblasts respond to changes in boundary stiffness and matrix tension [Rebuttal figure 7 B-E]. Interestingly, we found that SSc dermal fibroblasts exhibited a gene expression pattern consistent with fibro-inflammatory activation with upregulation of *FN^(EDA)*, *IL6*, *IL8*, *ACTA2*, *PDPN* and *TLR4* mRNA levels, albeit with modest differences compared to tendon-derived SSc cells [Rebuttal figure 7 B and D]. Unlike tendon-derived fibroblasts, dermal fibroblasts showed significant upregulation of matrix crosslinking enzymes *LOX*, *LOXL2*, *TGM2* transcripts [Rebuttal figure 7 C]. However, there were no differences in levels of *COL1A1* and *COL3A1* transcripts between SSc and healthy control dermal fibroblasts, and irrespective of the boundary rigidity.

We have added this data to (New supplementary Figure 8), and updated the results section in Line [282 – 299] to reflect these new findings.

Rebuttal figure 7 | Mechano-culture of systemic sclerosis-derived dermal stromal fibroblasts. (A) Quantitative analysis of tissue traction forces of tendon (top) and dermal (bottom) fibroblasts. Left insets: Evolution of tissue traction forces as function of time. Right insets: Violin plots of forces per cell, following normalization to initial seeding density. Horizontal lines indicate the median and interquartile range. (B) mRNA

expression of ECM-related genes, (C) cross-links related enzymes (D) stromal activation markers, and (E) immune/inflammatory genes in dermal fibroblasts tethered to different mechanical rigidities. ($n = 8$ replicates/group from 3 independent experiments, with each data point representing ΔCt , horizontal lines indicate the median, Two-way ANOVA (boundary stiffness, disease stage) with Holm-Sidak *post-hoc* test). All individual gene expression is shown normalized to *RPL13A* and *GAPDH* reference genes.

3. It is unreliable to draw conclusions from a single patient sample. It would be helpful if additional material could be obtained from a range of SSc patients and perhaps compared with other fibrosing disorders such as Dupuytren's contracture that may be more available. Although 3 biologically independent experiments are performed it is not clear how tis was achieved with a single donor.

We agree with the reviewer, and we acknowledge that including one human SSc donor is a limitation of our study. Nonetheless, these specimens are exceptionally rare and, despite our best effort, only one systemic sclerosis autopsy had been available for research at our specialized tertiary center over a period of six years. To extend the generalizability of the SSc tendon findings, we have: (1) Provided additional supporting evidence from SSc-derived dermal fibroblasts (cells derived from 3 human SSc and control donors), as explained in our response to point no. 2. (2) Included SSc tendons and cells originating from different anatomical locations for the single SSc donor. This data is now included in (Figure 4 A-C, 4F): "*Baseline characteristics of ECM transcription, total collagen content, mature collagen crosslinks in human tendons*".

Unfortunately, and despite our best effort, we were not able to source fibroblast cells from Dupuytren's contracture at our surgical center.

"Although 3 biologically independent experiments are performed it is not clear how tis was achieved with a single donor."

Apologies for the oversight. We realized we were not sufficiently clear on how we defined the "biological unit" for replication.

When it comes to the "mechanical" experiments- i.e. the mechano-culture of hydrogels under tension- we find the simplistic intuition behind the conventional dichotomy of "biological" vs. "technical" replication breaks down for multiple reasons. Individual gels compact and develop tension in a way that highly varies between gels, even when cells were originating from the same biological donor (whether human donors or rodents).

Our experience consistently showed that the inter-gel/inter-fascicle variability is always higher than the inter-donor/inter-animal variability.(35)

To overcome this conceptual challenge, we followed the guiding statistical principles of Lazic *et. al.* who argued that "*the frequent and common distinction made between 'biological' and 'technical' replication is unhelpful because they are inconsistently defined, do not capture the important characteristics of an experiment, and do not clarify what to replicate.*"(36, 37) Lazic attributes this shortcoming to oversimplification of the 'biological' vs. 'technical' dichotomy which fails to capture the complex, multi-level hierarchy of biological organization. They further argued that it is often that the hypothesis being tested, the experimental interventions, and the technical measurements may each operate at distinct levels of biological organization, thus creating ambiguity regarding which level should dictate the sample size for replication. In an ideal scenario, the hypothesis, interventions, and readouts would all align at the same level of biological organization. However, more often than not, this is not the case as it is the case with our mechano-culture experiments.

Lazic and colleagues propose a hierarchical organization of experimental design differentiating between biological units (BU), experimental units (EU), and observational units (OU).(37) Understanding the distinction between these units clarifies where replication should take place. They recommend there are three criteria that must be fulfilled for an experimental unit to qualify as a genuine replicate: 1) Experimental Units (EUs) must be allocated independently to each experimental condition or intervention. 2) Each experimental intervention must be applied independently to every EU. 3) EUs should not exert influence on each other, particularly concerning the measured variable.

Against the backdrop of this recommendation, we defined the individual gels as the biological units (BU) and experimental (EU) for replication upon which we aim to test the hypothesis, and the boundary rigidity as the experimental intervention. This way the genuine replication at the level of individual gels (rather than the human/animal donors) is more relevant to the hypothesis we are testing which is about differences in the emergent stiffness of tensioned hydrogels as a function of the boundary rigidity. That is to say, we replicated and perturbed the individual gels and boundary rigidity as a biological unit–intervention pair as the basis for the statistical inference. We believe that this reasoning better captures the nuance related to testing our experimental hypothesis.

Furthermore, we are aware of the importance of increasing confidence in any biological findings by testing cells originating from multiple donors/cell lines. As such, we performed all SSc fibroblast experiments with cell from at least 3 different donors (for dermal fibroblasts), or 3 different anatomic origins of the single SSc tendons donor (for tendon fibroblasts). The statistical inference performed at the level of individual gels rather than the average of gels (from a single donor).

We acknowledge this critical nuance was not explained in sufficient details in the original submission. Therefore, we removed the reference to “biological replication” which may create some confusion. Line [398]: Sentence “*due to the lack of biological replication*” is removed.

4. I am somewhat confused by references to the *fosl2* mutant mice as well as Fra2 and the authors should clarify this and how the different mouse strains were examined (page 29 line 449)

Fosl-2 (Fos Like 2) and *Fra-2* (Fos-Related Antigen 2) are synonyms aliases for the same gene *Fos/2*. In this work, we used a single mouse strain that overexpresses transcription factor *Fosl-2*. To avoid confusion, we updated the text and figures with one alias (*Fosl-2^{Tg}*) to ensure consistency across the whole manuscript.

5. The transcription factor MRTFA is considered critical in mechano-sensing in fibroblasts, and it would be helpful to have this included in the relevant gene expression and protein assays to explore the impact on myofibroblasts differentiation and contractility.

We agree with the reviewer here. We performed RT-qPCR to probe the expression of *Mrtfa* in tendon fibroblasts tethered to different mechanical boundaries (**Rebuttal figure 8A**). We found no statistical significance between compliant vs. rigid boundaries ($P = 0.2345$). Furthermore, we have also plotted the normalized expression values of *Mrtfa* in *Fosl-2*(Tg) and wild-type control mice, which is collated in (**Rebuttal figure 8B**). Interestingly, expression of *Mrtfa* was significantly downregulated in *Fosl-2*(Tg) fibrotic tendons relative WT control in the Established group ($P = 0.042$). Computational analysis of key upstream regulatory transcription factors (TFs) did not predict *Mrtfa* among the key regulatory element behind the differentially expressed genes (**Rebuttal figure 8C**).

We have added part of this data to (New supplementary Figure 4), and updated the results section in Line [222 – 224] to reflect these new findings.

Rebuttal figure 8 | Expression of transcription factor *Mrtfa* in mechano-culture and *Fosl-2^{Tg}* models.

(A) mRNA expression of *Mrtfa* in rat tail tendon stromal cells tethered to different mechanical rigidities. (n = 8 replicates/group from 3 independent experiments, with each data point representing a Δ Ct value of 2-3 pooled tissues, horizontal lines indicate the median, Two-tailed Mann Whitney test). All individual gene expression is shown normalized to *Eif4a2* and *Gapdh* reference genes. (B) Normalized RNA-Seq expression values of *Mrtfa* in tail tendons of (*Fosl-2^{Tg}*) vs. Wild-type) mice. (*Fosl-2^{Tg}*: N = 7 mice – WT: N = 6 mice). Horizontal lines indicate the mean. ANOVA Tukey's multiple comparisons test. (C) Top 10 significantly enriched transcriptional regulators of shared targets in *Fosl-2^{Tg}* vs. WT tendon and lung using TRRUST module. Color code denotes $-\log_{10}(\text{p-value})$ of hypergeometric enrichment test.

- The relative importance of motility, contractility and matricellular interaction in determining the contraction in this culture system should be considered as it is unlikely that cell contractility is a major determination of tendon biomechanics that is driven more by extrinsic skeletal muscle contraction. This may be a limitation of the current approach and should be clarified.

The reviewer raises a good point. We agree that multiple cellular processes likely contribute to the emergent behavior of this mechano-culture system. While this work focused on a single variable, i.e. matrix tension as a function boundary rigidity, the versatility of the platform makes it amenable to repurposing to parse out the potential contribution of myriad of factors to mechanisms of tendon pathology. For example, the platform can be easily adjusted to investigate the role of cell migration in tendon healing (54, 55), and to isolate the contribution of deposited matricellular molecules in tendon tissue repair.(49, 56)

Although we truly appreciate the reviewer's views on the role of cell contractility, work by several groups have shown that contractility significantly contributes to tendon biomechanics and homeostasis. The work by Schiele *et. al.* investigated the cellular contribution to the mechanical properties of embryonic tendons during developmental stages.(57) They specifically focused on the effect of actin cytoskeleton on the cell-scale elastic modulus of early-stage embryonic tendons. Disruption of tenocytes cytoskeleton with blebbistatin, a myosin II inhibitor, significantly reduced tissue elastic modulus by approximately 20% compared to controls.(57)

This suggests that cytoskeletal reinforcement of tendon cells (whether contractility or passive intracellular tension) actively contributes to the structural mechanical properties of tendons, at least during early developmental stages. Furthermore, non-muscle myosin II (NMII)-dependent cellular contractility indirectly contributes to ECM secretion and alignment in tendon. Kalson *et. al.* showed that assembly and transport of collagen fibrils at the plasma membrane is facilitated by the contractile NMII actomyosin machinery, and that this process is potentially critical for tendon biomechanical homeostasis by maintaining collagen fibril alignment and by transmitting

mechanical loads back to the cell.(58) Similarly, work by Dakota and colleagues has illuminated a potential role for (NMII)-dependent contractility in regulating the secretion of newly synthesized tendon matrix proteins. Inhibition of (NMII)-dependent cellular contractility with blebbistatin significantly reduced ECM deposition by tendon fibroblasts.(59)

Collectively, this highlights that cellular contractility and the associated cell-ECM interactions are critical not only for tendon development during embryonic morphogenesis, but also for the assembly of mechanically robust extracellular matrix during healing and repair processes.

7. The findings reported here concerning inflammatory cells interfering with normal tension induced down regulation of ECM may be relevant to other conditions such as adhesive capsulitis and this could be discussed.

This is indeed a great suggestion, and we agree with the reviewer that adhesive capsulitis shares many of the fibro-inflammatory features with fibrotic tendons. Like tendon fibroblasts, capsular fibroblasts are embedded in a dense, highly-aligned stroma composed primarily of collagen type-I.(60, 61) Similarly, adhesive capsulitis pathology is characterized by increase in pro-inflammatory cytokines, infiltration of immune cells, fibroblasts activation, and a collagen type-I to type-III matrix switch in the synovium and joint capsule, resembling many of the molecular features of tendinopathy.(61-68)

As adhesive capsulitis pathology progresses towards fibrosis, soft tissue structures within the joint (i.e. capsule, synovium, ligaments and tendons) get thicker, stiffer and more disorganized in both human adhesive capsulitis and animal models of joint immobilization.(69-72) This results in significant changes in tissue mechanical properties. For instance, the stiffness of the shoulder coracohumeral ligament, supra/ Infraspinatus tendons are significantly higher in symptomatic patients with adhesive capsulitis than in the unaffected shoulder or healthy subjects.(73-75)Such changes in the mechanical properties are concomitantly accompanied by alterations in the underlying ECM biophysical cues that tissue-resident cell experience.(76)

Whether and how the altered mechanics of the fibrotic niche in adhesive capsulitis contributes to the proportional differences of immune cells is unknown. However, evidence from well-controlled in vitro systems revealed that matrix biophysical cues (i.e. tension and collagen alignment) accelerate the recruitment of macrophages in models of fibrosis.(77) Contractile fibroblasts remodel the surrounding ECM generating regions of tensioned and highly-aligned ECM. The active contractility of fibroblasts and the highly tensioned collagen matrix serve as far-reaching signals that attract macrophages to the proximity of contractile myofibroblasts.(78) Close interaction between macrophages and fibroblasts, via cadherin 11, creates potent profibrotic niches where TGF- β -producing macrophages and TGF- β -activating myofibroblasts sustain the progression of fibrosis through well-described feed-forward loops.(19, 77)

With regards to immune cells infiltration, recent studies that investigated the fibro-inflammatory changes associated with adhesive capsulitis identified several immune cells that are enriched in capsular tissues including macrophages, T cells and mast cells.(62, 63)

Akbar *et. al.* mapped the immune landscape in adhesive capsulitis, and discovered that the immune milieu in diseased shoulder capsules is dominated by T cells with a clear shift from macrophages to T cells-enriched populations compared to the resting control tissues.(62) Furthermore, work by Ng and colleagues reported similar observations where they further revealed that T cells and macrophages populate unique microanatomical niches in the capsular tissues of adhesive capsulitis.(79) Adhesive capsulitis tissues were enriched in T cells in the sublining layer and MERTK⁺LYVE1⁺MRC1⁺ macrophages subpopulation in the lining layer. While the authors attributed these differences to the potential activation of developmental cellular

programs, it remains to be seen if the underlying biophysical cues of the capsular matrix contribute to this differential homing of macrophages and T cells in adhesive capsulitis.(79)

To begin to address this point, we performed a set of bulk RNA-sequencing experiments on tail tendons from early and established phases of Fosl-2^(Tg) and wild-type controls. We analyzed the data in an unbiased way with the goal of potentially identifying enriched inflammatory and/or immune signatures that overlap between Fosl-2^(Tg) tendons and human adhesive capsulitis. We also performed in silico deconvolution of the cellular fractions in bulk transcriptomic signature using CIBERSORT pipeline.

First, we found that Fosl-2^(Tg) fibrotic tendons are significantly enriched in GO terms related to inflammation and immune response [New Figure 6, New supplementary figure 18 A-D]. Interestingly, in silico deconvolution of the cellular fractions predicted that Fosl-2^(Tg) tendons with established fibrosis contain significantly higher fractions of immune cells and nerves cells compared to wild-type controls [New Figure 6 J-L, New supplementary figure 16].

Next, we examined the mRNA expression of established markers of fibroblast activation in tendinopathy and adhesive capsulitis (*Cd248*, *Mcam*, *Vcam1*, *Pdpr*, *Cd34*, *Fap*). (64, 80) We found no difference between the expression levels between the Fosl-2^(Tg) and wild-type controls [Rebuttal figure 9 A, New supplementary figure 18 A]. Then we performed a comparative analysis of the whole bulk transcriptome of the Fosl-2^(Tg) tendons and capsular tissues from human patients with adhesive capsulitis [Rebuttal figure 9 B-C, New supplementary figure 18 B-D]. We found strong and significant overlap between the two datasets. Of interest was that 8 out of the 20 overlapped GO terms and signatures were related to matrisome, cell-ECM interaction and inflammatory processes [Rebuttal figure 9 C, New supplementary figure 18 B and C].

The TF-gene interactions using TRRUST database predicted that only 4 transcription factors were regulating the overlapped features between the two datasets. Interestingly, these TFs are known to be involved in tendon inflammation (RELA, NFKB1)(81) and in the cross-talk between fibroblasts and macrophages in fibrosis (SP1, SP3)(82), [Rebuttal figure 9 D, New supplementary figure 18 D].

We have added this data to the (New supplementary Figure 18, and updated the results section in Line [393 – 401] and discussion [568 – 578] to reflect these new insights.

Expression of fibroblast activation markers in Established (*Fosl-2^{Tg}*) vs. WT) tendons

Comparative analysis of bulk transcriptome of *Fosl-2^{Tg}* vs. WT fibrotic tendons and adhesive capsulitis

Rebuttal figure 9 | Gene list-based comparative analysis of mouse tendons with established fibrosis with human capsular tissue with adhesive capsulitis. (A) Normalized RNA-Seq expression values of selected fibroblast activation markers in tendons with established phenotype (*Fosl-2^{Tg}*) vs. Wild-type). (*Fosl-2^{Tg}*: $N = 7$ mice – WT: $N = 6$ mice). Horizontal lines indicate the median. Welch's t -test. (B) Circos plot of overlapped input genes and enrichment features between the two groups. Dark orange arc and purple lines represent the differentially expressed genes that are shared between the two groups. Blue lines connect the genes that fall under the same statistically significantly enriched ontology term, reflecting the extent of functional overlap between the two comparison groups. (C) Heatmap of top significantly enriched terms (GO, KEGG, canonical pathways, etc.) using Metascape. (D) Top 5 significantly enriched transcriptional regulators of shared targets in (C) using TRRUST database for transcription factor analysis. Color code depicts $-\log_{10}$ (hypergeometric p -value) of statistical test.

Abbreviations: (R) Reactome, (W) WikiPathways, (K) KEGG, (H) Hallmark, (GO) Gene Ontology, (MSigDB) Molecular Signatures Database.

References

1. Hussien AA, Niederoest B, Bollhalder M, Goedecke N, Snedeker JG. The Stiffness-Sensitive Transcriptome of Human Tendon Stromal Cells. *Adv Healthc Mater.* 2023;12(7):e2101216. Epub 2022/12/13. doi: 10.1002/adhm.202101216. PubMed PMID: 36509005.
2. Gomez B, Jr., Ardakani S, Evans BJ, Merrell LD, Jenkins DK, Kung VT. Monoclonal antibody assay for free urinary pyridinium cross-links. *Clin Chem.* 1996;42(8 Pt 1):1168-75. Epub 1996/08/01. PubMed PMID: 8697572.
3. Urena P, Ferreira A, Kung VT, Morieux C, Simon P, Ang KS, et al. Serum Pyridinoline as a Specific Marker of Collagen Breakdown and Bone Metabolism in Hemodialysis-Patients. *Journal of Bone and Mineral Research.* 1995;10(6):932-9. PubMed PMID: WOS:A1995RM87600013.
4. Seyedin SM, Kung VT, Daniloff YN, Hesley RP, Gomez B, Nielsen LA, et al. Immunoassay for urinary pyridinoline: the new marker of bone resorption. *J Bone Miner Res.* 1993;8(5):635-41. Epub 1993/05/01. doi: 10.1002/jbmr.5650080515. PubMed PMID: 8511991.
5. Seibel MJ, Robins SP, Bilezikian JP. Urinary pyridinium crosslinks of collagen: specific markers of bone resorption in metabolic bone disease. *Trends Endocrinol Metab.* 1992;3(7):263-70. Epub 1992/09/01. doi: 10.1016/1043-2760(92)90129-o. PubMed PMID: 18407110.
6. Aronoff MR, Hiebert P, Hentzen NB, Werner S, Wennemers H. Imaging and targeting LOX-mediated tissue remodeling with a reactive collagen peptide. *Nat Chem Biol.* 2021;17(8):865-71. Epub 2021/07/14. doi: 10.1038/s41589-021-00830-6. PubMed PMID: 34253910.
7. Eyre DR, Weis MA, Wu JJ. Advances in collagen cross-link analysis. *Methods.* 2008;45(1):65-74. Epub 2008/04/30. doi: 10.1016/j.jymeth.2008.01.002. PubMed PMID: 18442706; PubMed Central PMCID: PMC2398701.
8. Hynes RO. The extracellular matrix: not just pretty fibrils. *Science.* 2009;326(5957):1216-9. Epub 2009/12/08. doi: 10.1126/science.1176009. PubMed PMID: 19965464; PubMed Central PMCID: PMC3536535.
9. Shyer AE, Rodrigues AR, Schroeder GG, Kassianidou E, Kumar S, Harland RM. Emergent cellular self-organization and mechanosensation initiate follicle pattern in the avian skin. *Science.* 2017;357(6353):811-5. doi: 10.1126/science.aai7868. PubMed PMID: WOS:000408327900045.
10. Yang S, Palmquist KH, Nathan L, Pfeifer CR, Schultheiss PJ, Sharma A, et al. Morphogens enable interacting supracellular phases that generate organ architecture. *Science.* 2023;382(6673):eadg5579. Epub 2023/11/23. doi: 10.1126/science.adg5579. PubMed PMID: 37995219.
11. Iyer AKV, Tran KT, Griffith L, Wells A. Cell surface restriction of EGFR by a tenascin cytotactin-encoded EGF-like repeat is preferential for motility-related signaling. *Journal of Cellular Physiology.* 2008;214(2):504-12. doi: 10.1002/jcp.21232. PubMed PMID: WOS:000252163400025.
12. Vaday GG, Lider O. Extracellular matrix moieties, cytokines, and enzymes: dynamic effects on immune cell behavior and inflammation. *Journal of Leukocyte Biology.* 2000;67(2):149-59. doi: DOI 10.1002/jlb.67.2.149. PubMed PMID: WOS:000086630800002.

13. Wijelath ES, Rahman S, Namekata M, Murray J, Nishimura T, Mostafavi-Pour Z, et al. Heparin-II domain of fibronectin is a vascular endothelial growth factor-binding domain - Enhancement of VEGF biological activity by a singular growth factor/matrix protein synergism. *Circulation Research*. 2006;99(8):853-60. doi: 10.1161/01.Res.0000246849.17887.66. PubMed PMID: WOS:000241217000011.
14. Wang XM, Harris RE, Bayston LJ, Ashe HL. Type IV collagens regulate BMP signalling in. *Nature*. 2008;455(7209):72-U49. doi: 10.1038/nature07214. PubMed PMID: WOS:000258890200038.
15. Pakshir P, Hinz B. The big five in fibrosis: Macrophages, myofibroblasts, matrix, mechanics, and miscommunication. *Matrix Biology*. 2018;68-69:81-93. doi: 10.1016/j.matbio.2018.01.019. PubMed PMID: WOS:000438480300006.
16. Hinz B. The extracellular matrix and transforming growth factor-beta1: Tale of a strained relationship. *Matrix Biol*. 2015;47:54-65. Epub 2015/05/12. doi: 10.1016/j.matbio.2015.05.006. PubMed PMID: 25960420.
17. Wipff PJ, Rifkin DB, Meister JJ, Hinz B. Myofibroblast contraction activates latent TGF-beta1 from the extracellular matrix. *J Cell Biol*. 2007;179(6):1311-23. Epub 2007/12/19. doi: 10.1083/jcb.200704042. PubMed PMID: 18086923; PubMed Central PMCID: PMC2140013.
18. Klingberg F, Chow ML, Koehler A, Boo S, Buscemi L, Quinn TM, et al. Prestress in the extracellular matrix sensitizes latent TGF- β 1 for activation. *Journal of Cell Biology*. 2014;207(2):283-97. doi: 10.1083/jcb.201402006. PubMed PMID: WOS:000343964900013.
19. Lodyga M, Cambridge E, Karvonen HM, Pakshir P, Wu B, Boo S, et al. Cadherin-11-mediated adhesion of macrophages to myofibroblasts establishes a profibrotic niche of active TGF- β . *Science Signaling*. 2019;12(564). doi: ARTN eaao3469 10.1126/scisignal.aao3469. PubMed PMID: WOS:000455752300001.
20. Tsukui T, Sun KH, Wetter JB, Wilson-Kanamori JR, Hazelwood LA, Henderson NC, et al. Collagen-producing lung cell atlas identifies multiple subsets with distinct localization and relevance to fibrosis. *Nature Communications*. 2020;11(1). doi: ARTN 1920 10.1038/s41467-020-15647-5. PubMed PMID: WOS:000529508000005.
21. Schuster R, Younesi F, Ezzo M, Hinz B. The Role of Myofibroblasts in Physiological and Pathological Tissue Repair. *Cold Spring Harbor Perspectives in Biology*. 2023;15(1). doi: ARTN a041231 10.1101/cshperspect.a041231. PubMed PMID: WOS:000906982500004.
22. Zhou X, Franklin RA, Adler M, Carter TS, Condiff E, Adams TS, et al. Microenvironmental sensing by fibroblasts controls macrophage population size. *Proceedings of the National Academy of Sciences of the United States of America*. 2022;119(32). doi: ARTN e2205360119 10.1073/pnas.2205360119. PubMed PMID: WOS:000931973700063.
23. Grinstein M, Dingwall HL, O'Connor LD, Zou K, Capellini TD, Galloway JL. A distinct transition from cell growth to physiological homeostasis in the tendon. *Elife*. 2019;8. Epub 2019/09/20. doi: 10.7554/eLife.48689. PubMed PMID: 31535975; PubMed Central PMCID: PMC6791717.
24. Podolsky MJ, Yang CD, Valenzuela CL, Datta R, Huang SK, Nishimura SL, et al. Age-dependent regulation of cell-mediated collagen turnover. *Jci Insight*. 2020;5(10). PubMed PMID: WOS:000536037700022.

25. Dudaryeva OY, Bucciarelli A, Bovone G, Huwyler F, Jaydev S, Broguiere N, et al. 3D Confinement Regulates Cell Life and Death. *Advanced Functional Materials*. 2021;31(52). doi: ARTN 2104098
10.1002/adfm.202104098. PubMed PMID: WOS:000698859100001.
26. Marturano JE, Schiele NR, Schiller ZA, Galassi TV, Stoppato M, Kuo CK. Embryonically inspired scaffolds regulate tenogenically differentiating cells. *Journal of Biomechanics*. 2016;49(14):3281-8. doi: 10.1016/j.jbiomech.2016.08.011. PubMed PMID: WOS:000386984400020.
27. Ruitter FAA, Morgan FLC, Roumans N, Schumacher A, Slaats GG, Moroni L, et al. Soft, Dynamic Hydrogel Confinement Improves Kidney Organoid Lumen Morphology and Reduces Epithelial-Mesenchymal Transition in Culture. *Adv Sci*. 2022;9(20). doi: ARTN 2200543
10.1002/advs.202200543. PubMed PMID: WOS:000795026700001.
28. Koester J, Miroshnikova YA, Ghatak S, Chacon-Martinez CA, Morgner J, Li XP, et al. Niche stiffening compromises hair follicle stem cell potential during ageing by reducing bivalent promoter accessibility. *Nature Cell Biology*. 2021;23(7):771-+. doi: 10.1038/s41556-021-00705-x. PubMed PMID: WOS:000670884800004.
29. Nava MM, Miroshnikova YA, Biggs LC, Whitefield DB, Metge F, Boucas J, et al. Heterochromatin-Driven Nuclear Softening Protects the Genome against Mechanical Stress-Induced Damage. *Cell*. 2020;181(4):800-17 e22. Epub 2020/04/18. doi: 10.1016/j.cell.2020.03.052. PubMed PMID: 32302590; PubMed Central PMCID: PMC7237863.
30. Netti P, D'Amore A, Ronca D, Ambrosio L, Nicolais L. Structure-mechanical properties relationship of natural tendons and ligaments. *J Mater Sci-Mater M*. 1996;7(9):525-30. doi: Doi 10.1007/Bf00122175. PubMed PMID: WOS:A1996VF42600001.
31. Fessel G, Snedeker JG. Evidence against proteoglycan mediated collagen fibril load transmission and dynamic viscoelasticity in tendon. *Matrix Biology*. 2009;28(8):503-10. doi: 10.1016/j.matbio.2009.08.002. PubMed PMID: WOS:000273043400008.
32. Nam S, Hu KH, Butte MJ, Chaudhuri O. Strain-enhanced stress relaxation impacts nonlinear elasticity in collagen gels. *Proc Natl Acad Sci U S A*. 2016;113(20):5492-7. Epub 2016/05/04. doi: 10.1073/pnas.1523906113. PubMed PMID: 27140623; PubMed Central PMCID: PMC4878492.
33. Jagiello A, Castillo U, Botvinick E. Cell mediated remodeling of stiffness matched collagen and fibrin scaffolds. *Sci Rep*. 2022;12(1):11736. Epub 2022/07/12. doi: 10.1038/s41598-022-14953-w. PubMed PMID: 35817812; PubMed Central PMCID: PMC9273755.
34. Foolen J, Wunderli SL, Loerakker S, Snedeker JG. Tissue alignment enhances remodeling potential of tendon-derived cells - Lessons from a novel microtissue model of tendon scarring. *Matrix Biol*. 2018;65:14-29. Epub 2017/06/22. doi: 10.1016/j.matbio.2017.06.002. PubMed PMID: 28636876.
35. Wunderli SL, Widmer J, Amrein N, Foolen J, Silvan U, Leupin O, et al. Minimal mechanical load and tissue culture conditions preserve native cell phenotype and morphology in tendon—a novel ex vivo mouse explant model. *Journal of Orthopaedic Research*. 2018;36(5):1383-90. doi: <https://doi.org/10.1002/jor.23769>.

36. Lazic SE. Experimental design for laboratory biologists: maximising information and improving reproducibility: Cambridge University Press; 2016.
37. Lazic SE, Clarke-Williams CJ, Munafò MR. What exactly is 'N' in cell culture and animal experiments? *Plos Biology*. 2018;16(4). doi: ARTN e2005282
10.1371/journal.pbio.2005282. PubMed PMID: WOS:000431480000029.
38. Discher DE, Janmey P, Wang YL. Tissue cells feel and respond to the stiffness of their substrate. *Science*. 2005;310(5751):1139-43. Epub 2005/11/19. doi: 10.1126/science.1116995. PubMed PMID: 16293750.
39. Guimaraes CF, Gasperini L, Marques AP, Reis RL. The stiffness of living tissues and its implications for tissue engineering. *Nature Reviews Materials*. 2020;5(5):351-70. doi: 10.1038/s41578-019-0169-1. PubMed PMID: WOS:000514941200001.
40. Chaudhuri O, Cooper-White J, Janmey PA, Mooney DJ, Shenoy VB. Effects of extracellular matrix viscoelasticity on cellular behaviour. *Nature*. 2020;584(7822):535-46. Epub 2020/08/28. doi: 10.1038/s41586-020-2612-2. PubMed PMID: 32848221; PubMed Central PMCID: PMC7676152.
41. Palmquist KH, Tiemann SF, Ezzeddine FL, Yang SC, Pfeifer CR, Erzberger A, et al. Reciprocal cell-ECM dynamics generate supracellular fluidity underlying spontaneous follicle patterning. *Cell*. 2022;185(11):1960-+. doi: 10.1016/j.cell.2022.04.023. PubMed PMID: WOS:000808171400001.
42. Pfeifer CR, Shyer AE, Rodrigues AR. Creative processes during vertebrate organ morphogenesis: Biophysical self-organization at the supracellular scale. *Current Opinion in Cell Biology*. 2024;86. doi: ARTN 102305
10.1016/j.ceb.2023.102305. PubMed PMID: WOS:001152801600001.
43. Vining KH, Mooney DJ. Mechanical forces direct stem cell behaviour in development and regeneration. *Nat Rev Mol Cell Biol*. 2017;18(12):728-42. Epub 2017/11/09. doi: 10.1038/nrm.2017.108. PubMed PMID: 29115301; PubMed Central PMCID: PMC5803560.
44. Chaudhuri O, Koshy ST, Branco da Cunha C, Shin JW, Verbeke CS, Allison KH, et al. Extracellular matrix stiffness and composition jointly regulate the induction of malignant phenotypes in mammary epithelium. *Nat Mater*. 2014;13(10):970-8. Epub 2014/06/16. doi: 10.1038/nmat4009. PubMed PMID: 24930031.
45. Kural MH, Billiar KL. Regulating tension in three-dimensional culture environments. *Exp Cell Res*. 2013;319(16):2447-59. Epub 2013/07/16. doi: 10.1016/j.yexcr.2013.06.019. PubMed PMID: 23850829; PubMed Central PMCID: PMC3876487.
46. Billiar KL. The mechanical environment of cells in collagen gel models. *The mechanical environment of cells in collagen gel models*. 2011. doi: 10.1007/8415_2010_30.
47. John J, Quinlan AT, Silvestri C, Billiar K. Boundary stiffness regulates fibroblast behavior in collagen gels. *Ann Biomed Eng*. 2010;38(3):658-73. Epub 2009/12/17. doi: 10.1007/s10439-009-9856-1. PubMed PMID: 20012205; PubMed Central PMCID: PMC2841707.
48. Zhao R, Boudou T, Wang WG, Chen CS, Reich DH. Decoupling cell and matrix mechanics in engineered microtissues using magnetically actuated microcantilevers. *Adv Mater*. 2013;25(12):1699-705. Epub 2013/01/29. doi: 10.1002/adma.201203585. PubMed PMID: 23355085; PubMed Central PMCID: PMC4037409.

49. Legant WR, Pathak A, Yang MT, Deshpande VS, McMeeking RM, Chen CS. Microfabricated tissue gauges to measure and manipulate forces from 3D microtissues. *Proc Natl Acad Sci U S A*. 2009;106(25):10097-102. Epub 2009/06/23. doi: 10.1073/pnas.0900174106. PubMed PMID: 19541627; PubMed Central PMCID: PMC2700905.
50. Negru T. Self-organization and autonomy: Emergence of degrees of freedom in dynamical systems. *Filos Unisinos*. 2016;17(2):121-31. doi: 10.4013/fsu.2016.172.05. PubMed PMID: WOS:000398325900006.
51. Keller EF. Organisms, machines, and thunderstorms: A history of self-organization, part one. *Hist Stud Nat Sci*. 2008;38(1):45-75. doi: 10.1525/hsns.2008.38.1.45. PubMed PMID: WOS:000255873000003.
52. Holt LJ, Delarue M. Macromolecular crowding: Sensing without a sensor. *Current Opinion in Cell Biology*. 2023;85. doi: ARTN 102269 10.1016/j.ceb.2023.102269. PubMed PMID: WOS:001102770000001.
53. van den Hoogen F, Khanna D, Fransen J, Johnson SR, Baron M, Tyndall A, et al. 2013 classification criteria for systemic sclerosis: an American College of Rheumatology/European League against Rheumatism collaborative initiative. *Arthritis Rheum*. 2013;65(11):2737-47. Epub 2013/10/15. doi: 10.1002/art.38098. PubMed PMID: 24122180; PubMed Central PMCID: PMC3930146.
54. Sakar MS, Eyckmans J, Pieters R, Eberli D, Nelson BJ, Chen CS. Cellular forces and matrix assembly coordinate fibrous tissue repair. *Nat Commun*. 2016;7:11036. Epub 2016/03/17. doi: 10.1038/ncomms11036. PubMed PMID: 26980715; PubMed Central PMCID: PMC4799373.
55. Griebel M, Vasan A, Chen C, Eyckmans J. Fibroblast clearance of damaged tissue following laser ablation in engineered microtissues. *APL Bioeng*. 2023;7(1):016112. Epub 2023/03/21. doi: 10.1063/5.0133478. PubMed PMID: 36938481; PubMed Central PMCID: PMC10017124.
56. Legant WR, Chen CS, Vogel V. Force-induced fibronectin assembly and matrix remodeling in a 3D microtissue model of tissue morphogenesis. *Integr Biol (Camb)*. 2012;4(10):1164-74. Epub 2012/09/11. doi: 10.1039/c2ib20059g. PubMed PMID: 22961409; PubMed Central PMCID: PMC3586566.
57. Schiele NR, von Flotow F, Tochka ZL, Hockaday LA, Marturano JE, Thibodeau JJ, et al. Actin cytoskeleton contributes to the elastic modulus of embryonic tendon during early development. *J Orthop Res*. 2015;33(6):874-81. Epub 2015/02/28. doi: 10.1002/jor.22880. PubMed PMID: 25721681; PubMed Central PMCID: PMC4889338.
58. Kalson NS, Starborg T, Lu Y, Mironov A, Humphries SM, Holmes DF, et al. Nonmuscle myosin II powered transport of newly formed collagen fibrils at the plasma membrane. *Proc Natl Acad Sci U S A*. 2013;110(49):E4743-52. Epub 2013/11/20. doi: 10.1073/pnas.1314348110. PubMed PMID: 24248360; PubMed Central PMCID: PMC3856828.
59. Jones DL, Hallstrom GF, Jiang X, Locke RC, Evans MK, Bonnevie ED, et al. Mechanoepigenetic regulation of extracellular matrix homeostasis via Yap and Taz. *Proc Natl Acad Sci U S A*. 2023;120(22):e2211947120. Epub 2023/05/22. doi: 10.1073/pnas.2211947120. PubMed PMID: 37216538; PubMed Central PMCID: PMC10235980.

60. Rangan A, Brealey SD, Keding A. Management of adults with primary frozen shoulder in secondary care (UK FROST): a multicentre, pragmatic, three-arm, superiority, randomised clinical trial (vol 396, pg 977, 2020). *Lancet*. 2021;397(10269):98-. PubMed PMID: WOS:000607269000024.
61. Millar NL, Meakins A, Struyf F, Willmore E, Campbell AL, Kirwan PD, et al. Frozen shoulder. *Nature Reviews Disease Primers*. 2022;8(1). doi: ARTN 59
10.1038/s41572-022-00386-2. PubMed PMID: WOS:000852406200002.
62. Akbar M, Crowe LAN, McLean M, Garcia-Melchor E, MacDonald L, Carter K, et al. Translational targeting of inflammation and fibrosis in frozen shoulder: Molecular dissection of the T cell/IL-17A axis. *Proc Natl Acad Sci U S A*. 2021;118(39). Epub 2021/09/22. doi:
10.1073/pnas.2102715118. PubMed PMID: 34544860; PubMed Central PMCID: PMC8488623.
63. Ng MTH, Borst R, Gacaferi H, Davidson S, Ackerman JE, Johnson PA, et al. A single cell atlas of frozen shoulder capsule identifies features associated with inflammatory fibrosis resolution. *Nat Commun*. 2024;15(1):1394. Epub 2024/02/20. doi: 10.1038/s41467-024-45341-9. PubMed PMID: 38374174; PubMed Central PMCID: PMC10876649 competing interests to declare.
64. Dakin SG, Buckley CD, Al-Mossawi MH, Hedley R, Martinez FO, Wheway K, et al. Persistent stromal fibroblast activation is present in chronic tendinopathy. *Arthritis Res Ther*. 2017;19(1):16. Epub 2017/01/27. doi: 10.1186/s13075-016-1218-4. PubMed PMID: 28122639; PubMed Central PMCID: PMC5264298.
65. Dakin SG, Newton J, Martinez FO, Hedley R, Gwilym S, Jones N, et al. Chronic inflammation is a feature of Achilles tendinopathy and rupture. *British Journal of Sports Medicine*. 2018;52(6):359-67. doi: 10.1136/bjsports-2017-098161 PMID - 29118051.
66. Kendal AR, Layton T, Al-Mossawi H, Appleton L, Dakin S, Brown R, et al. Multi-omic single cell analysis resolves novel stromal cell populations in healthy and diseased human tendon. *Scientific Reports*. 2020;10(1). doi: ARTN 13939
10.1038/s41598-020-70786-5. PubMed PMID: WOS:000571226600001.
67. Croft AP, Campos J, Jansen K, Turner JD, Marshall J, Attar M, et al. Distinct fibroblast subsets drive inflammation and damage in arthritis. *Nature*. 2019;570(7760):246-+. doi: 10.1038/s41586-019-1263-7. PubMed PMID: WOS:000471297600054.
68. Marsh LJ, Kembler S, Reis Nisa P, Singh R, Croft AP. Fibroblast pathology in inflammatory joint disease. *Immunol Rev*. 2021;302(1):163-83. Epub 2021/06/08. doi: 10.1111/imr.12986. PubMed PMID: 34096076.
69. Tamai K, Hamada J, Nagase Y, Morishige M, Naito M, Asai H, et al. Frozen shoulder. An overview of pathology and biology with hopes to novel drug therapies. *Mod Rheumatol*. 2023. Epub 2023/08/27. doi: 10.1093/mr/road087. PubMed PMID: 37632764.
70. Kim DH, Lee KH, Lho YM, Ha E, Hwang I, Song KS, et al. Characterization of a frozen shoulder model using immobilization in rats. *J Orthop Surg Res*. 2016;11(1):160. Epub 2016/12/10. doi:
10.1186/s13018-016-0493-8. PubMed PMID: 27931231; PubMed Central PMCID: PMC5146898.
71. Okajima SM, Cubria MB, Mortensen SJ, Villa-Camacho JC, Hanna P, Lechtig A, et al. Rat Model of Adhesive Capsulitis of the Shoulder. *J Vis Exp*. 2018(139). Epub 2018/10/16. doi: 10.3791/58335. PubMed PMID: 30320752; PubMed Central PMCID: PMC6235346.

72. Oki S, Shirasawa H, Yoda M, Matsumura N, Tohmonda T, Yuasa K, et al. Generation and characterization of a novel shoulder contracture mouse model. *J Orthop Res*. 2015;33(11):1732-8. Epub 2015/05/28. doi: 10.1002/jor.22943. PubMed PMID: 26014262.
73. Wu CH, Chen WS, Wang TG. Elasticity of the Coracohumeral Ligament in Patients with Adhesive Capsulitis of the Shoulder. *Radiology*. 2016;278(2):458-64. Epub 2015/09/01. doi: 10.1148/radiol.2015150888. PubMed PMID: 26323030.
74. Yun SJ, Jin W, Cho NS, Ryu KN, Yoon YC, Cha JG, et al. Shear-Wave and Strain Ultrasound Elastography of the Supraspinatus and Infraspinatus Tendons in Patients with Idiopathic Adhesive Capsulitis of the Shoulder: A Prospective Case-Control Study. *Korean J Radiol*. 2019;20(7):1176-85. Epub 2019/07/05. doi: 10.3348/kjr.2018.0918. PubMed PMID: 31270981; PubMed Central PMCID: PMC6609436.
75. Wada T, Itoigawa Y, Yoshida K, Kawasaki T, Maruyama Y, Kaneko K. Increased Stiffness of Rotator Cuff Tendons in Frozen Shoulder on Shear Wave Elastography. *J Ultrasound Med*. 2020;39(1):89-97. Epub 2019/06/21. doi: 10.1002/jum.15078. PubMed PMID: 31218712.
76. Castagna A, Cesari E, Gigante A, Di Matteo B, Garofalo R, Porcellini G. Age-Related Changes of Elastic Fibers in Shoulder Capsule of Patients with Glenohumeral Instability: A Pilot Study. *Biomed Res Int*. 2018;2018:8961805. Epub 2018/08/15. doi: 10.1155/2018/8961805. PubMed PMID: 30105260; PubMed Central PMCID: PMC6076904.
77. Adler M, Mayo A, Zhou X, Franklin RA, Meizlish ML, Medzhitov R, et al. Principles of Cell Circuits for Tissue Repair and Fibrosis. *Iscience*. 2020;23(2). doi: ARTN 100841 10.1016/j.isci.2020.100841. PubMed PMID: WOS:000518637100050.
78. Pakshir P, Alizadehgiashi M, Wong B, Coelho NM, Chen XY, Gong Z, et al. Dynamic fibroblast contractions attract remote macrophages in fibrillar collagen matrix. *Nature Communications*. 2019;10. doi: ARTN 1850 10.1038/s41467-019-09709-6. PubMed PMID: WOS:000465200300010.
79. Michael THN, Rowie B, Hamez G, Sarah D, Caio CM, Ian R, et al. Primed to resolve: A single cell atlas of the shoulder capsule reveals a cellular basis for resolving inflammatory fibrosis. *Biorxiv*. 2023:2023.01.16.522218. doi: 10.1101/2023.01.16.522218.
80. Akbar M, McLean M, Garcia-Melchor E, Crowe LA, McMillan P, Fazzi UG, et al. Fibroblast activation and inflammation in frozen shoulder. *PLoS One*. 2019;14(4):e0215301. Epub 2019/04/24. doi: 10.1371/journal.pone.0215301. PubMed PMID: 31013287; PubMed Central PMCID: PMC6478286.
81. Dakin SG, Martinez FO, Yapp C, Wells G, Oppermann U, Dean BJ, et al. Inflammation activation and resolution in human tendon disease. *Sci Transl Med*. 2015;7(311):311ra173. Epub 2015/10/30. doi: 10.1126/scitranslmed.aac4269. PubMed PMID: 26511510; PubMed Central PMCID: PMC4883654.
82. Feng P, Che Y, Gao CY, Chu XL, Li ZC, Li LG, et al. Profibrotic role of transcription factor SP1 in cross-talk between fibroblasts and M2 macrophages. *Iscience*. 2023;26(12). doi: ARTN 108484 10.1016/j.isci.2023.108484. PubMed PMID: WOS:001129943100001.

Response to reviewer 2

Point 3: Line 272 and Figure 5C – Since you are measuring the fascicles CSA and reporting differences in CSA between wild type and Fra2Tg fascicles why not calculate and plot failure stress instead of force to facilitate the comparison of properties between wild type and Fra2Tg fascicles? Or at least mention it in the text if there was no significant difference in failure stress measured between wild type and Fra2Tg fascicles.

Response: In figure 5C, we followed our established practice of focusing on tendon failure force and stiffness as structural properties that reflect biologically relevant tendon function, while providing elastic modulus as a proxy measure of "material quality".

Reply: I agree that reporting failure stress here, in addition to failure force, would be valuable, and recommend that results for failure stress are either included in the text or the figure, particularly as the authors describe generating stress-strain curves in Figure 5b.

Authors response: Thank you for pointing this out. We have reported failure stress and failure strain values in (**Supplementary figure 13 E-F**). We have now referred to failure stress and failure strain in the text, and have included their plots in a way that is consistent with the main figures.

Violin plots depict tail tendon failure stress (**E**) and failure strain (**F**). Each data point represents independent sample (i.e. fascicle). Horizontal lines indicate the median and interquartile range ($n = 18$ (Early WT), $n = 12$ (Early Fosl-2^(Tg)), $n = 23$ (Established WT), $n = 24$ (Established Fosl-2^(Tg)) fascicles from ≥ 6 mice/group. Two-way, ANOVA (genotype, disease stage) with Holm-Šidák's *post-hoc* test.

Final minor point: Some experiments had a wide variation of sample numbers per group eg Fig 5D $n=4-9$. Was that due to loss of samples in the process resulting in some groups having less samples than others or sample additions where it was deemed to be needed or cell numbers/availability?

Response: The variability in sample numbers is due to multiple factors. In Fosl-2(Tg) experiments, this often arises from the low RNA yield in Fosl-2 tendons (tissue crosslinking makes RNA extraction very challenging). Additionally, Fosl-2(Tg) mice often develop strong phenotype which necessitate euthanize the animal(s) for welfare reasons – this led to relatively high dropout rate of Fosl-2(Tg) mice from each litter relative to WT controls. Whenever we had limited number of samples, we prioritized the most important genes (e.g. Col1a1, Lox, and housekeeping genes).

Reply: It would be helpful to include this information in the supplementary information to allow the reader to understand the wide variation in sample numbers

Authors response: We agree. We have included a brief explanation in the Statistics section (Lines: 858 – 865).